# Uniform dynamics of cohesin-mediated loop extrusion in living human cells

Thomas Sabaté [1,2,3,7] ✉, Benoît Lelandais[1,4,8], Marie-Cécile Robert [2,8],
Michael Szalay[2], Jean-Yves Tinevez [4], Edouard Bertrand [2,9] ✉ &
Christophe Zimmer [1,5,6,9] ✉

Most animal genomes are partitioned into topologically associating
domains (TADs), created by cohesin-mediated loop extrusion and defined
by convergently oriented CCCTC-binding factor (CTCF) sites. The dynamics
of loop extrusion and its regulation remain poorly characterized in vivo.
Here we tracked the motion of TAD anchors in living human cells to
visualize and quantify cohesin-dependent loop extrusion across multiple
endogenous genomic regions. We show that TADs are dynamic structures
whose anchors are brought in proximity about once per hour and for
6–19 min (~16% of the time). Moreover, TADs are continuously extruded by
multiple cohesin complexes. Remarkably, despite strong differences in Hi-C
patterns across chromatin regions, their dynamics is consistent with the
same density, residence time and speed of cohesin. Our results suggest that
TAD dynamics is primarily governed by the location and affinity of CTCF
sites, enabling genome-wide predictive models of cohesin-dependent
chromatin interactions.

Genome-wide chromosome conformation capture methods (such as Hi-C[1]) revealed that mammalian genomes are folded into thousands of topologically associating domains (TADs) spanning 100–1,500 kb[2–5]. TADs are characterized by an increased frequency of internal contacts compared to outside their boundaries, appearing as blocks on the diagonal of Hi-C matrices[2,3]. Additionally, TADs can exhibit contact frequency peaks at their corners, interpreted as signatures of chromatin loops. Both TADs and loops depend on the cohesin complex, as the removal of its subunit RAD21 results in their genome-wide disappearance[6,7]. These structures arise from loop extrusion, wherein cohesin progressively enlarges a DNA loop by consuming ATP[6,8–11]. This process is halted by CCCTC-binding factor (CTCF) proteins bound to their respective sites in converging orientation, which define TAD boundaries by acting as barriers to cohesin[4,6,12–16]. Loop extrusion has

an important role in key nuclear functions such as gene expression regulation[17–22], DNA repair[23,24] and V(D)J recombination[25]; therefore, characterizing extrusion dynamics is essential to understand these processes. Critical questions include how frequently genomic regions undergo extrusion, how frequently and for how long boundaries come into contact and how rapidly DNA is extruded[6,10,11,26,27]. It also remains open how loop extrusion is regulated across the genome and whether factors beyond CTCF or cohesin modulate TAD dynamics.

TADs and loops are usually characterized using bulk Hi-C, which averages contacts over millions of fixed cells at a single time point. Although single-cell Hi-C[28] and multiplexed DNA fluorescence in situ hybridization[29–32] revealed cell-to-cell variability in chromatin structure, these techniques cannot directly characterize loop extrusion dynamics. Real-time visualization of loop extrusion in vitro measured

[1]Imaging and Modeling Unit, Institut Pasteur, Université Paris Cité, Paris, France. [2]Institut de Génétique Humaine, University of Montpellier, CNRS, Montpellier, France. [3]Collège Doctoral, Sorbonne Université, Paris, France. [4]Image Analysis Hub, Institut Pasteur, Université Paris Cité, Paris, France. [5]Rudolf Virchow Center for Integrative and Translational Bioimaging, Chair of Machine Biophotonics, Josef-Schneider-Straße 2 97080, University of Würzburg, Würzburg, Germany. [6]Center for Artificial Intelligence and Data Science, University of Würzburg, Würzburg, Germany. [7]Present address: Oncode Institute, Hubrecht Institute–KNAW and University Medical Center Utrecht, Utrecht, the Netherlands. [8]These authors contributed equally: Benoît Lelandais, Marie-Cécile Robert. [9]These authors jointly supervised this work: Edouard Bertrand, Christophe Zimmer. ✉e-mail: t.sabate@hubrecht.eu; christophe.zimmer@pasteur.fr; edouard.bertrand@igh.cnrs.fr

cohesin-mediated extrusion speeds of 0.5–1 kb s⁻¹ (refs. 10,11). However, it remains uncertain whether extrusion occurs at a similar rate in living cells, where many factors may slow down or accelerate this process. Likewise, other factors (for example, WAPL or STAG subunits) can potentially tune the loading and release rates of cohesin in vivo[33–36]. Two recent studies in living mouse embryonic stem cells (mESCs) provided crucial insights, revealing that TADs are dynamic structures whose anchors contact only rarely and briefly[26,27]. Being limited to a single locus each, however, these studies did not reveal how extrusion dynamics is regulated across the genome. Additionally, mESCs differ from differentiated cells[37–39], including less condensed chromatin and a short G1 phase (~1–2 h[40]). Thus, a broader characterization of loop extrusion in living human cells is needed.

Here we use live-cell microscopy and polymer simulations to visualize and quantitatively characterize cohesin-dependent loop extrusion across multiple endogenous TADs in human HCT116 cells. We show that TAD anchors are frequently brought together by loop extrusion, but only briefly. Moreover, we find that TADs are almost always folded into multiple loops, extruded by several cohesin complexes simultaneously. Finally, we provide evidence that cohesin dynamics is uniform across multiple genomic regions rather than locally tuned, and that TAD dynamics is primarily governed by CTCF binding.

## Results

### Visualizing the dynamics of TAD anchors

To visualize cohesin-mediated chromatin looping in living cells, we labeled endogenous TAD anchors by CRISPR-mediated insertion of TetOx96 and CuOx150 arrays, respectively bound by TetR-splitGFPx16-NLS and CymR-NLS-2xHalo, imaged with the bright and photostable dye JFX646 (ref. 41; Fig. 1a,b).

TADs span a large range of genomic sizes and exhibit a diversity of Hi-C patterns[42–44]. To account for this, we selected TADs with distinct sizes and Hi-C patterns: L1 (345 kb) and L2 (566 kb) display a strong corner peak characteristic of a loop, while T1 (918 kb) lacks a corner peak (Fig. 1c). All three domains exhibited chromatin immunoprecipitation followed by sequencing (ChIP–seq) peaks of SMC1, RAD21 and CTCF at both anchors and at least one pair of convergent and bound CTCF sites. L1 and L2 anchors each contained only one to two strong CTCF sites, while T1 contained one strong CTCF site and four weak CTCF sites at its anchors. Although all three TADs were located within the transcriptionally active A compartment, L1 and L2 exhibited weak signal for the repressive H3K27me3 histone mark, whereas T1 showed higher levels of H3K27me3 (Extended Data Fig. 1a). Moreover, these TADs lacked genes and enhancers at their anchors and displayed little or no gene expression. We also used Cre-mediated and flippase (FLP)-mediated recombination to eliminate transcription from the antibiotic resistance genes used for selecting repeat array integration, thus avoiding potential interferences of transcription with loop extrusion[45] (Fig. 1a and Extended Data Fig. 1a). We additionally generated two control cell lines. The first, hereafter called 'half TAD' (576 kb, similar to L2), has one of the two labeled loci outside the TAD, far from CTCF

sites (Extended Data Fig. 1a). Therefore, prolonged contacts between these two loci are not expected. The second, hereafter called 'adjacent', features two fluorescence reporters genomically adjacent to each other (midarray distance of 6 kb) to approximate anchors in close spatial proximity (Fig. 1c).

We first confirmed, using Capture Micro-C, that fluorescent tagging of chromatin did not disrupt TAD formation (Extended Data Fig. 1b,c). Then, we used spinning disk confocal microscopy to image live cells in three-dimensions (3D), every 30 s during 2 h (Fig. 1b,d,e and Supplementary Video 1). We excluded cells containing replicated spots to only examine cells in G1 or early S phase, and we detected, localized and tracked fluorescent spots, and computed their 3D distance as function of time (Fig. 1d–f and Extended Data Fig. 2a–g). Notably, we computed the fundamental localization precision limit[46] for each fluorescent spot, allowing us to estimate uncertainties in distance measurements. We obtained 150–694 time series per locus and experimental condition, totaling 12,269–93,431 measured distances for each, with mean uncertainties of 70–105 nm (Extended Data Fig. 2d and Supplementary Table 1).

To analyze the effect of cohesin, we homozygously inserted the auxin-dependent degron mini-AID[47] in the endogenous *RAD21* gene. Auxin treatment resulted in efficient and rapid cohesin depletion of 91% and >94% after 1 and 3 h, respectively (Extended Data Fig. 3a–e), while minimizing basal degradation thanks to the AtAFB2 ubiquitin ligase[48,49] (on average 80% of endogenous RAD21 level remained in untreated cells; Extended Data Fig. 3d,f). As expected, auxin treatment led to TAD disappearance (Fig. 1c), and to increased anchor–anchor distances and chromatin motion (Fig. 1g and Extended Data Fig. 3g,h), in agreement with prior experiments[6,26,27,50]. Thus, we could visualize endogenous TAD anchors in living cells and track their 3D distance over time, in the presence or the absence of cohesin.

### TADs are dynamic structures

At any given time, a pair of anchors occupies only one of the following three states: (1) the open state, where the DNA between the anchors is free of loops; (2) the extruding state, where one or more loops are being extruded; and (3) the closed state, where the DNA between anchors is fully extruded and the anchors are maintained in direct contact (Fig. 1b). We first aimed to quantify the fraction, frequency and lifetime of closed states.

Accurately identifying closed states is challenging because stochastic motion of chromatin can bring the anchors in close proximity even without loop extrusion (Fig. 1f). Additionally, distance measurements are affected by noise from random localization errors and the genomic distance between reporters and anchors[51,52]. Thus, as a proxy for closed states, we defined 'proximal states' as temporally sustained intervals during which anchor–anchor distances remained small (Fig. 2a and Extended Data Fig. 4a). We segmented time series into proximal states using a method involving a spatial and temporal threshold (Supplementary Table 2), previously validated on polymer simulations[51]. In untreated cells, we determined average proximal state

---

**Fig. 1 | Tracking TAD anchors in living human cells. a**, TetOx96 and CuOx150 repeat arrays were inserted at TAD anchors in HCT116 cells and visualized using TetR-splitGFPx16 (GFP11 and GFP1_10) and CymR-2xHalo[JFX646], respectively. Multimerized GFP11 fragments are not shown for clarity. Antibiotic cassettes (Bsd-TK and Neo) were removed by Cre (through loxP sites) and FLP (through FRT sites) recombinases to avoid interference of transcription with loop extrusion. A Capture Micro-C contact map corresponding to TAD L2 is shown. **b**, Cells were imaged without or with auxin treatment, which leads to RAD21 depletion. The 3D distance (*d*) between the two fluorescent reporters was computed as function of time. In absence of auxin, the chromatin region between the two anchors is in one of three states: open (no loop), extruding (that is, containing one or more DNA loop(s)) or closed (the two anchors are maintained in direct contact). **c**, Capture Micro-C maps of cells left untreated or treated with auxin

for 3 h. Green and magenta spots indicate the genomic locations of the inserted repeat arrays. The genomic distance between TAD anchors is indicated below each Micro-C map. All Micro-C maps show a 1,125-kb region, at 2-kb resolution. The T1 Micro-C map was used to illustrate the Adjacent locus. The white space in the L1 map corresponds to a region that was not covered by capture probes. **d**, Live-cell images of L1 TAD anchors at *t* = 0 min. The arrows indicate spots that were identified as not replicated. Example image from 1 of 30 image time series. Scale bars = 5 µm. **e**, Time-lapse images of the region inside the dotted white box in **d**. Scale bars = 1 µm. **f**, Time series of 3D anchor–anchor distances of the L1 TAD in untreated cells (red, corresponding to images in **d** and **e**) or auxin-treated cells (blue) and the adjacent locus (gray). **g**, CDF of 3D anchor–anchor distances. All images are maximum-intensity projections of 20 *z*-stacks. CDF, cumulative distribution function; a.u., arbitrary units.

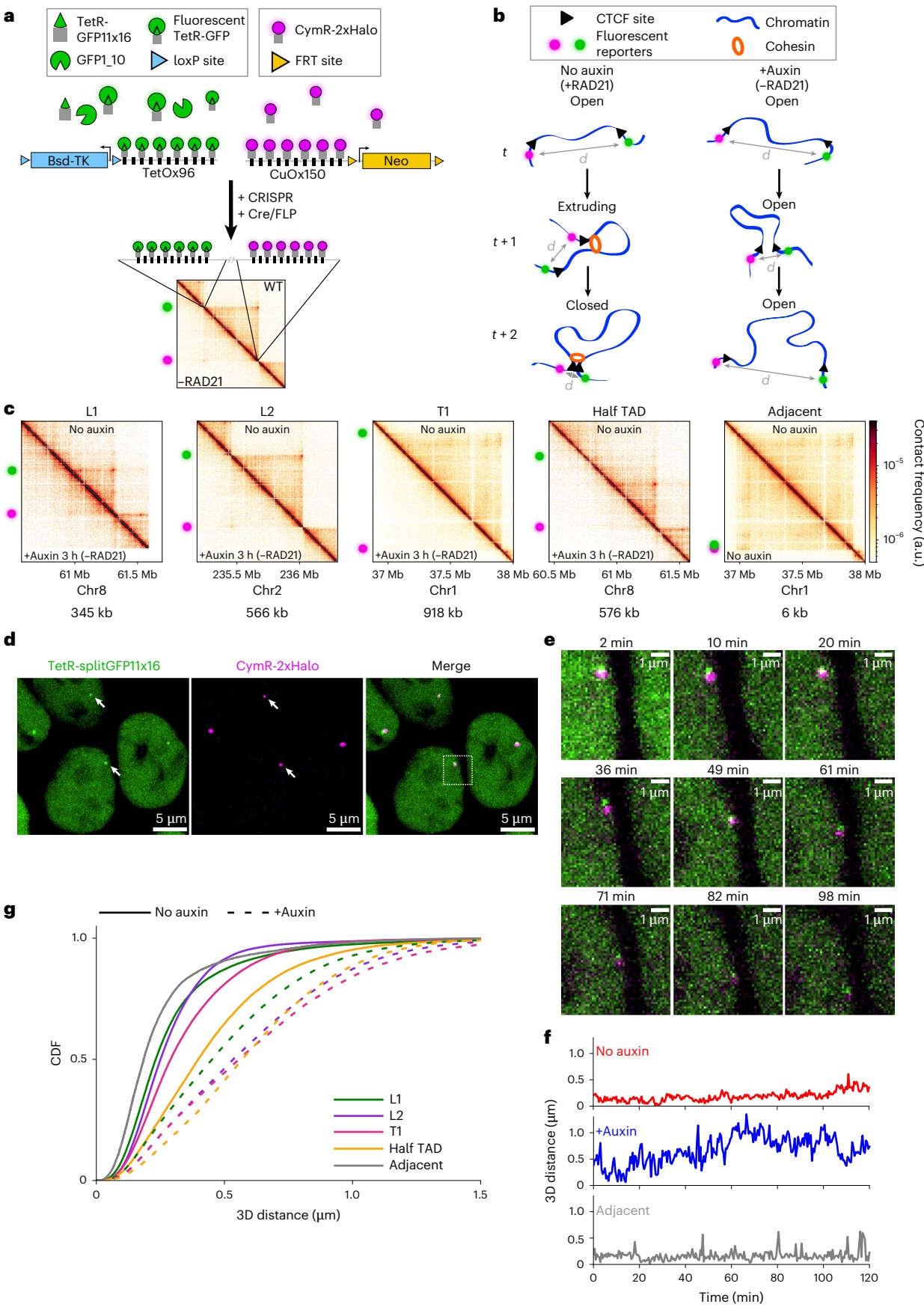

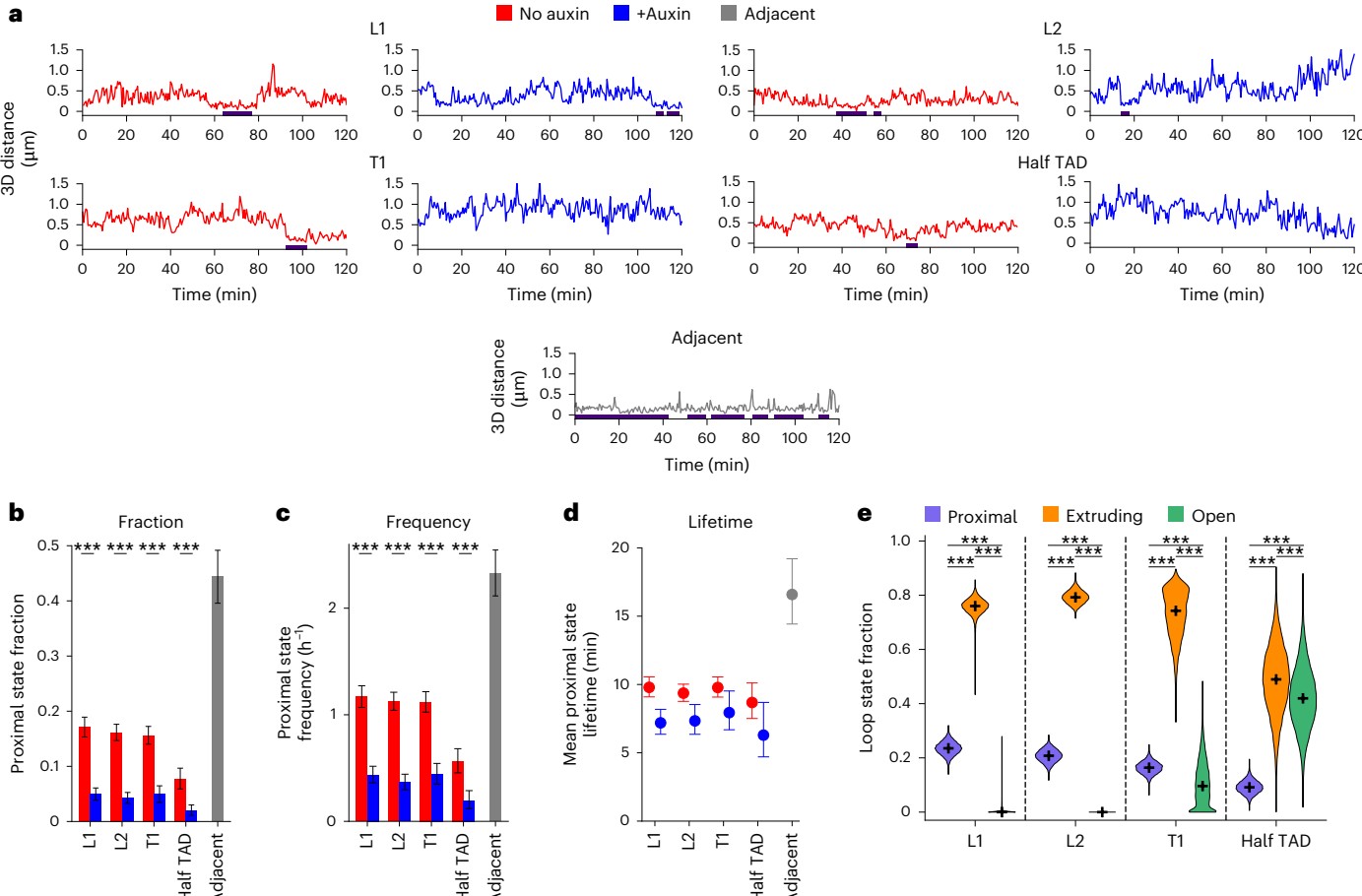

**Fig. 2 | TADs are dynamic structures. a**, Time series of 3D anchor–anchor distances with or without auxin treatment. The blue bar indicates segmented proximal states. **b–d**, Fraction (**b**), frequency (**c**) and lifetime (**d**) of proximal states. In **b** and **c**, bars represent means and errors bars show the 2.5–97.5 percentile ranges of 10,000 bootstrap samples. ***$P < 0.001$, two-sided Mann–Whitney $U$ test. Exact $P$ values can be found in Source Data. In **d**, data are represented as mean ± 95% confidence interval. The number of time

series for each condition can be found in Supplementary Table 1. **e**, Fraction of each loop state in untreated cells, estimated from an analytical model of anchor–anchor vectors. The black cross indicates the median and violin plots extend from the minimum to the maximum of $n = 10,000$ bootstrap samples. ***$P < 0.001$, post hoc Dunn's test corrected with Bonferroni correction to adjust for multiple comparisons after a Kruskal–Wallis test. Exact $P$ values can be found in Source Data.

fractions of 17% for L1, 16% for L2 and T1 and 8% for half TAD (Fig. 2b). Auxin treatment reduced these fractions by 3.1-fold to 3.9-fold, high-lighting the importance of cohesin in establishing long-range chro-matin interactions (Fig. 2b). Distances in the Adjacent locus were the smallest (Fig. 1g), but slightly larger than expected, as previously observed for adjacent chromatin loci[26,27], possibly due to increased chromatin stiffness at repeat arrays (Extended Data Fig. 4b,c). Never-theless, the Adjacent locus exhibited the highest fraction of proximal states (44%; Fig. 2b).

Next, we estimated the frequency of proximal states, that is, the number of transitions to proximal states per hour. Proximal states occurred on average 1.1–1.2 h⁻¹ for L1, L2 and T1, 0.6 h⁻¹ for half TAD and 2.3 h⁻¹ for the Adjacent locus while auxin-treated cells exhibited lower frequencies (-0.4 h⁻¹), as expected (Fig. 2c). Finally, we estimated mean proximal state lifetimes of 9.8 min for L1 and T1, 9.4 min for L2 and 8.7 min for half TAD, and the longest lifetimes (16.6 min) for the Adja-cent locus (Fig. 2d and Extended Data Fig. 4d). Auxin treatment short-ened lifetimes by only -1.3-fold (Fig. 2d), a counterintuitive result that nonetheless agrees with polymer simulations (Extended Data Fig. 4e).

To assess the robustness of these estimates on the temporal thresh-old, we varied it such that proximal state fractions in auxin-treated cells ranged from 1% to 10%. In untreated cells, this resulted in mean proximal state fractions of 8–26%, 8–25%, 7–24% and 3–12% for L1, L2, T1 and half TAD, respectively (Extended Data Fig. 4f). Therefore, proximal states

always represented a minor fraction of loop states. The mean proximal state frequency varied in the range of 0.3–2.7 h⁻¹ for L1 and L2, 0.3–2.6 h⁻¹ for T1 and 0.2–1.4 h⁻¹ for half TAD (Extended Data Fig. 4g). Although mean proximal state lifetimes slightly depended on the threshold, their averages remained within the ranges 6–19 min for L1 and T1, 6–16 min for L2 and 5–15 min for half TAD (Extended Data Fig. 4h).

In summary, we found that proximal states between TAD anchors occur about 3–27 times during a 10-h G1 phase and last for 6–19 min, for a total of 0.8–2.6 h in G1.

## TADs are constantly extruded

We then quantified the fractions of each loop state (proximal, open and extruding) using a different method, also validated on simula-tions, which relies only on the distribution of anchor–anchor vec-tors and on knowing the proximal and open state distributions separately[51] (Extended Data Fig. 5a). We used segmented proximal states in untreated cells for the former distribution and data from RAD21-depleted cells for the latter.

In auxin-treated cells, we estimated open state fractions of 89–94%, proximal state fractions of 9–11% for L1, L2 and T1 and 5% for half TAD (similar to estimates obtained by segmenting proximal state intervals; Fig. 2b), and extruding state fractions of 0% for all four regions (Extended Data Fig. 5b), consistent with the disappearance of chromatin loops following RAD21 depletion[6] (Fig. 1c). In untreated

cells, proximal state fractions increased to 24%, 21%, 16% and 9% for L1, L2, T1 and half TAD, respectively (Fig. 2e), in agreement with our previous results (Fig. 2b). Most strikingly, we found that the open state was completely absent in L1 and L2 (median state fractions of 0%) and accounted for only 11% in T1 and 42% in half TAD (Fig. 2e). By contrast, we estimated extruding state fractions of 76%, 79%, 73% and 49% for L1, L2, T1 and half TAD, respectively. This quasi-absence of open states in TADs was confirmed by a model-free analysis showing that open state fractions cannot exceed 18%, 4% and 5% for L1, L2 and T1, respectively (Extended Data Fig. 5c). These findings indicate that TADs are rarely, if ever, in a completely relaxed (open) state but almost continuously undergo cohesin-dependent loop extrusion.

## Quantification of TAD dynamics is robust to spatiotemporal resolution

Quantifications of TAD dynamics from live-cell microscopy inevitably face limits in spatiotemporal resolution. To evaluate the impact of these limitations on our conclusions, we first compared experimental time series with polymer simulations (described below) that included or excluded localization errors. Distance fluctuations primarily arose from stochastic chromatin motion rather than from localization errors, indicating that our measurements reflect biological variations in anchor distances (Extended Data Fig. 6a–c).

Second, we applied a stringent quality filtering to raw distance time series, which eliminated on average threefold more time points (22–70% versus 8–26%) than in the original dataset (Extended Data Fig. 6d,e,g,h). Despite this reduced measurement noise, the estimated fractions of proximal states remained close to our initial estimates: 13–15% with stringent filtering versus 16–17% originally for L1, L2 and T1, and 6% versus 8% for half TAD. Similarly, proximal state frequencies and lifetimes, as well as the loop state fractions, were largely unaltered (Extended Data Fig. 6d,e).

Third, we acquired live-cell images at a higher frequency (one image per 9 s instead of 30 s) for L2 and half TAD. Strikingly, proximal state fractions remained similar: 14% versus 16% for L2 and 9% versus 8% for half TAD in the high frequency and original datasets, respectively. Likewise, proximal state frequencies, lifetimes and loop state fractions remained largely unaffected (Extended Data Fig. 6d,f). Thus, our findings that proximal states are transient and that TADs most often undergo extrusion are robust to both spatial and temporal resolution.

## TAD anchors are brought together at rates of ~0.1 kb s$^{-1}$

Next, we aimed to determine the speed at which TAD anchors are brought together. We hereafter distinguish the two following quantities, both expressed in bp s$^{-1}$: (1) the motor speed, that is, the molecular speed at which cohesin pulls out DNA strands to form a loop (estimated at 0.5–1 kb s$^{-1}$ in vitro[10,11]) and (2) the closing rate (CR), that is, the rate at which the unextruded DNA between anchors is reduced (Fig. 3a). For a single cohesin complex extruding DNA without obstacles, the CR equals the motor speed. However, in the presence of multiple cohesin complexes and internal CTCF sites where cohesin can stall, these quantities may differ.

We first focused on the CR. Reasoning that time points preceding closed states should be undergoing extrusion, we aligned distance time series on the start of proximal states[51] ($t_{start}$), ignoring proximal states preceded by another one (Fig. 3b). The progressive decrease in anchor–anchor distance expected from extrusion can be obscured by stochastic chromatin motion and localization errors[51]. To reduce these fluctuations, we averaged many time series[51] and fitted a constant plateau followed by a linear decrease to the averaged squared distances (Fig. 3c). We used the slope to estimate the CR unless the decrease was not supported by at least three time points (Extended Data Fig. 7a).

As a control, we randomly shuffled all time points within a single time series, thereby destroying any signature of processive dynamics (Fig. 3b,d). Randomly shuffled time series were consistent with constant distances, as expected, in ~90% of bootstrap samples. By contrast, ~97% of bootstrap samples in unshuffled time series showed a linear decrease, as expected from processive loop extrusion (Extended Data Fig. 7b). We estimated median CRs of 0.07 kb s$^{-1}$ for both L1 and L2, 0.16 kb s$^{-1}$ for T1 and 0.09 kb s$^{-1}$ for half TAD (Fig. 3e) and similar CRs of 0.08–0.14 kb s$^{-1}$ using time series with stringent quality filtering (Extended Data Fig. 6d,e).

Because the relation between CR and cohesin motor speed is complicated by the locus-specific locations and binding affinities of CTCF sites, we then turned to polymer simulations to better understand the impact of cohesin dynamics on TAD structure and motion.

## Polymer modeling of cohesin-dependent and CTCF-dependent TAD dynamics

To investigate how cohesin dynamics, together with CTCF site locations and affinities, affect contact patterns and TAD anchor dynamics, we simulated 2.6-Mb-long polymers centered on each genomic region, accounting for cohesin and CTCF binding dynamics to chromatin. We independently varied cohesin residence times (2–33 min), density (1–40 Mb$^{-1}$), considered two motor speeds (0.25 and 1 kb s$^{-1}$) and assumed that cohesin stalls at CTCF with 50% probability (Fig. 4a), encompassing previous estimates[9,26,27,35,53–58]. CTCF ChIP–seq peaks were used to determine the local residence time of CTCF at each binding site, assuming a genome-wide residence time of 2.5 min[53,59] (Fig. 4b).

From the simulated polymer conformations, we computed contact maps and p($s$), the averaged contact frequencies as function of genomic distance $s$. The simulated contact maps featured TADs, corner peaks and stripes (Fig. 4c). Longer cohesin residence times yielded longer loops and flatter p($s$) curves, while higher cohesin densities produced sharp stripes and increased overall contact frequencies (Extended Data Fig. 8a). Reducing cohesin motor speed from 1 to 0.25 kb s$^{-1}$ at constant residence time shortened loops, reflecting reduced genomic processivity (Fig. 4d).

Next, we generated simulated time series of anchor–anchor distances, including reporter–anchor separations and random localization errors consistent with experiments, and increasing with time due to photobleaching. Even without extrusion, stochastic polymer motion occasionally brought anchors together (Fig. 4e). Notably, closed state fractions did not exceed 12%, even at a cohesin density of 40 Mb$^{-1}$ (Extended Data Fig. 8b). Nevertheless, increasing cohesin density reduced overall distances, and six cohesin complexes per Mb sufficed to reduce distances by ~50% (Extended Data Fig. 8c). The collective action of multiple cohesin complexes therefore greatly contributed to reducing anchor–anchor distances.

Increasing cohesin density constrained anchor motion, as reflected by a decrease in the plateau of two-point mean-squared displacement (MSD) curves (Extended Data Fig. 8d), in agreement with previous studies[26,27]. Interestingly, cohesin residence time had a nonmonotonous effect. For intermediate residence times (10–20 min), cohesin complexes reached TAD anchors and constrained their motion. By contrast, shorter or longer residence times caused cohesin complexes to fall off before reaching anchors or to bypass them, respectively, in both cases reducing constraints on anchor motion (Extended Data Fig. 8d–f). Polymer simulations thus predict the complex interplay between cohesin dynamics and CTCF site locations and affinities.

## CTCF binding and uniform cohesin dynamics govern TAD dynamics

Analyzing multiple genomic regions allowed us to ask whether cohesin dynamics varies across the genome. Using polymer simulations, we determined the cohesin parameters consistent with both live-cell imaging and Micro-C data, considering the four chromatin regions independently. Differences between these parameters across genomic domains may indicate locus-specific and CTCF-independent regulation of TAD

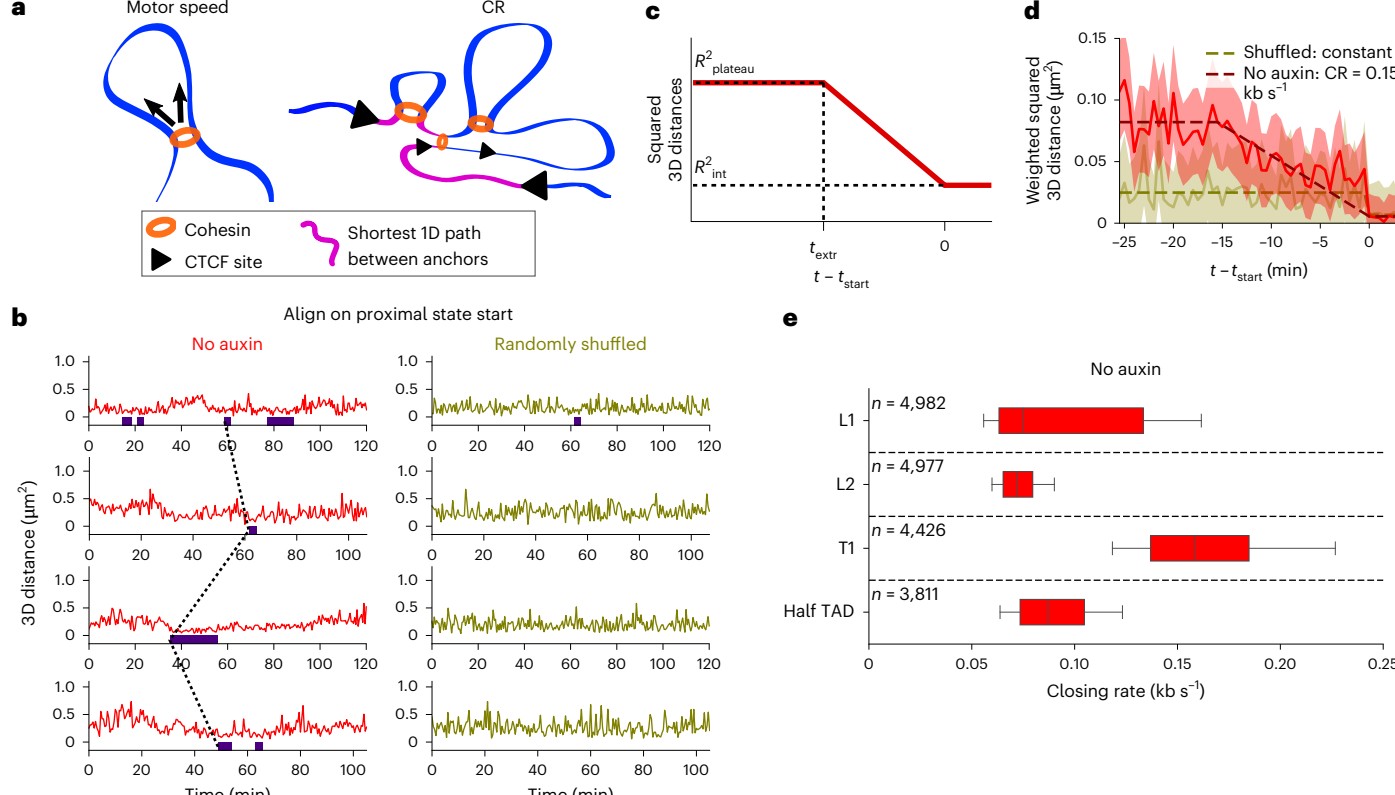

**Fig. 3 | TAD anchors are brought together at rates of ~0.1 kb s⁻¹ in living cells.** **a**, Cohesin motor speed and CR are two different measures of extrusion speed. The cohesin motor speed is the number of DNA base pairs extruded by a single cohesin complex per unit time on DNA devoid of obstacles such as CTCF sites (left). The CR measures the rate at which the effective loop-free DNA between TAD anchors—the shortest 1D path between them (pink)—decreases due to extrusion (right). The CR reflects the action of single or multiple cohesin complexes extruding simultaneously, as well as cohesin stalling at internal CTCF sites. Cohesin complexes not involved in the shortest 1D path do not contribute to the CR. **b**, Distance time series of the T1 TAD without auxin treatment before (left) and after (right) random shuffling of time points. Indigo bars indicate segmented proximal state intervals. Dotted lines indicate alignment of the

starting times of proximal states ($t_{start}$) across single time series. **c**, Fitting strategy to determine CRs. A piecewise linear model with three parameters ($R^2_{plateau}$, $R^2_{int}$, $t_{extr}$) is fitted to the mean squared 3D anchor–anchor distances as function of time after alignment to $t_{start}$. The slope of the linear decrease defines the CR. **d**, Aligned time series of mean-squared 3D distances measured on the T1 TAD, weighted by localization precisions, before (red, $n = 46$ time series) or after (olive, $n = 19$ time series) random shuffling. The piecewise linear fit is shown as a dashed line. The shaded area represents the weighted s.e.m. **e**, CRs estimated from time series of untreated cells. Box plot lines show the median, lower and upper quartiles, and whiskers extend from 10th to 90th percentiles; $n$ indicates the number of bootstrap samples including at least 15 time series and associated with nonconstant distances out of a total of 5,000 bootstrap samples.

dynamics, whereas similar values would indicate uniform cohesin dynamics across these regions.

For Micro-C data, we considered the p(s) curve and, for imaging data, we considered two-point MSD curves, anchor–anchor distances, and fractions, frequencies and lifetimes of proximal states (Extended Data Fig. 9a). We varied cohesin density and residence time, initially assuming a motor speed of 1 kb s⁻¹, and computed the deviation of polymer simulations from experimental data (Fig. 5a). Before analyzing experiments, we validated our parameter estimation method with polymer simulations. We successfully recovered the ground truth cohesin density for residence times longer than 5 min, covering the 5–24-min range of experimental estimates[35,53,55,57,58], and identified a broad range of cohesin residence times that correctly encompassed the ground truth value (Extended Data Fig. 9b). Thus, our approach accurately estimated cohesin density and provided a broad but reliable range for residence times. Turning to experiments, we first separately considered Micro-C and live-cell imaging data. The best parameter combinations estimated separately from Micro-C or imaging datasets were in excellent agreement with each other for L1 and L2, and overlapped for T1 and half TAD (Fig. 5b), thereby providing reciprocal validation from two completely orthogonal experimental techniques.

We then analyzed Micro-C and live-cell imaging data together. Strikingly, cohesin parameters estimated independently across the

four chromatin regions largely agreed with each other (Fig. 5c). Thus, the observed dynamics of these four regions could be reproduced using the same cohesin density and residence time, with differences in anchor dynamics arising solely from differences in CTCF binding. In auxin-treated cells, the different metrics used to compute the deviation between simulations and experiments yielded divergent parameter estimates (Extended Data Fig. 9c), suggesting that a biological process not included in our simulations may influence chromatin dynamics in the absence of cohesin, whereas loop extrusion dominates TAD anchor dynamics in the presence of cohesin.

We then sought to estimate cohesin motor speed. Because both motor speeds of 1 and 0.25 kb s⁻¹ allowed similar agreements between experiments and simulations (Fig. 5a and Extended Data Fig. 9d), we turned to a different approach and compared CRs measured in experiments to simulations with cohesin motor speeds of 0.125, 0.2, 0.25, 0.5 or 1 kb s⁻¹ (Fig. 5d,e). For computational tractability, we only varied the motor speed, setting cohesin density to 12 Mb⁻¹ and residence time to 22 min, a parameter combination consistent with experiments for motor speeds of both 0.25 and 1 kb s⁻¹ (Fig. 5c). We first considered distance time series generated by simulations with varying motor speeds. For motor speeds of 0.5 and 1 kb s⁻¹, we failed to detect the progressive reduction in distances expected from loop extrusion (68% of bootstrapped samples yielded flat profiles; Fig. 5e and Extended Data Fig. 9e).

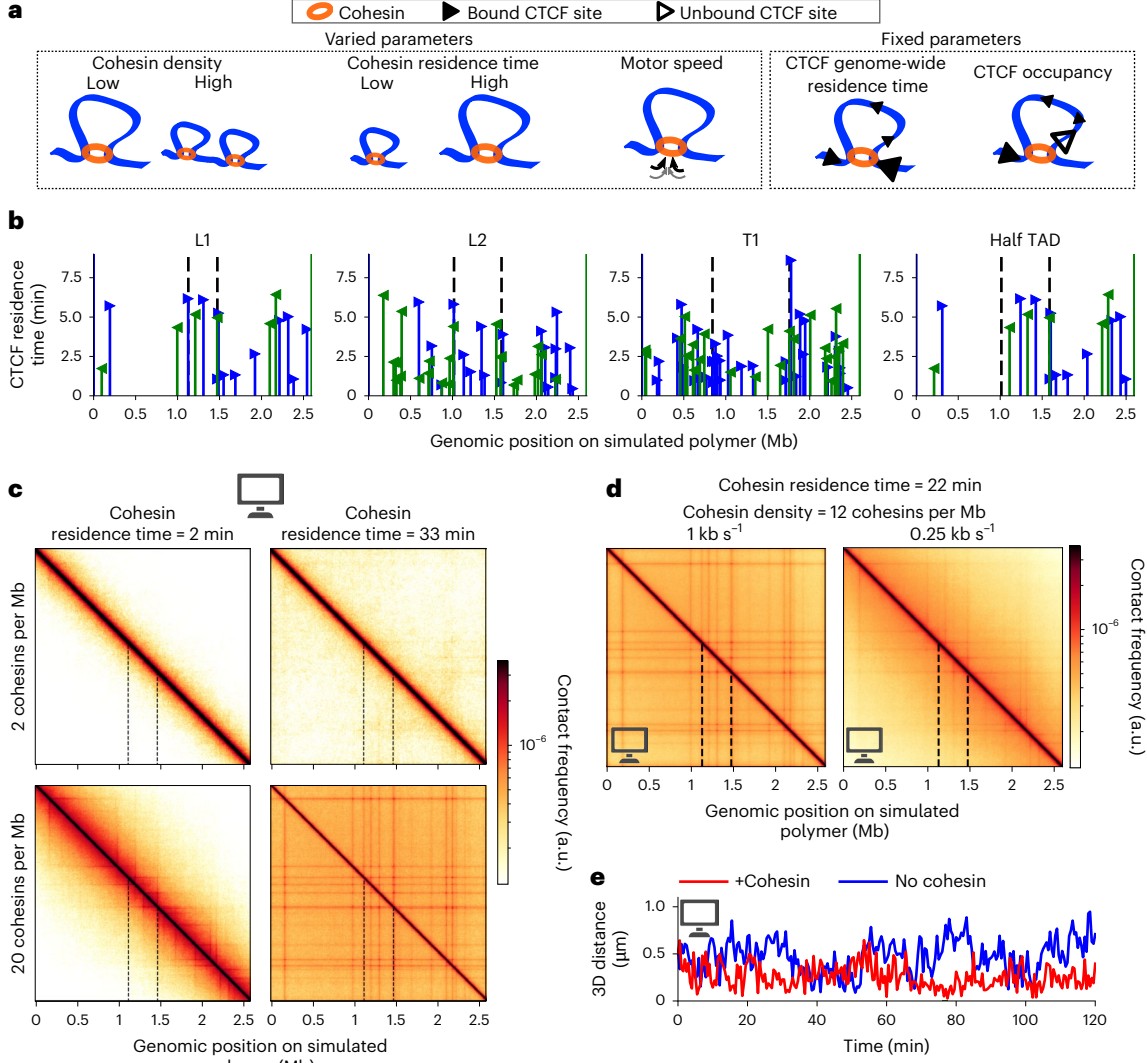

**Fig. 4 | Polymer simulations of cohesin- and CTCF-dependent TAD dynamics. a**, Parameters used to model loop extrusion in polymer simulations. Three parameters characterizing cohesin dynamics (density, residence time and motor speed) were varied systematically, while the CTCF genome-wide residence time and occupancy of CTCF sites were held constant. **b**, CTCF residence times used for simulations at the four genomic regions. The color and orientation of arrowheads indicate CTCF site orientation, with height reflecting CTCF residence time. **c**,**d**, Contact maps from polymer simulations of the L1 TAD for different combinations of cohesin residence time, density and motor speed. In **c**, the

motor speed is fixed to 1 kb s$^{-1}$; in **d**, the cohesin density and residence time are fixed to 12 Mb$^{-1}$ and 22 min, respectively. In **b**–**d**, dashed black lines indicate the location of TAD anchors (or of fluorescence reporters for the half TAD locus). **e**, Simulated 3D anchor–anchor distance time series of the L1 TAD. Simulations include random localization errors and photobleaching consistent with experiments, and reporter–anchor separations. Simulations assume a cohesin density of 12 Mb$^{-1}$, a residence time of 22 min and a motor speed of 0.25 kb s$^{-1}$. Panels **c**–**e** were created in part with BioRender.com.

Therefore, speeds of 0.5 kb s$^{-1}$ or higher cannot be measured at our imaging frequency (Extended Data Fig. 9e–g). At speeds of 0.125, 0.2 and 0.25 kb s$^{-1}$, however, we could detect the expected distance decrease for all four genomic regions (in ~86% of bootstrap samples; Extended Data Fig. 9e). Critically, we consistently detected this linear decrease in experimental data from all genomic regions (~97% of bootstrap samples; Extended Data Fig. 7b). Thus, our analysis indicates that cohesin extrudes DNA at speeds below 0.5 kb s$^{-1}$ in living cells.

Together, our results argue against strong variations in cohesin density and instead indicate that this parameter is uniform across different genomic regions. While we cannot rule out differences in residence times or motor speeds across regions, our data are also consistent with uniform values of these parameters. Therefore, the observed variations in TAD dynamics across different genomic regions can be explained without major changes in cohesin dynamics, but rather by differences in CTCF site locations and binding strengths.

## Implications for TAD dynamics

To further characterize TAD dynamics, we analyzed polymer simulations consistent with experimental data. We fixed cohesin density, residence time and motor speed to 12 Mb$^{-1}$, 22 min and 0.25 kb s$^{-1}$, respectively, a parameter combination matching experimental data across all four regions (Fig. 5c). Using these parameters, simulations reproduced experimental contact maps (Fig. 6a) and p(s) curves (Extended Data Fig. 9h), two-point MSD curves (Extended Data Fig. 9i) and anchor–anchor distance distributions (Fig. 6b,c) for L1, L2 and T1 at once, although agreement was poorer for half TAD (Fig. 5a). Notably, closed state fractions were <1% for all TADs, indicating that direct anchor–anchor contacts are rare (Extended Data Fig. 10a–e).

We estimated the number of loops connecting TAD anchors by computing the shortest one-dimensional (1D) (unextruded) DNA path between them. TAD anchors were connected by 2.5–3.7 internal loops on average (Fig. 6d, Extended Data Fig. 9j and Supplementary Video 2)

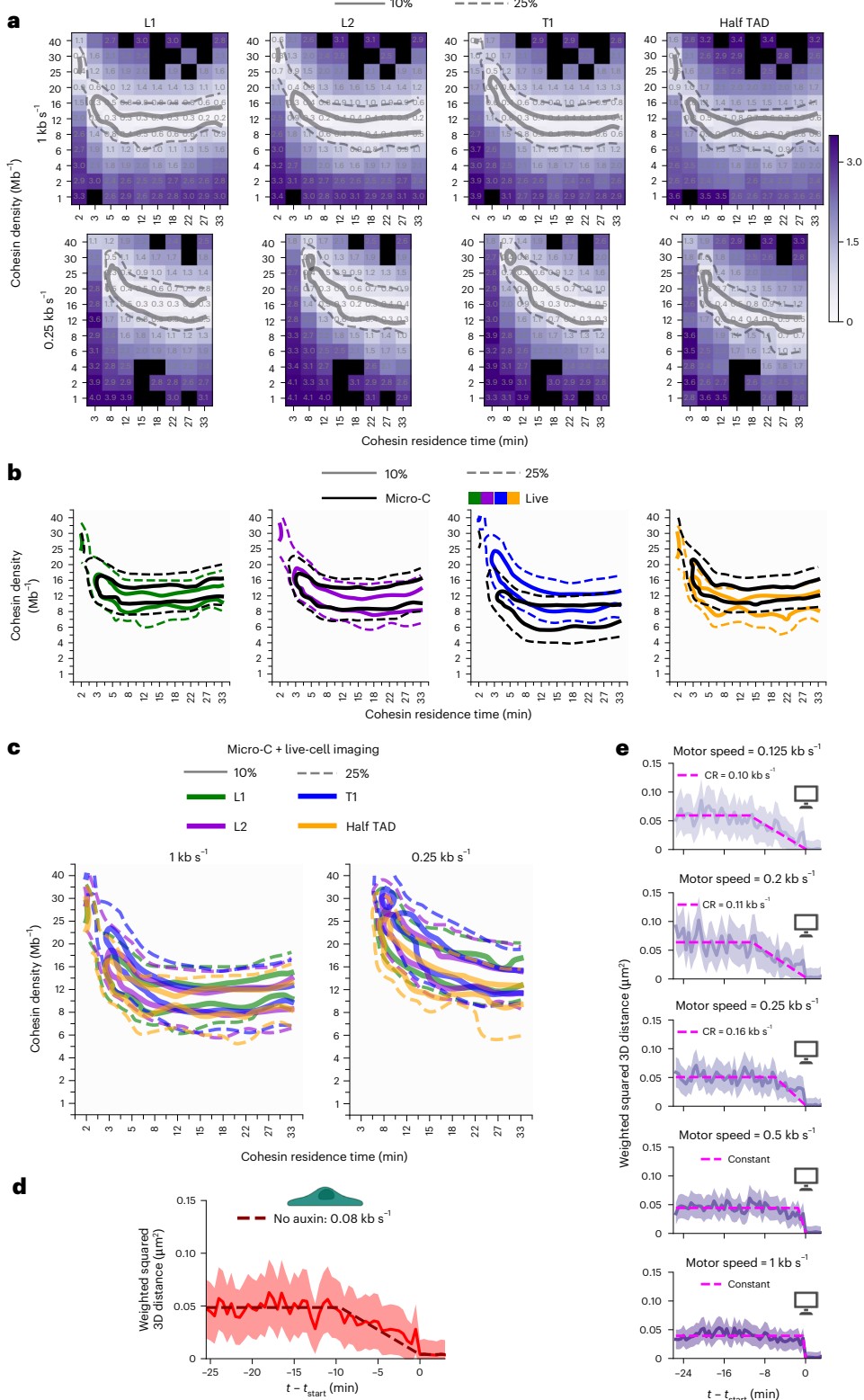

**Fig. 5 | TAD dynamics is consistent with uniform cohesin dynamics and is governed by CTCF binding. a**, Deviation of simulations from experiments, assuming a cohesin motor speed of 1 kb s⁻¹ (top) or 0.25 kb s⁻¹ (bottom). Black squares correspond to nonassessed parameter combinations. **b**, Deviation of simulations from experiments, separately considering Micro-C (black) or live-cell imaging (colors) data, and assuming a cohesin motor speed of 1 kb s⁻¹. **c**, Superposed contour plots for all genomic regions, based on Micro-C and live-cell imaging data taken together, for a motor speed of 1 kb s⁻¹ (left) or 0.25 kb s⁻¹ (right). In **a**–**c**, solid and dashed lines indicate the 10% and 25% best parameter sets, respectively. **d**,**e**, Average squared 3D anchor–anchor distance

time series weighted by localization precision, aligned on the starting times of proximal states ($t_{start}$), from experiments (**d**) and simulations (**e**) of the L1 TAD. The shaded area indicates the weighted s.e.m. Simulations (**e**) assume cohesin motor speeds ranging from 0.125 kb s⁻¹ (top) to 1 kb s⁻¹ (bottom), as indicated, and assume a cohesin density and residence time of 12 Mb⁻¹ and 22 min, respectively. Dashed lines show a fitted piecewise linear function used to estimate the CR. For simulations with motor speeds of 0.5 and 1 kb s⁻¹, the time series do not exhibit a linear decrease and CR cannot be estimated. Panels **d** and **e** were created in part with BioRender.com.

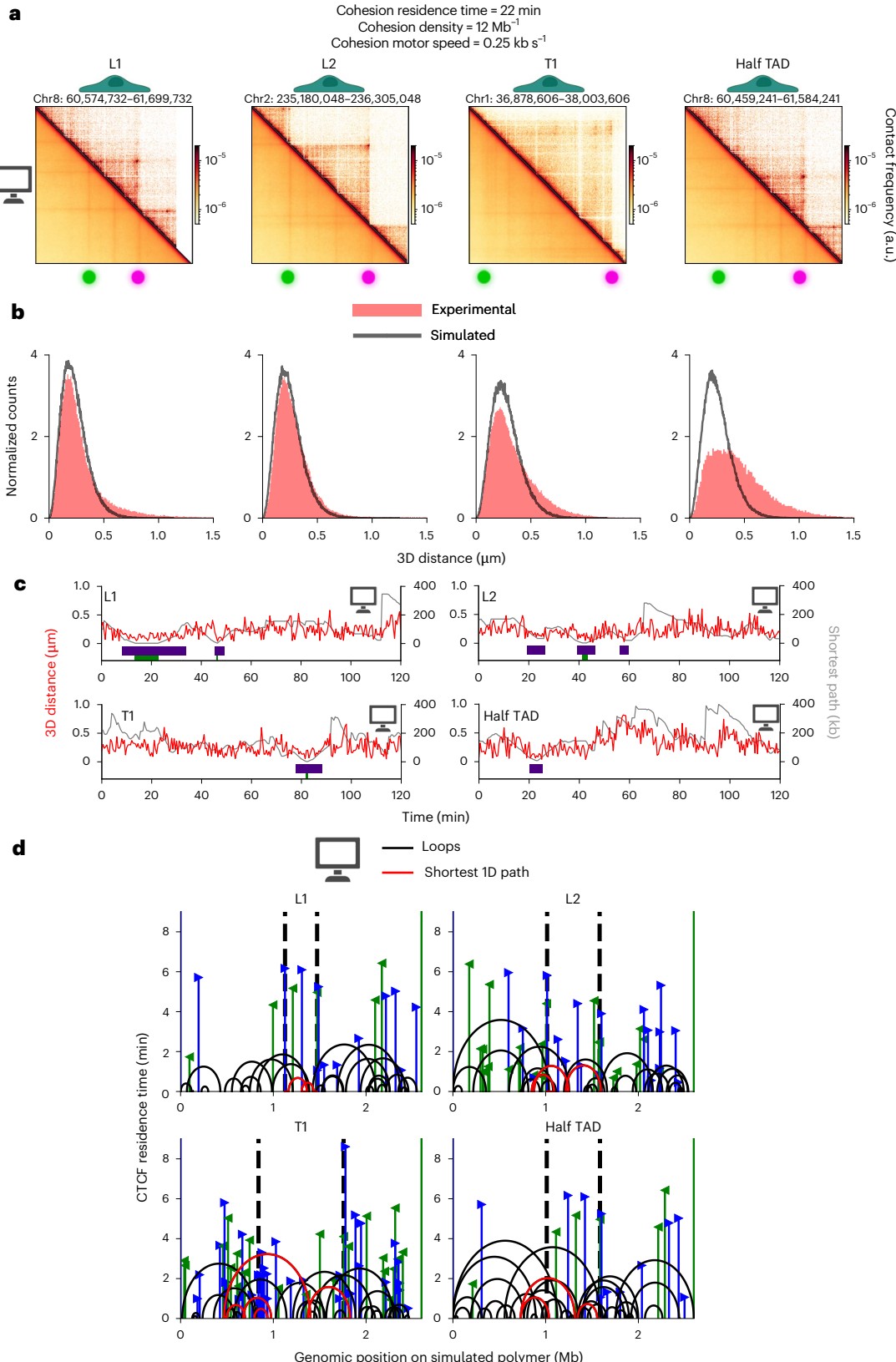

**Fig. 6 | TAD dynamics from polymer simulations consistent with experiments.**
**a,b**, Contact maps (**a**) and distance distributions (**b**) from experiments and simulations. Contact maps display a 1,125-kb region centered around TAD anchors (indicated by the green and magenta spots), at 2-kb resolution. **c**, Anchor–anchor 3D distance time series (red) from polymer simulations. The length of the shortest 1D path (unextruded DNA length) connecting TAD anchors is shown in gray. Green and purple rectangles indicate closed states and segmented proximal states, respectively. **d**, Snapshots of 1D representations of extruding loops. Red curves show the shortest 1D path between TAD anchors. Black vertical dotted lines indicate the location of TAD anchors. Blue and green lines with arrowheads indicate CTCF sites as in Fig. 4b. Simulations assume a cohesin density of 12 Mb$^{-1}$, a residence time of 22 min and a motor speed of 0.25 kb s$^{-1}$. Panels **a**, **c** and **d** were created in part with BioRender.com.

**Table 1 | Summary of findings on TAD dynamics of multiple regions in the human genome**

| | L1 | L2 | T1 | Half TAD | Estimated from |
|---|---|---|---|---|---|
| Genomic size | 345kb | 566kb | 918kb | 576kb | NA |
| Proximal fraction | 8–26% | 8–25% | 7–24% | 3–12% | Experiments |
| Proximal lifetime | 6–19min | 6–16min | 6–19min | 5–15min | Experiments |
| Proximal frequency | 0.3–2.7h⁻¹ | 0.3–2.7h⁻¹ | 0.3–2.6h⁻¹ | 0.2–1.4h⁻¹ | Experiments |
| Closing Rate | 0.07kbs⁻¹ | 0.07kbs⁻¹ | 0.16kbs⁻¹ | 0.09kbs⁻¹ | Experiments |
| Proximal fraction[a] | 24% | 21% | 16% | 9% | Experiments[a] |
| Closed fraction | 0–1% | 0–0% | 0–0% | 0–0% | Simulations |
| Extruding fraction[a] | 76% | 79% | 73% | 49% | Experiments[a] |
| Extruding fraction | 99–100% | 100–100% | 100–100% | 100–100% | Simulations |
| Open fraction[a] | 0% | 0% | 11% | 42% | Experiments[a] |
| Open fraction | 0–1% | 0–0% | 0–0% | 0–0% | Simulations |
| Cohesin density | 12–18Mb⁻¹ | 12–18Mb⁻¹ | 12–18Mb⁻¹ | 12–18Mb⁻¹ | Simulations |
| Cohesin residence time | 15–25min | 15–25min | 15–25min | 15–25min | Simulations |
| Cohesin loading rate | 0.5–1.2min⁻¹Mb⁻¹ | 0.5–1.2min⁻¹Mb⁻¹ | 0.5–1.2min⁻¹Mb⁻¹ | 0.5–1.2min⁻¹Mb⁻¹ | Simulations |
| Cohesin motor speed | <0.5kbs⁻¹ | <0.5kbs⁻¹ | <0.5kbs⁻¹ | <0.5kbs⁻¹ | Simulations |
| Number of loops connecting TAD anchors | 2.4–3.1 | 2.9–3.9 | 3.4–5.8 | 3.0–3.9 | Simulations |

Quantitative estimates obtained from a direct analysis of experimental data or from polymer simulations. For simulations, we used the 10% best parameter sets across all regions, at a motor speed of 0.25 kb s⁻¹. The reported ranges of cohesin dynamics parameters are 95% CIs. [a]Estimations from experiments using a three-state analytical model. NA, not applicable.

and separated by 112, 128 and 161 kb of unextruded DNA for L1, L2 and T1, respectively. Thus, despite a 2.7-fold size difference between these TADs (345–918 kb), their unextruded DNA length varied by only 1.4-fold, suggesting that loop extrusion tends to homogenize effective genomic distances (Extended Data Fig. 8h). Counterintuitively, DNA sequences outside TADs and up to 600 kb from the anchors could also contribute to the shortest 1D path (Fig. 6d and Supplementary Video 2). Thus, the dynamics of TADs can be influenced by CTCF and cohesin positioned far from TAD anchors, highlighting the importance of neighboring regions for cohesin-dependent chromatin interactions at a specific locus.

## Discussion

We used live-cell microscopy to track the motion of endogenous TAD anchors at multiple genomic regions in human cells, in the presence or absence of cohesin. We quantitatively characterized TAD dynamics and used polymer simulations to estimate the parameters of cohesin dynamics governing DNA loop extrusion.

First, our data show that TADs are highly dynamic structures and that anchor–anchor contacts are transient. We thereby extend findings from mESCs[26,27] to human cells, which have a ~tenfold longer G1 phase. For TADs of 345–918 kb, our analysis indicates that their anchors are in proximity for 7–26% of the time, but that direct anchor–anchor contacts are rare (closed state fractions of 0–1%; Table 1). This is consistent with previous results in mESCs where closed state fractions ranged from 2–3% for an endogenous 505 kb TAD[26] to 20–31% for a strong synthetic 150-kb TAD[27]. We further estimated proximal state durations of 6–19 min, again in agreement with lifetimes of 5–45 min in mESCs[26,27], despite the different species and analysis methods. While proximal states are transient, our data indicate that they are relatively frequent, because we estimated that TAD anchors come into proximity 0.3–2.7 times per hour, that is, 3–27 times during a 10 h-long G1 phase. Thus, our results establish the highly dynamic nature of TADs in human cells, suggesting that the dynamic process of loop extrusion itself is functionally more important than anchor–anchor interactions (Fig. 7).

Second, we found that TADs are predominantly in a partially extruded state, almost constantly undergo loop extrusion, and are rarely, if ever, in a fully open state (Table 1 and Fig. 7). This agrees with

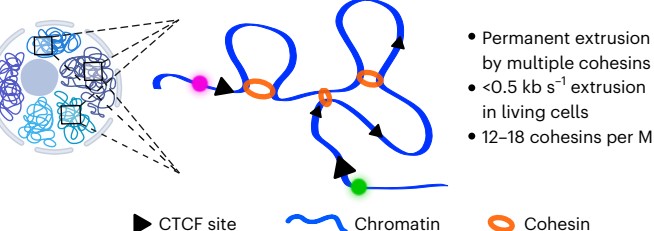

Uniform cohesin dynamics across genomic regions
TAD dynamics is controlled primarily by CTCF

- Permanent extrusion by multiple cohesins
- <0.5 kb s⁻¹ extrusion in living cells
- 12–18 cohesins per Mb

▶ CTCF site ～ Chromatin ⬭ Cohesin

**Fig. 7 | Model of TAD dynamics.** Cohesin dynamics is uniform across multiple genomic regions. The figure was created in part with BioRender.com.

a previous estimate of 92% extruding state in mESCs[26], and supports a model where TADs emerge from a collection of growing loops rather than from a single loop, with several cohesin complexes simultaneously bound within a single TAD[60].

Third, our study provides constraints on the speed of DNA loop extrusion in vivo. We determined CRs of 0.07–0.16 kb s⁻¹ and an upper bound for the cohesin motor speed of 0.5 kb s⁻¹, which agrees with a previous estimation of 0.125 kb s⁻¹ in mESCs[26], but is slower than the 0.5–1 kb s⁻¹ measured in vitro with purified proteins[10,11]. This deceleration in vivo possibly results from the crowded chromatin environment (for example, nucleosomes acting as roadblocks[61]) or from cellular factors directly reducing cohesin motor speed.

Fourth, polymer simulations indicate a cohesin density of 12–18 Mb⁻¹ (Table 1), in line with previous estimations of 4–8 Mb⁻¹ (refs. 9,26,54,62), 9–15 Mb⁻¹ (ref. 59) or 8–32 Mb⁻¹ (ref. 27). This corresponds to 2.8–4.1 cohesin complexes simultaneously bound within a 'median' loop of 230 kb. Our estimates translate into ~74,000–112,000 chromatin-bound cohesin complexes extruding the 6.2-Gb-long human genome of the diploid HCT116 cells in G1, aligning with absolute quantifications of cohesin complexes (43,000–114,000 for a virtually 'diploid' HeLa cell)[55]. The estimated densities further confirm that multiple cohesin complexes simultaneously extrude loops within single TADs.

Fifth, our analyses are consistent with cohesin residence times ranging from 15 to 25 min (Table 1), in accordance with previous estimations of 3–25 min[27,35,53,55–58]. Combined with the estimated cohesin density and an assumed motor speed of 0.25 kb s$^{-1}$, and taking into account stalling at CTCF sites, this predicts effective genomic processivities of 190–340 kb, intersecting previous estimations of 120–240 kb[9,26]. The estimated cohesin loading rates fell between 0.5 and 1.2 Mb$^{-1}$ min$^{-1}$ (Extended Data Fig. 8i), narrower than previous estimates of 0.06–1.2 Mb$^{-1}$ min$^{-1}$ in mESCs[27]. Thus, within a median 230-kb loop, a total of 69–166 cohesin complexes extrude loops during a 10-h G1 phase. We note that these parameters agree with experimental observations of cohesin binding kinetics after mitotic exit[6,63] (Extended Data Fig. 9k).

Sixth, our study analyzed multiple genomic regions featuring different CTCF site distributions, histone marks, Hi-C patterns and dynamics with the same methods, allowing us to compare extrusion dynamics and their determinants across chromatin regions. Strikingly, we found that their strong differences can be largely explained by a single combination of cohesin density, residence time and motor speed. Polymer simulations indicate that differences between regions mainly result from the different locations and strengths of CTCF sites rather than from local variations in cohesin dynamics. Thus, our study suggests uniform dynamics of cohesin across the genome, rather than local regulation (Fig. 7). Moreover, the consistency of our estimates with previous studies in mESCs raises the possibility that cohesin dynamics is conserved among mammalian species[26,27]. At the same time, these results suggest the crucial and potentially exclusive regulatory role of CTCF in controlling TAD dynamics. Because the sole knowledge of CTCF site locations and affinities appears sufficient to quantitatively predict TAD dynamics from polymer physics, our study enables predictive models of loop extrusion across the genome, and of nuclear functions such as enhancer–promoter contacts regulating gene transcription[19,20,64].

Nonetheless, we acknowledge several limitations. First, our conclusions are based on only three genomic regions, with particularly strong TADs (Extended Data Fig. 1d,e). Hence, some quantitative estimates (for example, proximal state fraction, frequency and lifetime) may be larger than for an 'average' TAD. Second, these regions were all located in the A compartment. Although a previous study suggested that differences in Hi-C patterns between A and B compartments result from differences in CTCF binding rather than from differences in cohesin dynamics[65], it remains important to assess cohesin dynamics in B compartments. Third, our analyses are contingent on the assumption that loop extrusion occurs in vivo and that 2 h of RAD21 depletion affect anchor dynamics through its loop extrusion activity specifically, as supported by multiple lines of evidence[6,7,9–14,26,27]. Fourth, our analyses provide only an upper bound on the cohesin motor speed. Narrower estimates would require a more systematic exploration of the simulation parameter space. Fifth, although we experimentally minimized interference from biological processes such as transcription, we cannot rule out that other factors may be influencing chromatin interactions (for example, compartmentalization or other extruding complexes such as SMC5/SMC6 (refs. 12,50,66,67)), which our simulations ignored. This could potentially lead to discrepancies between simulations and experiments. Finally, our simulations ignored the reported increase in cohesin residence time upon interaction with CTCF[16,26]. Accounting for such interactions may lead to decreased estimates of cohesin density and motor speed.

Together, our results describe the highly dynamic nature of cohesin-induced chromatin interactions in the human genome. They support a model in which cohesin complexes almost constantly extrude loops and form transient, rather than stable, contacts between TAD anchors. The proposed uniformity of cohesin dynamics and the crucial regulatory role of CTCF will empower predictive models of TAD dynamics and extrusion-dependent genomic functions.

## Online content

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

## Methods

This research complies with all relevant ethical regulations as required by Institut Pasteur and the CNRS.

### Experimental methods

**Cell line culture, generation and treatment conditions.** *Cell culture.* HCT116 cells (ATCC, CCL-247) were cultured in McCoy's medium supplemented with GlutaMAX (Thermo Fisher Scientific, 36600021), 10% FBS (Sigma-Aldrich, F7524) and penicillin–streptomycin (50 U ml⁻¹ and 50 µg ml⁻¹, respectively; Thermo Fisher Scientific, 15140122). Cells were grown at 37 °C in a humidified incubator with 5% $CO_2$ and split every 2–3 days. Cells were tested monthly for the presence of *Mycoplasma* spp., *Ureaplasma* spp. and *Acholeplasma laidlawii* by qPCR[68]. None of the proteins directly involved in cohesin-dependent loop extrusion (RAD21, NIPBL, WAPL, CTCF, STAG1 or STAG2) carries mutations in HCT116 cells[69].

*Cell line generation and genome editing.* CRISPR-based genome editing was performed using a Cas9 nickase to reduce off-target effects. We cotransfected a repair plasmid (0.6 µg), a nickase Cas9 expressing plasmid (0.4 µg; Addgene, 42335 (ref. 70)) and a pair of single guide RNAs (sgRNAs; 0.5 µg each) using JetPrime (Polyplus, 101000001) according to the manufacturer's protocol. The pairs of sgRNAs were designed using ChopChop (v3 (ref. 71)) with the parameters: 'hg19', 'nickase' and 'knock-in'. For repeat array insertion, homology arms were PCR-amplified and cloned by a four-fragment Gibson assembly (NEB, E2621S) within the loxP-Blasticidin-HSVTK-loxP-TetOx96 and CuOx150-FRT-Neomycin-FRT plasmids[72] (Supplementary Tables 3 and 4). We used homology arms of 581–1,436 bp.

A total of 300,000 cells were seeded in a six-well plate and transfected 24 h later. Less than 24 h after transfection, cells were detached from the well and split into four different 10-cm plates for antibiotic selection. Each 10-cm plate contained a different dilution of the initial six-well plate (from 1/40 to 4/5). Cells were kept under antibiotic selection until unique colonies were seen (about 2 weeks). Single colonies were then picked using cloning disks (Merck, Z374431) and put into 24-well plates. Once sufficiently grown, each clone was split in half. One half was used for clone expansion and the other half was seeded on a glass slide for image-based screening. Three days after seeding, Halo tag was labeled, if needed, with 100-nM JFX646 (ref. 41; gift from Luke Lavis Lab) in culture medium by incubating the cells for 15 min at 37 °C. Cells were then fixed with 4% formaldehyde (Thermo Fisher Scientific, 28908), diluted in PBS (Sigma-Aldrich, D8537), for 20 min, washed thrice in PBS, and slides were mounted in Vectashield Antifade Medium with DAPI (Vector, H-1200-10). Clones were imaged with a widefield microscope (Zeiss Axioimager, ×63 1.4-NA Plan-Apochromat objective with an LED Xcite 120LED as illumination source and an ORCA-Flash4 LT Hamamatsu camera with 2048 × 2048 pixels and a pixel size of 6.5 µm). Clones that displayed one or two fluorescent spots per nucleus were further split into a six-well plate and genomic DNA (gDNA) was extracted (Lucigen, QE0905T). PCR (Promega, M7801) genotyping was performed by amplifying 5' and 3' junctions (Supplementary Table 3)[72]. The unmodified wild-type (WT) band was amplified to assess the zygosity of the insertion. High-quality gDNA of clones verified by microscopy and PCR was purified (Promega, A1120) and each genotyping PCR fragment was sequenced. Finally, we checked the integrity of the TetOx96 array by PCR amplifying and sequencing the whole array.

To enable auxin-dependent RAD21 degradation, we first homozygously inserted the RAD21–mAID–SNAP–IRES–Hygromycin fusion at the endogenous *RAD21* gene locus, using the WT Cas9 (0.4-µg Cas9; Addgene, 41815 (ref. 73), and 0.8 µg each for the repair DNA and sgRNA plasmids), and 100 µg ml⁻¹ Hygromycin B (Thermo Fisher Scientific, 10687010) for selection. Next, we inserted the AtAFB2-weakNLS-IRES-Puromycin at the AAVS1 locus, using a WT Cas9 (Addgene, 72833 (ref. 74)) and by selecting cells with 1 µg ml⁻¹ puromycin (Invivogen, ant-pr-1). The AtAFB2 ubiquitin ligase was reported to

minimize basal degradation of the degron-tagged protein, as compared to the more common OsTIR1 degron[48]. The weak NLS allowed depletion of both nuclear and newly synthesized cytoplasmic RAD21 (ref. 48). Cell lines were regularly cultured with 1 µg ml⁻¹ puromycin for 1 week to ensure high expression of the AtAFB2 degron[48] before switching to regular culture medium.

We then expressed the fluorescence reporters required to visualize the repeat arrays. Using an optimized piggyBac transposase (we transfected 0.3 µg of transposase plasmid to insert a single copy of each reporter[75], among 2 µg of total DNA), we inserted CymR-NLS-2xHalo (Addgene, 119907 (ref. 76)) to visualize CuO repeats and TetR-GFP11x16-GB1-NLS to visualize TetO repeats in the cells. We infected the cells with lentiviruses carrying the GB1-GFP1_10-GB1-NLS fragment (bearing the A206K mutation to prevent dimerization[77]) to reconstitute splitGFP. Cells were sorted thrice, once a week, to retain only low-expressing levels of the reporter proteins. Then, we re-infected the cells with the GFP1_10 lentiviruses to increase the ratio of GFP1_10 over TetR-GFP11x16-GB1-NLS and optimize the signal from the multimerized GFP11 fragments[78]. These cells served as the parental cell line for repeat array insertion.

We then sequentially inserted the CuOx150-FRT-Neomycin-FRT and loxP-Blasticidin-HSVTK-loxP-TetOx96 arrays at each anchor of the TADs (Fig. 1a). The insertion of these TetOx96 and CuOx150 repair plasmids was selected using 6 µg ml⁻¹ blasticidin (Invivogen, ant-bl-1) and 400 µg ml⁻¹ G418 (Invivogen, ant-gn-5), respectively. The expression of antibiotic resistance genes was designed to direct transcription outwards of the TAD interior to avoid interference with cohesin complexes extruding from inside the TAD (Fig. 1a). Finally, once TetOx96 and CuOx150 array insertions on the same allele were verified by microscopy, we removed the neomycin and blasticidin antibiotic cassettes using Cre and FLP recombinases. Because transcription is known to alter chromatin dynamics[79,80] and RNA polymerase II can slow down, or push extruding cohesin, acting as a mobile barrier for cohesin[45], the absence of strong transcription from antibiotic cassettes allows us to measure cohesin-mediated motion of TAD anchors as purely as possible. A total of 1 µg of Cre (Addgene, 123133 (ref. 81)) and 1 µg of FLP (Addgene, 13793 (ref. 82)) expressing plasmids were transfected. Clones that lost the antibiotic cassettes were selected for loss of the herpes simplex virus thymidine kinase (HSVTK) gene, making cells sensitive to 8 µg ml⁻¹ ganciclovir (Invivogen, sud-gcv). Thus, 5', 3' junctions and the WT allele were sequenced as previously described. These clones were subsequently used for live-cell imaging.

For all cell lines, we obtained heterozygous insertions of the repeat arrays, allowing us to track one pair of green and far-red spots in the cells. For the L2 cell line, the CuOx150 array was inserted homozygously, resulting in two distinct far-red spots, while the TetOx96 cassette only integrated in a single allele. Upon sequencing of the L2 clone, we found a plasmid fragment of 396 bp integrated within the inserted exogenous sequence, at the 3' end of the CuO repeats. This sequence contained a lac operon and promoter, Gateway attB2 sites and a catabolite activator protein binding site. In the T1 cell line, a small fraction of cells retained the antibiotic resistance genes after Cre and FLP recombinations, as assessed by PCR amplification of the nonrecombined alleles.

TetOx96 and CuOx150 arrays were inserted at the following locations in the human genome (hg19): L1 (chr8: 60,964,180 and chr8: 61,310,370), L2 (chr2: 235,458,700 and chr2: 236,026,413), T1 (chr1: 36,980,442 and chr1: 37,901,640), half TAD (chr8: 60,733,873 and chr8: 61,310,370) and adjacent (chr1: 37,900,310 and chr1: 37,901,640). The distance between repeat arrays in the Adjacent locus was chosen to match reporter–anchor separations in other cell lines (Supplementary Table 2).

**Auxin-mediated RAD21 degradation and western blotting.** To deplete RAD21 fused to the mini auxin inducible degron (mAID)[74], we added auxin (Sigma-Aldrich, I5148) to a final concentration of

500 μM (from a 500X stock solution diluted in PBS) in fresh culture or imaging medium.

RAD21 depletion kinetics was measured by both live-cell imaging and western blotting (Extended Data Fig. 3a–f). For western blotting, 300,000 cells were seeded and grown for 48 h in six-well plates and incubated with auxin for the indicated times. Cells were washed thrice with cold PBS and lysed with 200 μl of HNTG buffer (HEPES pH 7.4 50 mM, NaCl 150 mM, Glycerol 10%, Triton X-100 1%) with 1× protease inhibitor (Roche, 5056489001). After cell collection, lysates were rotated for 30 min at 4 °C, sonicated and rotated for 30 min at 4 °C before centrifugation, and supernatants were stored at −80 °C until loading. Protein levels were quantified using the Pierce BCA protein assay (Thermo Fisher Scientific, 23225). Samples were boiled for 5 min at 100 °C in 1× Laemmli buffer and 10 μg of protein extract were loaded into a 10% Mini-PROTEAN TGX gel (Bio-Rad, 4561036). Samples were run for 90 min at 110 V in Tris–glycine 1× (Euromedex, EU0550) and 0.5% sodium dodecyl sulfate (Euromedex, EU0660s) buffer, and proteins were transferred to a nitrocellulose membrane (Pall BioTrace, 66485) for 75 min at 100 V in Tris–glycine 1× and 20% ethanol (VWR, 83804.360) buffer. Membranes were blocked with 5% milk in 1× Tris-buffered saline (TBS; Tris 20 mM, NaCl 150 mM) for 1 h at room temperature. Immunostaining was performed overnight at 4 °C with primary antibodies (Rad21 1:1,500 (Abcam, ab154769), GAPDH 1:50,000 (Abcam, ab8245)) diluted in 5% milk in 1× TBS–Tween (0.02% Tween; Thermo Fisher Scientific, 11368311). The membrane was washed thrice for 5 min with 1× TBS–Tween at room temperature and incubated for 1 h with secondary antibodies diluted in 1× TBS–Tween at room temperature (antirabbit IR800 1:10,000 (Advansta, R-05060), antimouse IR800 1:10,000 (Advansta, R-05061)). The membrane was washed thrice in 1× TBS–Tween for 5 min and once in 1× TBS for 10 min. Before imaging, membranes were soaked in 70% ethanol and air-dried in a dark chamber. We measured fluorescence intensity with the Chemidoc MP Imaging system (Bio-Rad).

For live-cell quantification of auxin-mediated RAD21 degradation kinetics (Extended Data Fig. 3a,c), we used Rad21–mAID–SNAP cells containing the TetR-GFP11x16-NLS and CymR-NLS-2xHalo constructs (parental cell line). A total of 300,000 cells were seeded and cultured in glass-bottom imaging dishes (Ibidi, 81158) for 2 days. Before imaging, the SNAP JF646 dye (gift from Luke Lavis Lab) was added to fresh medium at a final concentration of 100 nM and cells were incubated for 90 min at 37 °C. Cells were washed thrice with warm PBS and imaging medium (DMEMgfp (Evrogen, MC102) supplemented with 10% FBS or FluoroBrite DMEM (Thermo Fisher Scientific, A1896701) supplemented with 1× GlutaMAX (Thermo Fisher Scientific, 35050061) and 10% FBS) was added to the cells. Time-lapse images were acquired in a bespoke microscope equipped with a 488-nm TA Deepstar Diode Laser (Omicron-Laserage Laserprodukte GmbH) and a 647-nm OBIS LX laser (Coherent) for excitation. The microscope was equipped with an Olympus UPLAPO ×60, 1.42-NA objective, and additional optics leading to a 102-nm pixel size in the final image. Green and far-red fluorescence emission was split at 580 nm by an FF580-FDi02-t3-25×36 dichroic mirror (Semrock) and filtered with 525/50 nm and 685/40 nm fluorescence filters (Alluxa), respectively. Two-color images were captured by two separate sCMOS cameras—a Zyla 4.2 plus for the green channel and a Zyla 4.2 for the far-red (Oxford Instruments) channel. The sample environment ($CO_2$ concentration, temperature and humidity) was controlled with a top-stage chamber (Okolab SRL). All devices of the microscope were controlled using Python-microscope[83] and using cockpit as graphical interface[84]. We took 31 z-slices separated by 0.4 μm each and z-stacks were taken every 15 min for 45 min at multiple positions. Then, cells were removed from the microscope stage and auxin was added to the medium. After placing the cells back in the microscope chamber, we selected new positions on the slide and imaged cells at the same frequency for 4 h.

To measure RAD21 levels in live-cell images, we used a custom Python script on maximum-intensity projection images. Nuclei were segmented using Labkit (v0.4 (ref. [85])) and tracked using TrackMate (v7 (ref. [86])), based on the green channel containing the TetR-splitGFPx16-NLS signal. We removed all dividing cells and cells at the edge of the image from the analysis. Using these segmentation masks, we measured the median fluorescence intensity in the RAD21 channel. Background intensity was subtracted, and fluorescence intensities were normalized to the first time point of imaging without auxin.

**Cell cycle analysis by fluorescence-activated cell sorting (FACS).** To assess the fraction of cells in G1 to early S phase and compare it with our image-based assessment of replicated spots, we used FACS with propidium iodide (PI; Sigma-Aldrich, P4864) staining of fixed cells (Extended Data Fig. 2e–g). A total of 300,000 cells were seeded and grown in six-well plates and cells were collected 48 h later. Cells were trypsinized, washed in PBS and resuspended in 500-μl PBS. Then, 5.5 ml of ice-cold 70% ethanol was added for fixation. Cells were kept at 4 °C in 70% ethanol for at least 12 h until staining. For staining, cells were washed twice in PBS and incubated for 5 min at room temperature with 50 μl of a 50 μg ml⁻¹ RNAse A solution (Promega, A7973). Finally, 400 μl of 50 μg ml⁻¹ PI solution was added and cells were incubated for 15 min at room temperature before FACS sorting. We used a Miltenyi MACSQuant Analyzer 10 Flow Cytometer with the 488-nm laser and a 692/75-nm bandpass filter. We gated cells based on forward scatter relative to side scatter, and single cells based on PI height relative to PI area. We did not consider polyploid cells (at least triploid) in the analysis (they represented <2.5% of cells). Finally, we analyzed at least 12,000 cells within the final gate of interest (Supplementary Fig. 1).

From the distribution of propidium intensity, the percentage of cells in each cell cycle phase was computed using the Dean–Jett–Fox model[87] without constraints in FlowJo (v10) software (Extended Data Fig. 2g).

**Capture Micro-C.** Micro-C libraries were generated with the Dovetail Micro-C kit (21006) protocol (v1.0) with tiling capture of genomic loci (Agilent) with minor modifications. Thus, 1.2 to $2 \times 10^6$ cells from each cell line containing the repeat arrays were washed in PBS, the supernatant was carefully removed and cell pellets were stored at −80 °C for at least a day. Prefreezing is required to get an optimal micrococcal nuclease (MNase) digestion profile. Cell pellets were then thawed and processed as prescribed by the Dovetail Micro-C protocol. The first crosslinking step was performed with 3 mM DSG (Thermo Fisher Scientific, A35392) in PBS for 10 min at room temperature with rotation, and formaldehyde was added to a final concentration of 1% for another 10 min. The MNase digest was carried out according to kit instructions and the digestion profile was routinely verified on a Bioanalyzer 2100 instrument (Agilent). For the end repair and adaptor ligation steps, the NEBNext Ultra II DNA Library Prep Kit for Illumina (NEB, E7645) was used. Afterwards, DNA was purified through solid-phase reversible immobilization (SPRI) beads (Beckman, A63880) as described in the Dovetail kit. Biotin pulldown and library amplification were performed using the Dovetail Micro-C Kit reagents. After verifying that the libraries had the correct concentrations and size distributions, we proceeded with the Agilent SureSelectXT HS2 kit (G9987A; design S3442002). Capture probe design was performed by Agilent. The coordinates of capture probes were as follows: chr8: 60,458,500–61,587,500 for L1 and half TAD regions; chr2: 235,182,500–236,297,500 for L2 and chr1: 36,700,000–38,175,000 for T1. We followed the manufacturer's protocol for prepooling eight sequencing libraries. Finally, we checked concentrations and size distributions before sending capture-sequencing libraries for sequencing. We sequenced with BGI Illumina 100-bp paired-end sequencing (PE100).

The data presented in this manuscript were pooled from two biological replicates for each cell line, except for the half TAD and untagged cell lines (Extended Data Fig. 1b,c), where a single experiment was performed. Raw sequencing data from BGI were checked by FastQC[88] (FastQC, v0.12.1). None of the replicates showed any irregularities. All raw fastq reads were trimmed to 50 bp using TrimGalore (Cutadapt, v0.6.10 (ref. [89])). Next, valid Micro-C contacts were obtained with the HiC-Pro pipeline[90] (HiC-Pro_v3.1.0). HiC-Pro uses Bowtie2 (v2.4.4 (ref. [91])) to map the reads to the human genome. All valid Micro-C contact pair files obtained from HiC-Pro were filtered for the corresponding region of interest. Contact matrices were generated using the Cooler package (v0.9.1 (refs. [92],[93])).

**Live-cell imaging of TAD anchors.** For live-cell microscopy of TAD anchors, cells were plated on a 35-mm glass-bottom imaging dish (Fluorodish, FD35-100). For 48–72 h after seeding, the medium was replaced with fresh medium containing 100 nM of JFX646 Halo dye and the cells were incubated for 15 min at 37 °C. Cells were washed twice with PBS and the medium was replaced with live-cell imaging medium (DMEMgfp supplemented with 10% FBS or Fluorobrite DMEM supplemented with 1× GlutaMAX and 10% FBS). For RAD21 degradation, cells were treated with 500-μM auxin. Auxin was maintained in the Halo labeling medium and in the imaging medium during acquisition. Cells were imaged for 2 h, starting 2 h after the addition of auxin. Before imaging, cells were allowed to equilibrate in the microscopy incubation chamber for at least 15 min at 37 °C and 5% $CO_2$.

Time-lapse image acquisition was performed with an inverted microscope (Nikon) coupled to the Dragonfly spinning disk (Andor) using a 100X Plan Apo 1.45-NA oil immersion objective. Excitation sources were 488-nm (150 mW) and 637-nm (140 mW) lasers. Exposure time was set to 85 ms for both channels, with 1% laser power in the far-red channel and 5–8% in the GFP channel, depending on the imaged cell line. z-stacks of 29 optical slices separated by 0.29 μm each were acquired every 30 s using the perfect focus system and five different stage positions were imaged for each 2-h acquisition. The two channels were acquired simultaneously on two distinct electron-multiplying charge-coupled device iXon888 cameras (1024 × 1024 pixels, effective pixel size = 0.121 μm). For the high-frequency dataset (Extended Data Fig. 6f,i), images were acquired every 9 s for a total duration of 36 min (instead of every 30 s for 2 h in the original dataset). For RAD21-depleted cells imaged at high frequency, we acquired images between 2 and 4 h after auxin treatment.

### Computational methods
All computational methods, including genomic analysis, image and data analysis, and polymer model and simulations, are detailed in the Supplementary information.

### Statistics and reproducibility
No statistical method was used to predetermine sample size. No randomization was performed as the study did not require sample allocation into different groups. Blinding was not possible for data collection, as data acquisition required identification of the samples for further processing. No data were excluded from the analysis, except during quality filtering of distance time series, as detailed in the Supplementary Information—'Quality filtering of time series'. Data analysis was not performed blindly to the experimental conditions, except for the Capture Micro-C analysis. Live-cell imaging experiments were performed in two to six biological replicates and all replicates showed consistent results. Unless indicated otherwise, we used bootstrapping to estimate the s.d. of our quantifications. To generate bootstrap samples, we randomly selected individual time series with replacements from the full dataset (creating bootstrap sets of the same size as the original dataset). Normality and homoscedasticity were tested before running any statistical test. The statistical test, sampling size and type

of error bars are indicated in the legend of Figs. 1–3, Fig. 5, Table 1 and Extended Data Figs. 2–7 and Extended Data Figs. 9,10.

### Reporting summary
Further information on research design is available in the Nature Portfolio Reporting Summary linked to this article.

## Data availability
Capture Micro-C data have been uploaded to the Gene Expression Omnibus (GEO) under accession GSE273257. This paper analyzed existing, publicly available Hi-C, ChIP–seq and PRO-Seq data from GEO under accession GSE104334. Raw and quality-filtered distance time series are available at Zenodo https://doi.org/10.5281/zenodo.16949930 (ref. [94]). All plasmids and cell lines generated in this study are available upon request to corresponding authors. Source data are provided with this paper.

## Code availability
The code (v0 (ref. [95])) used to generate polymer simulations, process live-cell microscopy images, quantify the distance between TAD anchors and process the Capture Micro-C data is available at https://github.com/imodpasteur/Sabate_et_al_TAD_Anchors and at Zenodo https://doi.org/10.5281/zenodo.16949930 (ref. [94]).

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

## Acknowledgements

The authors would like to thank G. Cavalli and A. Coulon for useful discussions. The authors acknowledge and thank members of the MRI facility, part of the national infrastructure France-BioImaging, supported by the French National Research Agency (ANR-10-INBS-04; Investments for the Future). The authors acknowledge the help of the HPC Core Facility of the Institut Pasteur for the use of computing resources. The authors thank X. Pichon (Institut de Génétique Moléculaire de Montpellier, IGMM UMR535 Montpellier, France) for the initial cloning of the splitGFP array and the Luke D. Lavis lab (Janelia Research Campus, Howard Hughes Medical Institute, Ashburn, VA, USA) for sharing the JFX646 dye. T.S. was supported by 'Contrat doctoral spécifique aux Normaliens' and 'Fondation ARC pour la Recherche sur le Cancer' (ARCDOC; 42021120004333). The authors thank La Ligue Nationale Contre le Cancer for financial support to E.B. ('équipe labellisée'). This work was supported by a government grant managed by the Agence Nationale de la Recherche, under the France 2030 program, with the reference ANR-24-EXCI-0002. The authors also acknowledge the Investissement d'Avenir (ANR-16-CONV-0005) for funding computing resources used in this work. Figures 4–7 were created in part with BioRender.com.

## Author contributions

T.S. designed the project, performed cell line construction and experiments, polymer simulations and the analysis, and wrote the paper with the input from B.L., M.S., J.-Y.T., C.Z. and E.B. M.-C.R. participated in cell line construction and validation. B.L. developed parts of the image-processing and data-analysis pipelines. M.S. performed and generated Capture Micro-C maps. J.-Y.T. contributed to image analysis tools. C.Z. and E.B. supervised the project.

## Competing interests

All authors declare no competing interests.

## Additional information

**Extended data** is available for this paper at https://doi.org/10.1038/s41588-025-02406-9.

**Correspondence and requests for materials** should be addressed to Thomas Sabaté, Edouard Bertrand or Christophe Zimmer.

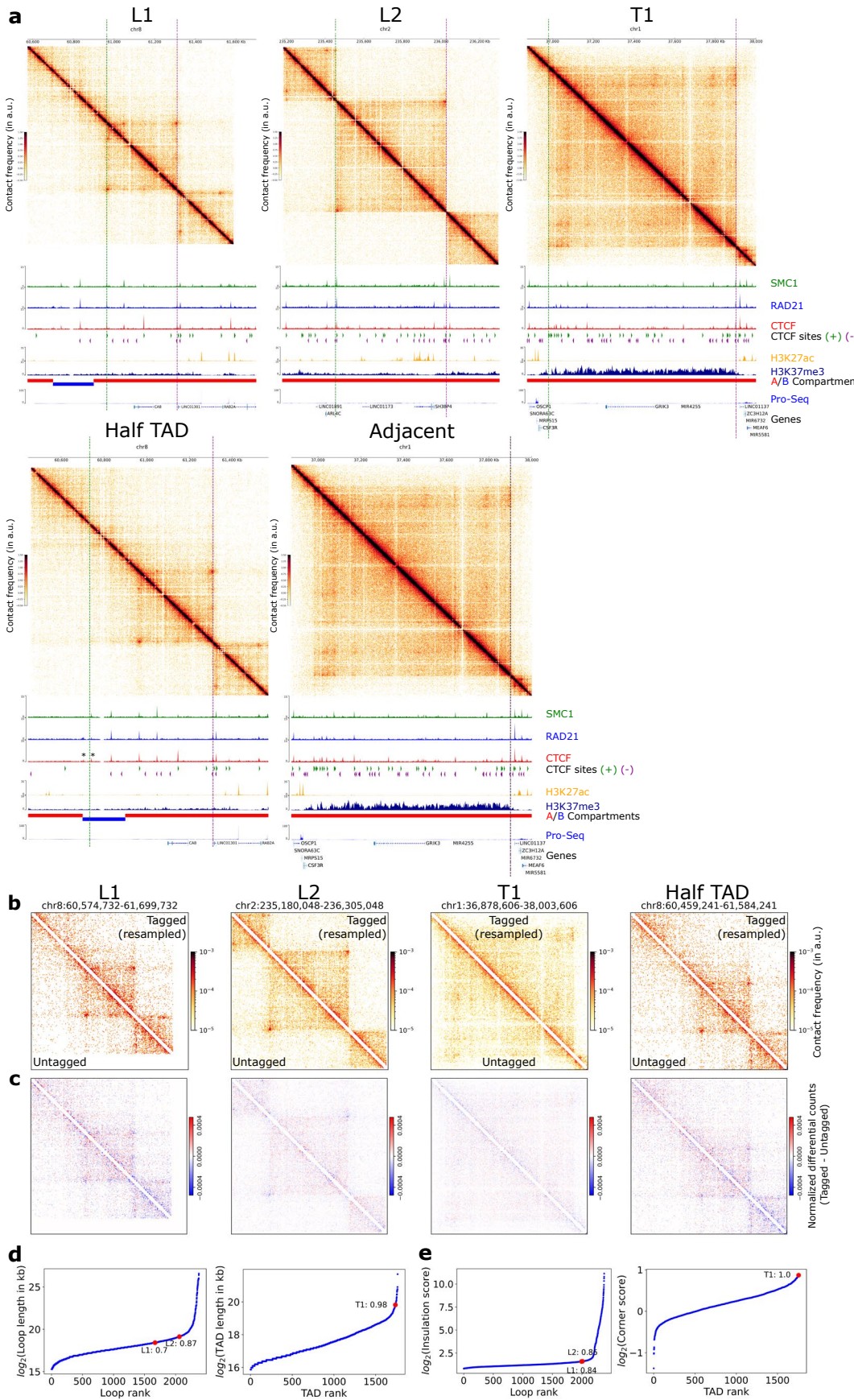

**Extended Data Fig. 1 | See next page for caption.**

**Extended Data Fig. 1 | Capture Micro-C maps and genomic context of chromatin regions. a**, Capture Micro-C maps show a 1,125 kb region centered on TAD anchors at 2 kb resolution. Below are shown ChIP–seq and PRO-Seq profiles (data from ref. 6), A/B compartments (computed from Hi-C maps in ref. 6), genes and orientation of CCCTC-binding factor (CTCF) sites. Green and magenta dotted lines indicate the genomic location of TetO and CuO repeat array insertion, respectively. The Capture Micro-C map generated from the T1 cell line was used to illustrate the adjacent locus. An asterisk indicates ChIP–seq peaks overlapping non-significant CTCF sites associated with high q-values (>0.34) in the half TAD control. These two CTCF sites are in divergent orientation as compared to the 3′ TAD anchor and thus not expected to form a loop. **b**, Capture Micro-C maps before (untagged) and after (tagged) insertion of repeat arrays.

To facilitate visual comparison, tagged maps were subsampled to match the total number of counts in the untagged map. **c**, Differential maps of normalized contact counts. Bins within 20 kb around the diagonal were excluded. Maps are shown at 5 kb resolution in **b** and **c**. **d**, Ranked genomic lengths of all loops (left) and TADs (right) in the genome. **e**, Ranked insulation scores (defined by the ratio of observed to expected contacts bottom left from HICCUPS; left) and corner scores (as computed by arrowhead from Juicer; right). In **d** and **e**, red dots correspond to the chosen TADs and loops, and their respective percentile is indicated. Only cohesin- and CTCF-dependent domains were considered, that is, loops and TADs exhibiting at least one ChIP–seq peak of SMC1, RAD21 and CTCF at both domain anchors.

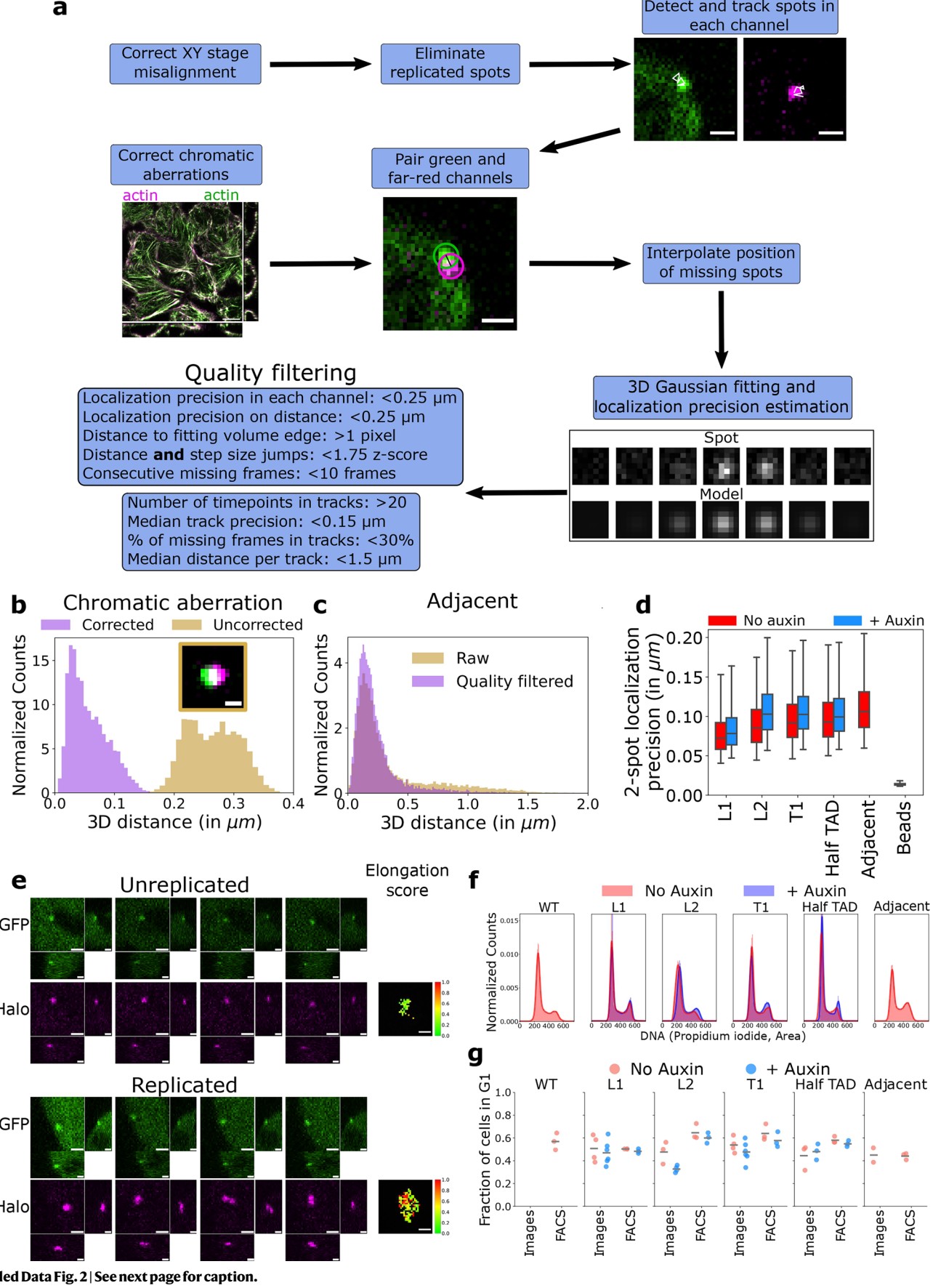

**Extended Data Fig. 2 | See next page for caption.**

**Extended Data Fig. 2 | Image analysis pipeline. a**, Overview of the image analysis pipeline. The actin image is shown before correction of chromatic aberrations, together with orthogonal projections on the sides. The 7 images labeled 'spot' show different z-planes of the raw fluorescent spot and the images labeled 'model' show the corresponding fitted Gaussian model. Scale bars = 10 µm for actin, 1 µm for fluorescent spot images. **b**, Histogram of 3D distances between green and far-red channels of fluorescent beads before and after correction of chromatic aberrations using dual actin labeling to estimate chromatic shifts. The inset shows an image of a fluorescent bead before correction of chromatic aberration. Scale bar = 0.5 µm. **c**, 3D anchor-anchor distance distribution before and after quality filtering for the adjacent locus. **d**, Localization precision of distance measurements estimated from Cramér-Rao bounds[46]. Horizontal bars indicate median, lower and upper quartiles. Whiskers extend from 2.5 to 97.5 percentiles for each of the following conditions: no auxin L1: n = 74,084 distances over 4 replicates, L2: n = 93,431 distances over 3 replicates, T1: n = 78,268 distances over 4 replicates, half TAD: n = 36,281 distances over 3 replicates, adjacent: n = 12,269 distances over 2 replicates; +auxin L1: n = 65,494 distances over 6 replicates, L2: n = 54,483 distances over 4 replicates, T1:

n = 34,040 distances over 6 replicates, half TAD: n = 24,016 distances over 3 replicates. All replicates are biological replicates. **e–g**, Elimination of cells in S or G2 phases of the cell cycle. **e**, Images of unreplicated (top) and replicated (bottom) fluorescent spots at different time points. The same contrast setting was applied for all images. Heatmaps on the right show the spot elongation score, computed to guide replication spot elimination, projected in z and time (Methods). Scale bar = 1 µm. **f**, Distributions of propidium iodide signal assessed by fluorescence-activated cell sorting (FACS) with (blue) or without (light red) a 3-h auxin treatment. **g**, Fraction of cells in G1 or early S phase as determined from images or FACS. For images, the fraction of cells remaining after elimination of cells with replicated spots is shown. Each dot shows the median fraction of cells in G1 (or early S phase) across all fields of view acquired during an imaging experiment. N = 3 biological replicates for FACS with at least 12,000 cells per replicate. For images, no auxin L1: n = 613 cells over 4 replicates, L2: n = 717 cells over 3 replicates, T1: n = 862 cells over 4 replicates, half TAD: n = 445 cells over 3 replicates, adjacent: n = 320 cells over 2 replicates; +auxin: L1: n = 874 cells over 6 replicates, L2: n = 561 cells over 4 replicates, T1: n = 746 cells over 6 replicates, half TAD: n = 320 cells over 3 replicates. All replicates are biological replicates.

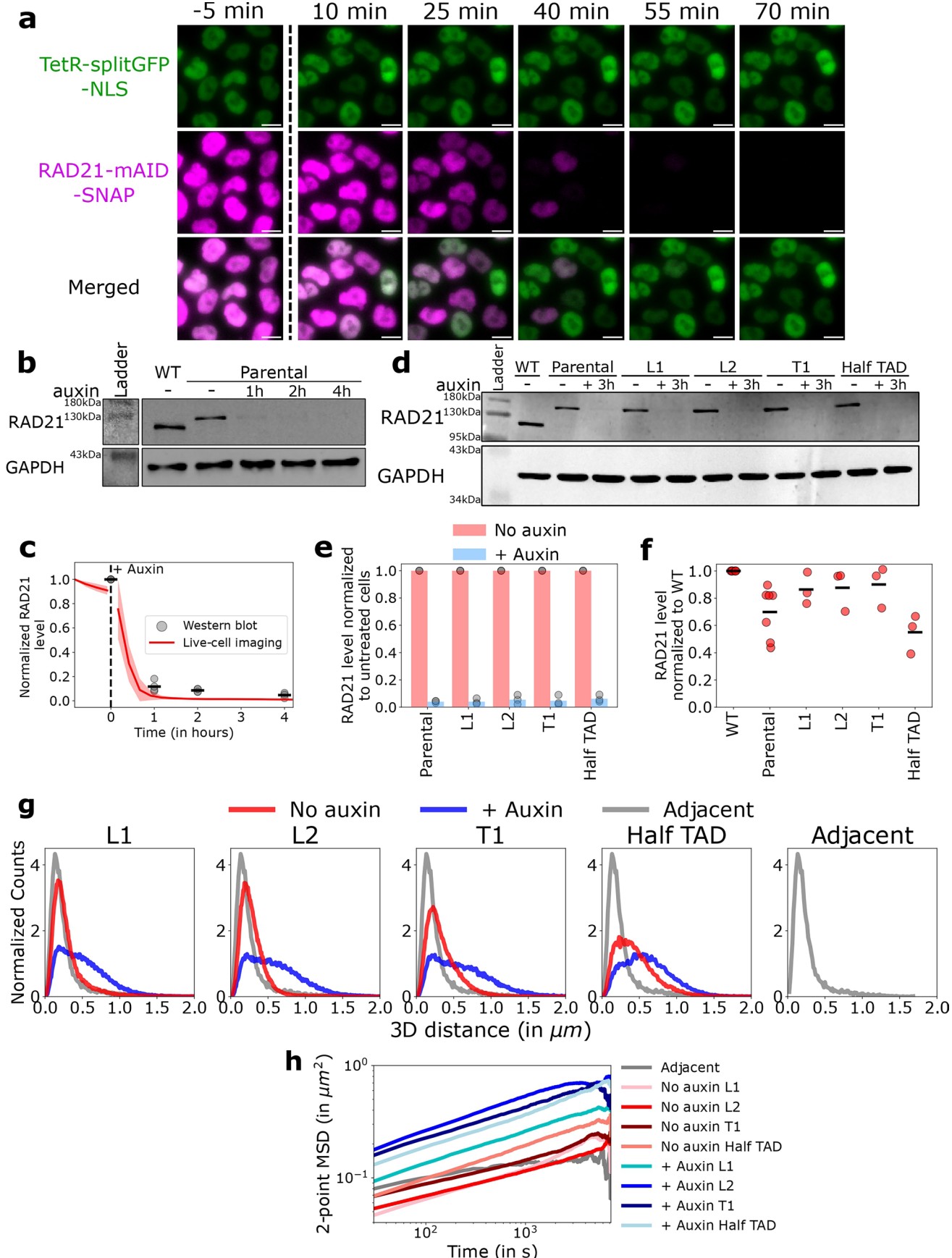

**Extended Data Fig. 3 | See next page for caption.**

**Extended Data Fig. 3 | Efficient and rapid auxin-dependent RAD21 degradation increases anchor-anchor distances and chromatin motion.**
**a**, Live-cell images of cell nuclei after auxin treatment. The green signal (TetR-split-GFPx16-NLS) was used to segment nuclei for quantification of RAD21 levels in **c**. Time after addition of auxin is indicated on top. Representative image from 1 of 4 biological replicates. Scale bar = 10 μm. **b**, Example western blot of RAD21 depletion kinetics upon auxin treatment in the parental cell line. **c**, Quantification of RAD21 level in the parental cell line as function of auxin treatment duration from live-cell imaging (red curve) and western blots (gray dots). For live-cell imaging, each timepoint shows the mean normalized RAD21 intensity over all fields of view from N = 4 biological replicates (no auxin: n = 252, 370, 502, 596 cells per replicate; +Auxin: n = 165, 274, 318, 347 cells per replicate). The red shaded area shows the 95% confidence interval. For western

blot, horizontal black lines indicate the mean over N = 4 biological replicates. The black vertical dashed line shows the timepoint of auxin addition to the medium. **d**, Same as **b** for each cell line containing repeat arrays with or without a 3-h auxin treatment. **e**, Quantification of RAD21 degradation from western blot in the different cell lines without (red) or with (blue) a 3-h auxin treatment. Bars indicate the mean of N = 3 replicates, and each replicate is shown as a distinct gray dot. **f**, Quantification of basal RAD21 degradation in untreated cell lines as compared to WT RAD21 levels. Black horizontal lines indicate the mean of N = 3 replicates (N = 7 replicates for WT and parental cell lines). **g**, Histograms of 3D anchor-anchor distances with (blue) or without (red) auxin treatment and the untreated adjacent locus (gray). **h**, Two-point mean squared displacements (MSD) for untreated (shades of red) and auxin-treated (shades of blue) cell lines, and the adjacent locus (gray).

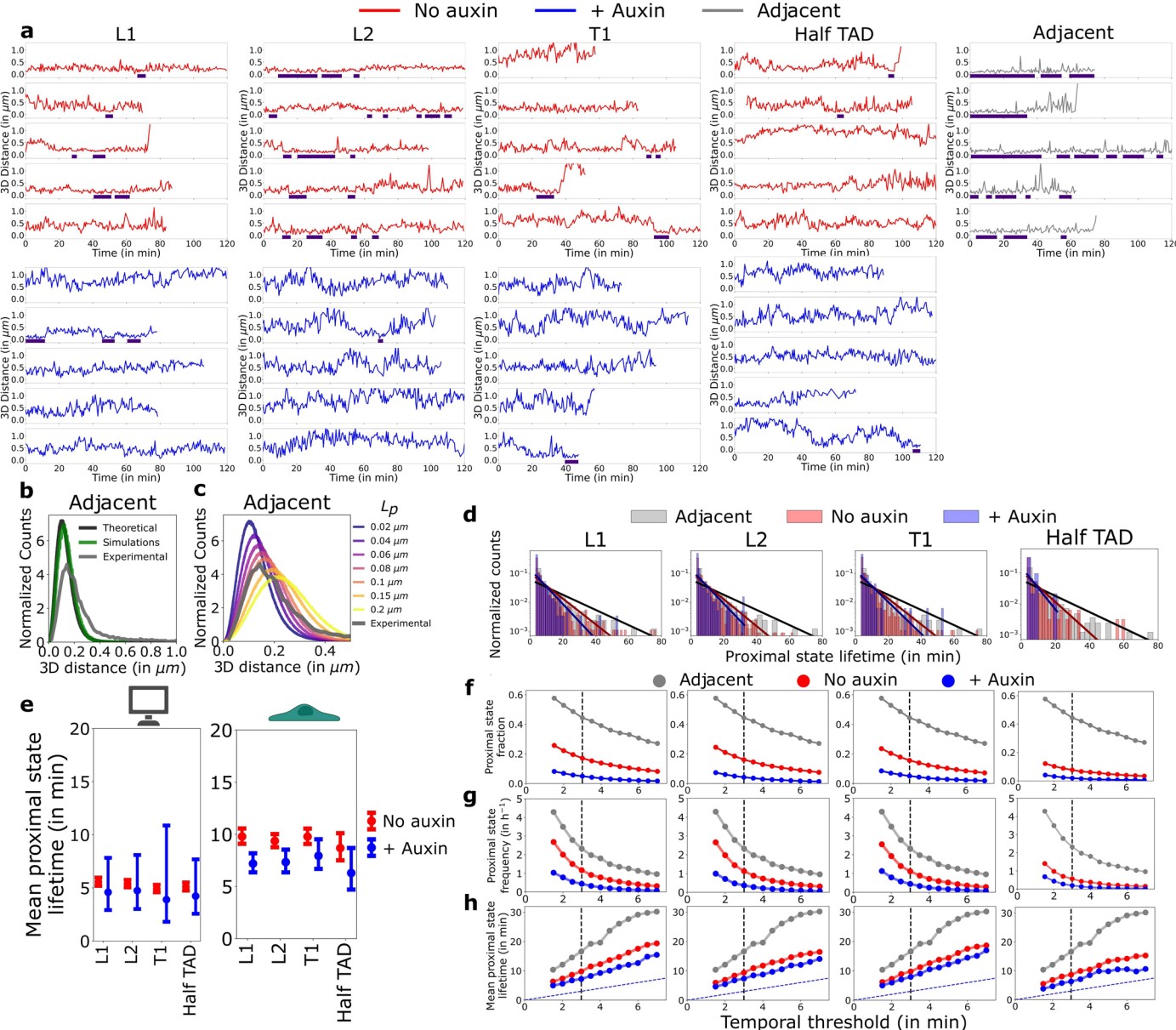

**Extended Data Fig. 4 | Quantification of proximal states. a**, Time series of 3D anchor-anchor distances in untreated (red) or auxin-treated (blue) cells and the adjacent locus (gray). Segmented proximal state intervals are indicated by indigo bars. **b**, Distributions of 3D anchor-anchor distances measured for the experimental adjacent locus (gray) and predicted by polymer simulations (green) or by a theoretical polymer model of closed states (black), assuming 6 kb between reporters, as in the adjacent locus, and taking into account localization errors and photobleaching measured in this cell line. **c**, Distributions of experimental (gray) and theoretical (colored) 3D distances for the adjacent locus. Theoretical distances assume persistence lengths $L_p$ (a measure of rigidity) ranging from 20 to 200 nm, in accordance with prior determinations of 16 to 134 nm in different model species[96–100]. The theoretical model accounts for localization errors, genomic separation between anchors and reporters, and photobleaching. **d**, Distribution of proximal state lifetimes and corresponding exponential fits (solid lines). **e**, Proximal state lifetimes computed from polymer

simulations (left) or estimated from experiments (right). A cohesin density of 12 Mb$^{-1}$ was assumed in simulations of untreated cells (red), and of 1 Mb$^{-1}$ in simulations of auxin-treated cells (blue), matching the experimental 94% RAD21 depletion upon auxin treatment. Simulations assume a cohesin residence time of 22 min and a motor speed of 0.25 kb/s. N = 450 simulated time series were used for estimating lifetimes, matching experimental sample size (Supplementary Table 1). Error bars show the 95% confidence interval. Polymer simulations predict a small decrease in proximal state lifetimes upon auxin treatment, which is also observed in experiments. Panel **e** was created in part with BioRender.com. **f–h**, Fraction (**f**), frequency (**g**) and lifetime (**h**) of proximal states, as function of the temporal threshold used for proximal state segmentation. Each dot corresponds to a distinct temporal threshold. The black dashed line indicates the temporal threshold used to segment proximal states elsewhere in our study. The dashed blue line in **h** shows the temporal threshold itself, and thus represents the smallest measurable proximal state lifetime.

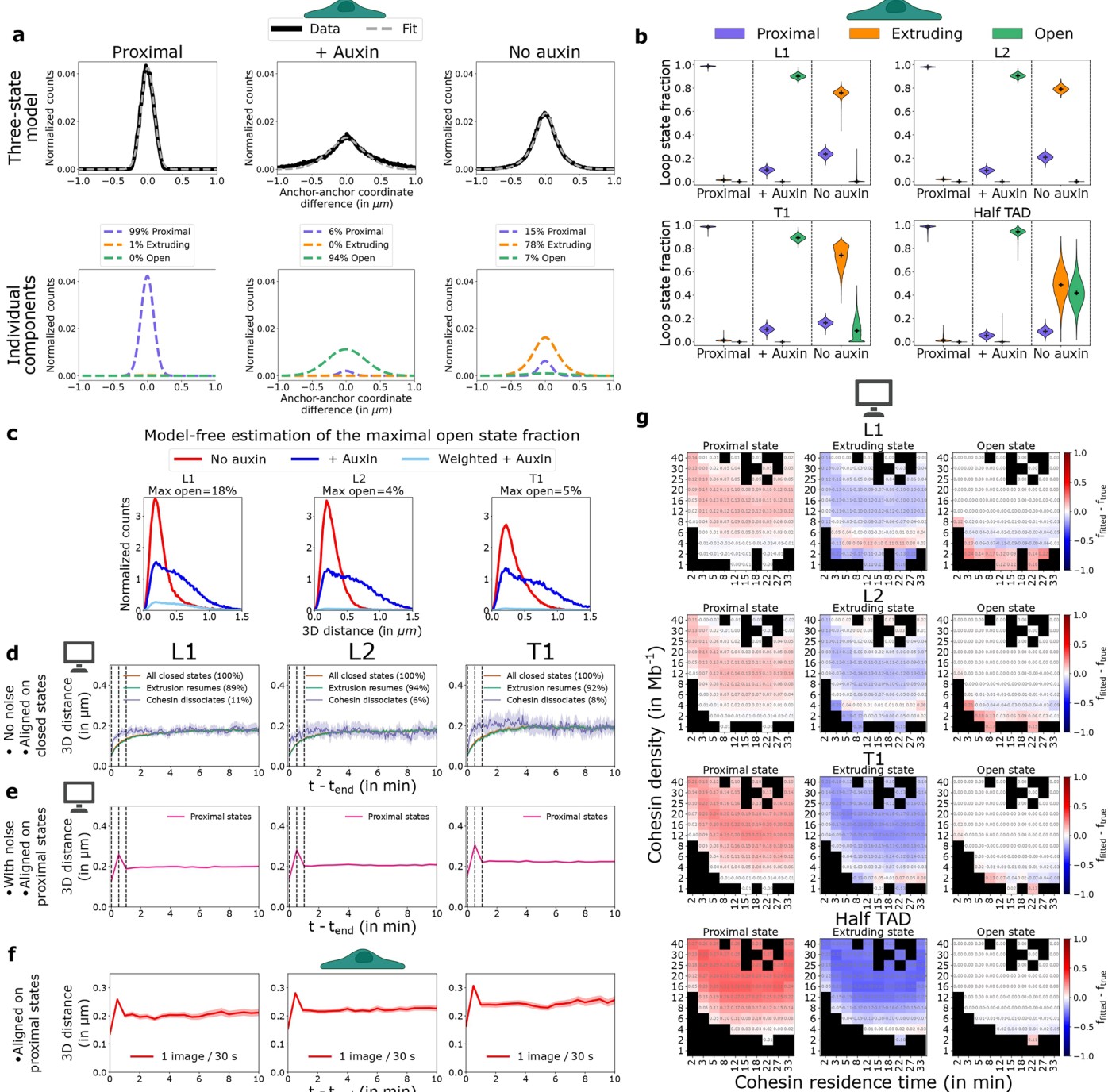

**Extended Data Fig. 5 | See next page for caption.**

**Extended Data Fig. 5 | Fractions of loop states estimated from an analytical model of coordinate difference distributions. a**, Top, experimental distribution of anchor-anchor coordinate differences (solid black lines) measured for the T1 TAD, for time points segmented as proximal states in untreated cells (left), and all time points in auxin-treated (middle) and untreated (right) cells, and the fitted three-state analytical model (dashed gray curves). Bottom, the three components (corresponding to the proximal, extruding, and open states) were estimated from the fitted three-state model for each of these three distributions. The weights of the three components (that is, loop state fractions) are indicated as percentages. **b**, Loop state fractions estimated from the anchor-anchor coordinate difference distributions, in untreated or auxin-treated cells and using only time points segmented as proximal states in untreated cells ('proximal'). The black cross indicates the median, violin plots extend from minimum to maximum values. All distributions of loop state fractions are significantly different from each other within each condition (conditions are separated by vertical dashed lines), as assessed by a Kruskal–Wallis test followed by a Dunn's post hoc test adjusted for multiple comparisons with the Bonferroni correction. N = 10,000 bootstrap samples. **c**, Model-free estimation of the upper limit of open state fractions from distributions of 3D distances. The cyan curve shows the auxin-treated distance distribution weighted by a multiplicative coefficient such that in 95% of distance bins the weighted curve fits below the untreated distance distribution (red). This coefficient provides an upper bound to the fraction of distances in the untreated cells corresponding to open states (indicated as 'max open'). This analysis confirms that open states are rare. **d**–**f**, Polymer relaxation is rare and too transient to be captured at our live-cell imaging frequency.

**d**,**e**, Simulated distance time series sampled at high frequency (1 snapshot per 3 s, **d**) or at the experimental imaging frequency (1 snapshot per 30 s, **e**) and aligned to the end $t_{end}$ of closed (**d**) or proximal (**e**) states. In **d**, time series were aligned on closed states followed by cohesin dissociation (purple), resumption of extrusion (that is, without cohesin dissociation, green) or using all closed states regardless of the cohesin fate (orange, overlapping with green). The relative proportions of cohesin dissociation and extrusion resumption are indicated. Simulated time series in **d** and **e** are without and with experimental noise, respectively. Simulations assume a cohesin density of 12 Mb$^{-1}$, a residence time of 22 min and a motor speed of 1 kb/s. Dotted lines indicate the first three time points corresponding to the experimental live-cell imaging frequency (1 image per 30 s). **f**, Same as **e**, but using experimental data. Shaded areas in **d**–**f** indicate the 95% confidence interval. In **e** and **f**, the first two time points are directly affected by the threshold-based segmentation of proximal states, whereby the distances for the first and second time points must be below and above the spatial threshold, respectively. **g**, Absolute errors in the estimated fractions of proximal, extruding and open states (Fig. 2e) assessed using polymer simulations, as function of cohesin residence time and density, assuming a motor speed of 1 kb/s. For each parameter pair N = 100 bootstrap samples were considered, and for each bootstrap we included the same number of time series as in the experimental dataset. Black squares correspond to pairs of parameters that were not explored or for which less than 50 distances were available to define the proximal state distribution. Panels **a**, **b**, **d**–**g** were created in part with BioRender.com.

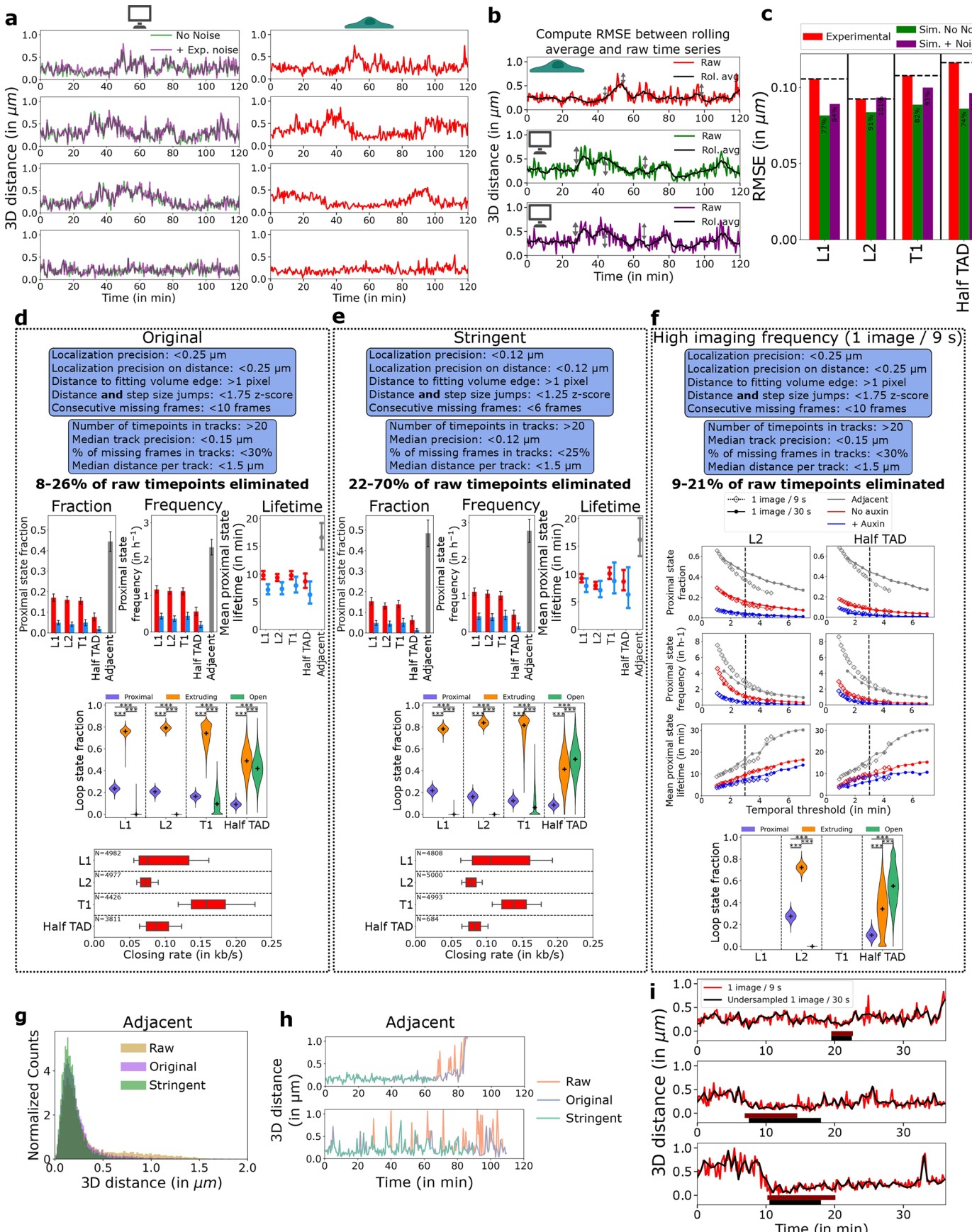

**Extended Data Fig. 6 | See next page for caption.**

**Extended Data Fig. 6 | Quantifications of TAD dynamics are robust to spatial and temporal resolution. a–c**, Comparison of 3D distance fluctuations in simulated and experimental time series. **a**, Simulated time series of 3D anchor-anchor distances with (purple) or without (green) measurement noise consistent with experiments, including localization errors, photobleaching and reporter-anchor separation (left), and experimental 3D distance time series (right). **b**, Raw (colored) and rolling averages (black) of 3D distance time series from experiments (red) or polymer simulations with (purple) or without (green) measurement noise. A 5 min time window was used to compute the rolling average. Panels **a** and **b** were created in part with BioRender.com. **c**, Bars show the root mean squared error (RMSE) between the raw and rolling averaged time series, as a measure of distance fluctuations, for experimental data (red), polymer simulations with (purple) and without (green) measurement noise. Percentages on the green and purple bars are relative to the experimental RMSE (dashed black lines). Experimental measurement noise accounts for only 9–14% of distance fluctuations (RMSE), whereas biophysical motion of the chromatin fiber accounts for 74–91% of RMSE. In **a–c**, simulations assume a cohesin density of 12 Mb$^{-1}$, a residence time of 22 min and a motor speed of 0.25 kb s$^{-1}$. The L2 TAD was used as an example in **a** and **b**. **d–f**, Comparison of results obtained with the original dataset used throughout the article ('original', **d**), the dataset obtained after stringent quality filtering ('stringent', **e**) and a dataset acquired at higher temporal frequency (one image per 9 s instead of 30 s, **f**). Blue boxes show criteria used for quality filtering of raw time series. The minimum and

maximum percentages of raw time points eliminated across all cell lines and treatments are indicated below. Bar plots for proximal state fractions and frequencies show mean values and error bars show the 2.5–97.5 percentiles of n = 10,000 bootstrap samples, boxplot lines of closing rates show the median, lower and upper quartiles, while whiskers extend from 10 to 90 percentiles of n = 5,000 bootstrap samples, for proximal state lifetimes, data are represented as mean ± 95% confidence interval. The number of time series for each condition is indicated in Supplementary Table 1. Violin plots of loop state fractions extend from the minimum to the maximum of n = 10,000 bootstraps and the black cross indicates the median. ***: P-value < 0.001 from a post hoc Dunn's test adjusted for multiple comparisons by the Bonferroni correction, following a Kruskal–Wallis test. Exact P-values can be found in Source Data. In **f**, proximal state fractions, frequencies and lifetimes are plotted as function of temporal thresholds used to segment proximal states for the high imaging frequency (dotted lines) and the original (solid lines) datasets. We could not estimate the closing rate in the high-frequency dataset because of an insufficient number of identified proximal states. The L1 and T1 TADs were not acquired at high imaging frequency. **g,h**, Histograms (**g**) and time series (**h**) of 3D anchor-anchor distances from the Adjacent locus in the raw, original and stringent datasets. **i**, Distance time series of the L2 TAD acquired at an imaging frequency of one image per 9 s (red) and the time series interpolated and undersampled at the original imaging frequency of one image per 30 s (black). Dark red and black bars indicate segmented proximal states using the high-frequency or undersampled time series, respectively.

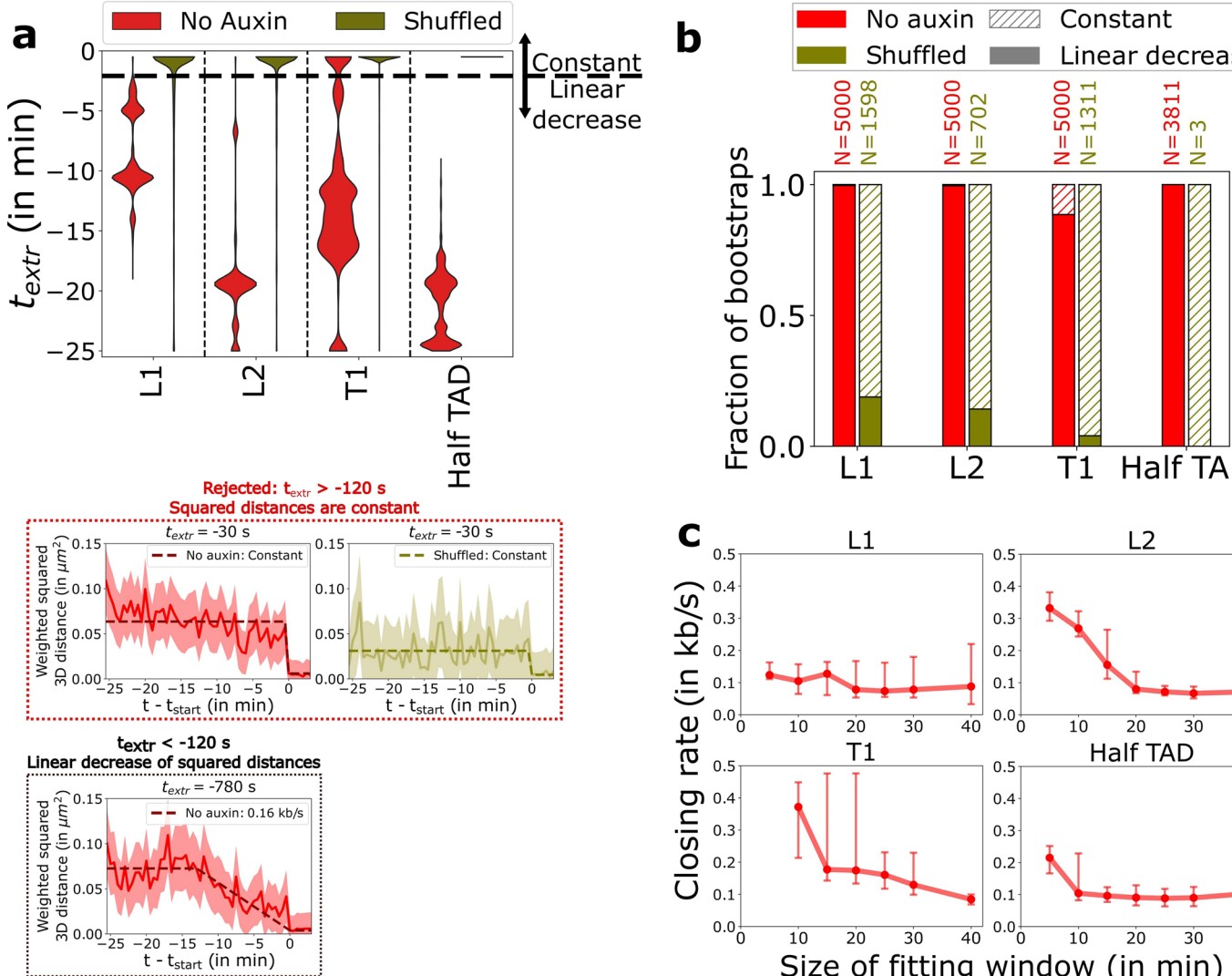

**Extended Data Fig. 7 | Estimation of closing rate from aligned squared distances. a**, Top, violin plots of $t_{extr}$ values for untreated (red) or shuffled (olive) time series. The horizontal dashed line shows the threshold (corresponding to three time points, that is, 120 s) used to distinguish between constant (above threshold) and linearly decreasing (below threshold) averaged squared distance time series. Violin plots extend from the minimum to the maximum of n = 5,000 bootstrap samples. Bottom, examples of averaged squared distance time series consistent with a constant (red dotted rectangle) or linearly decreasing (black dotted rectangle) function. The fitted value of $t_{extr}$ is indicated. Time series from the T1 TAD were used as examples. Shaded areas indicate the 95% confidence interval. **b**, Fraction of bootstrap samples exhibiting constant (hatched) or linearly decreasing (filled) squared distances for untreated (red) or randomly shuffled (olive) time series. The number of bootstrap samples with a minimum of 15 time series averaged, out of a total of N = 5,000 bootstrap samples, is indicated. **c**, Estimated closing rates as function of the fitting window size for untreated cells. Data are represented as mean values and error bars extend from the 10th to 90th percentiles of N = 5,000 bootstrap samples from which bootstraps with less than 15 time series or exhibiting constant distances were excluded.

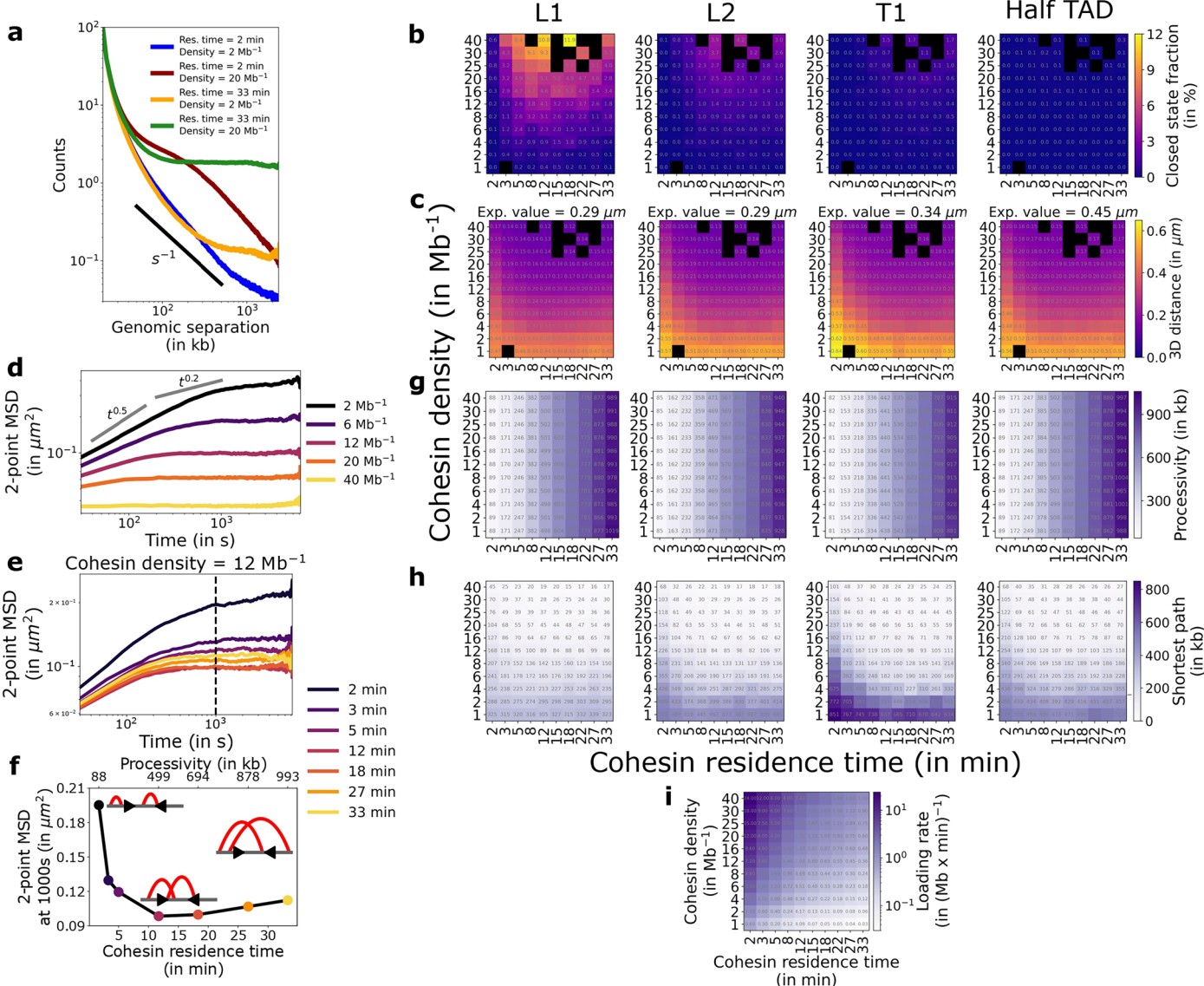

**Extended Data Fig. 8 | Influence of cohesin residence time and density on TAD dynamics. a**, Contact counts as function of genomic separation for the L1 TAD, corresponding to the simulated contact maps of Fig. 4c. **b,c,g–i**, Closed state fraction (**b**), 3D anchor-anchor distance (**c**), cohesin processivity (**g**), shortest 1D path connecting TAD anchors, that is, unextruded length (**h**) and cohesin loading rate (**i**) as function of cohesin density and residence time. **d,e**, Two-point MSD curves for polymers with different cohesin densities (**d**) or residence times (**e**), for the L1 TAD. **f**, Absolute 2-point MSD value for an interval of 1,000 s (dashed line in **e**) as function of cohesin residence time (bottom axis) or processivity (top axis). Schematics represent the length of loops at different cohesin residence times. Simulations used for this figure assume a cohesin motor speed of 1 kb/s.

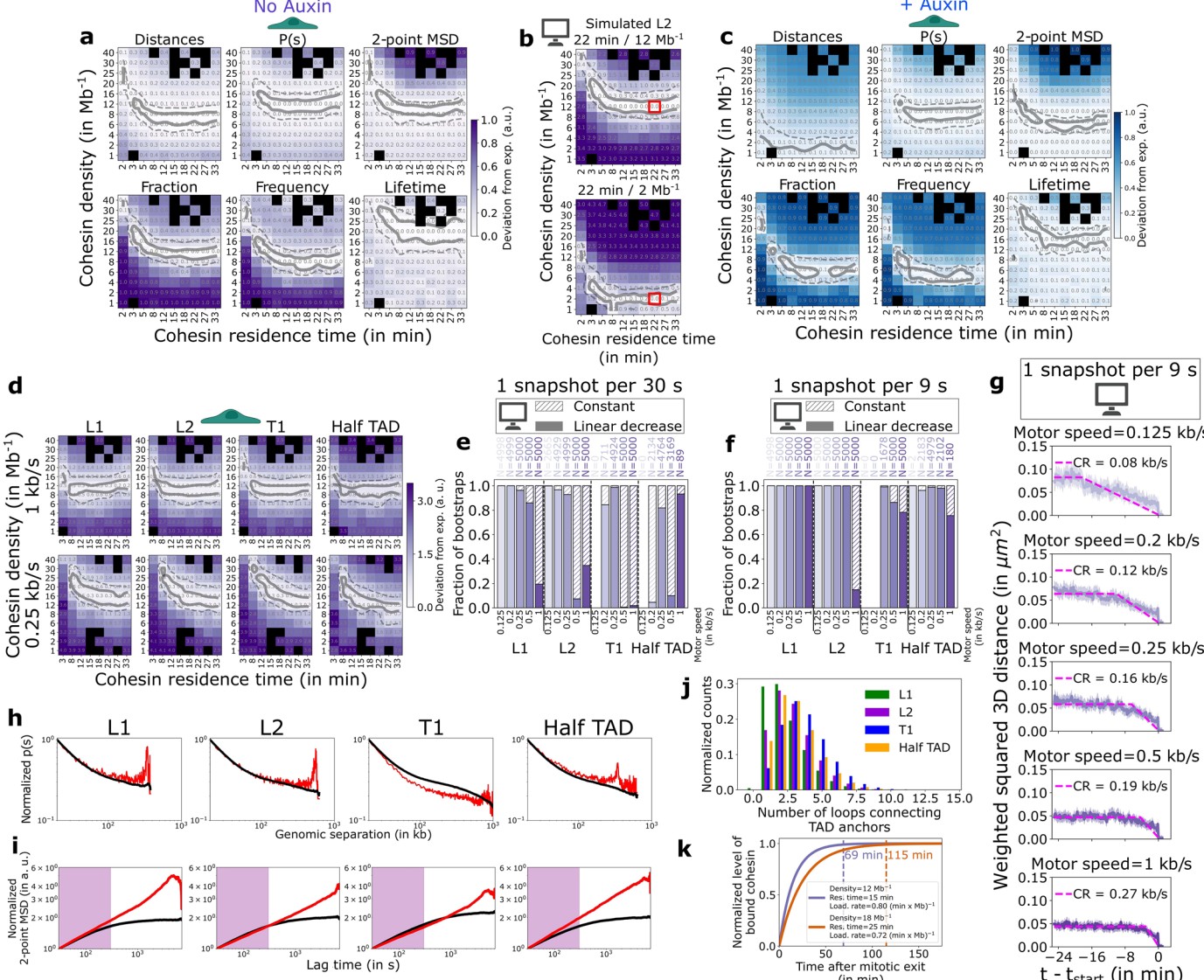

**Extended Data Fig. 9 | Comparison of simulated and experimental TAD dynamics. a**, Deviations of polymer simulations from experiments for each metric of the L2 TAD in untreated cells. The sum of these six deviation maps yields the deviation map shown in Fig. 5a. **b**, Deviations as in **a**, but where experimental data are replaced by data generated from polymer simulations. The assumed cohesin density and residence time parameters are indicated and highlighted by red boxes. The L2 TAD was used as an example. **c**, Same as **a**, but using experimental data from auxin-treated cells. The parameter sets consistent with the data do not agree between different metrics: distances point to the expected low (<1 Mb⁻¹) cohesin density in auxin-treated cells, whereas proximal fraction or frequency point to a density of 6 Mb⁻¹. Simulations used in **a**–**c** assume a cohesin motor speed of 1 kb/s. **d**, Deviation of polymer simulations from experiments, as in Fig. 5a, but using the same sets of parameter combinations for both motor speeds. In **a**–**d**, solid and dashed gray lines indicate the 10% and 25% best parameter sets, respectively. **e**,**f**, Fraction of bootstrap samples consistent with constant (hatched) or linearly decreasing (filled) mean squared distances

from simulations used in Fig. 5e and using a temporal sampling of 1 snapshot per 30 s as in our original imaging experiments (**e**) or 1 snapshot per 9 s as in the high frequency experiments (**f**). The number N of bootstrap samples with at least 15 time series is indicated. At the lower imaging frequency of 1 snapshot per 30 s, N = 0 for T1 and half TAD. **g**, Same as Fig. 5e, but using the higher temporal sampling of 1 snapshot per 9 s instead of 30 s. Panels **a**–**g** were created in part with BioRender.com. **h**,**i**, Simulated (black) and experimental (red) normalized p(s) (**h**) and 2-point MSD (**i**) curves. In **i**, the purple area shows the part of the curve used to compare simulations and experiments. **j**, Number of loops connecting TAD anchors by the shortest 1D path at any timepoint. Simulations used for **h**–**j** assume a cohesin density of 12 Mb⁻¹, a residence time of 22 min and a motor speed of 0.25 kb/s. **k**, Analytical prediction of the time evolution of chromatin-bound cohesin following mitotic exit using the parameters estimated from comparing polymer simulations to experiments. Vertical dashed lines indicate the timepoint at which 99% of the cohesin is bound. The predictions are consistent with experimental observations of cohesin reestablishment times of 60–120 min[6,63].

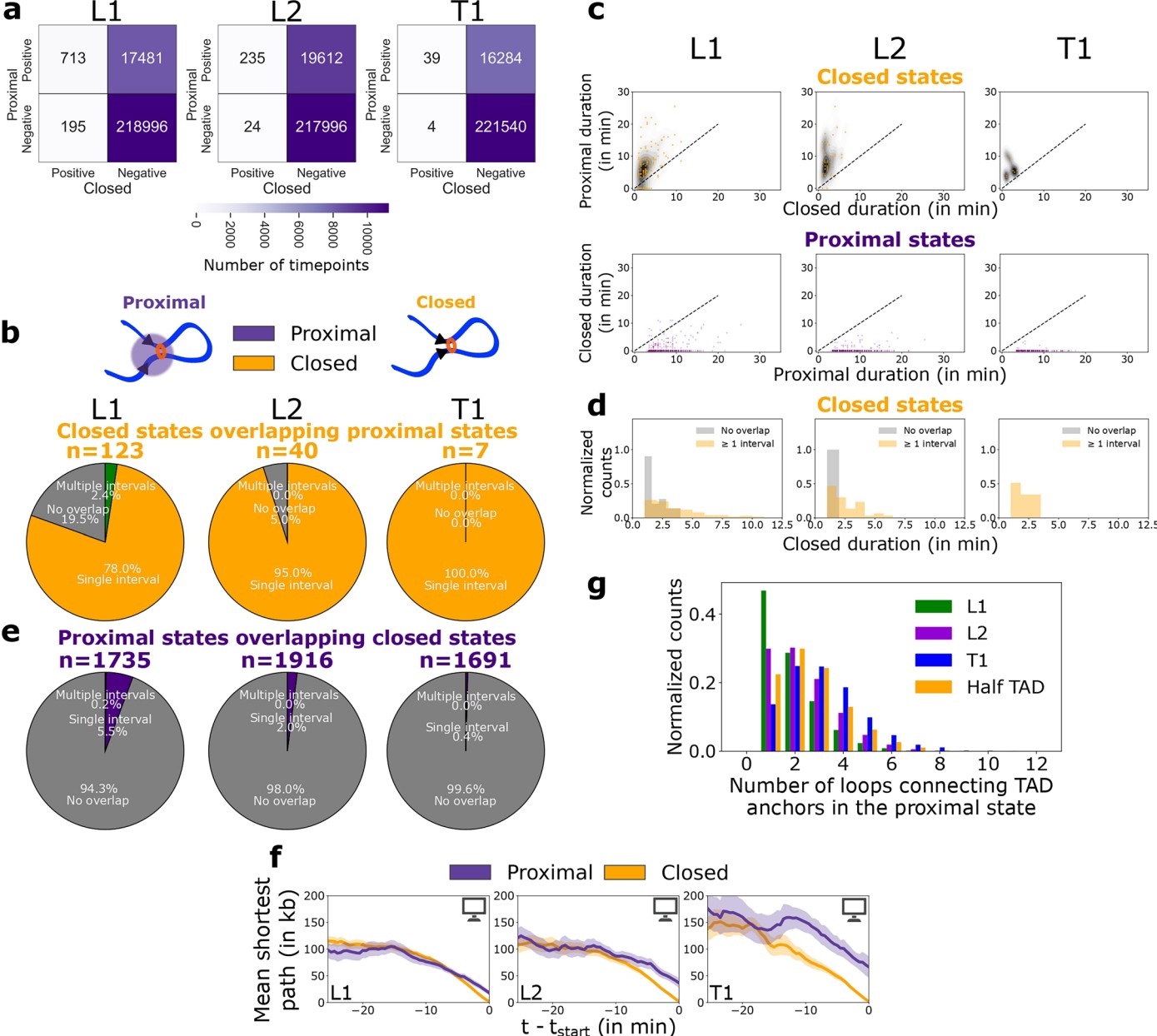

**Extended Data Fig. 10 | Comparison of closed and proximal states.**
**a**–**f**, Comparison of closed and proximal states using polymer simulations. **a**, The number of time points assigned to each class is shown. **b**,**e**, Percentages of closed state intervals overlapping proximal state intervals (**b**, orange) and percentages of proximal state intervals overlapping closed state intervals (**e**, purple). 'Multiple intervals' (green) designate cases with more than one overlapping interval. **c**, Duration of all closed states compared to proximal states (top) and duration of all proximal states compared to closed states (bottom). If multiple intervals overlapped a single one, the duration of the overlapping intervals was summed. Kernel density estimations are shown in gray. **d**, Histograms show the distribution of closed state durations for closed state intervals overlapped by at least one proximal state (orange) or not (gray). **f**, Simulated time series show the mean shortest 1D path (unextruded DNA length) aligned on the starting time $t_{start}$ of closed (orange) or of segmented proximal (purple) states. Shaded areas indicate the 95% confidence interval. Panels **b** and **f** were created in part with BioRender.com. **g**, Number of loops connecting TAD anchors by the shortest 1D path in proximal states. All simulations used in this figure assume a cohesin density of 12 Mb$^{-1}$, a residence time of 22 min and a motor speed of 0.25 kb/s.

Edouard Bertrand
Christophe Zimmer

# Reporting Summary

## Statistics

For all statistical analyses, confirm that the following items are present in the figure legend, table legend, main text, or Methods section.

| n/a | Confirmed | |
|---|---|---|
| ☐ | ☒ | The exact sample size (*n*) for each experimental group/condition, given as a discrete number and unit of measurement |
| ☐ | ☒ | A statement on whether measurements were taken from distinct samples or whether the same sample was measured repeatedly |
| ☐ | ☒ | The statistical test(s) used AND whether they are one- or two-sided *Only common tests should be described solely by name; describe more complex techniques in the Methods section.* |
| ☐ | ☒ | A description of all covariates tested |
| ☐ | ☒ | A description of any assumptions or corrections, such as tests of normality and adjustment for multiple comparisons |
| ☐ | ☒ | A full description of the statistical parameters including central tendency (e.g. means) or other basic estimates (e.g. regression coefficient) AND variation (e.g. standard deviation) or associated estimates of uncertainty (e.g. confidence intervals) |
| ☐ | ☒ | For null hypothesis testing, the test statistic (e.g. *F*, *t*, *r*) with confidence intervals, effect sizes, degrees of freedom and *P* value noted *Give P values as exact values whenever suitable.* |
| ☒ | ☐ | For Bayesian analysis, information on the choice of priors and Markov chain Monte Carlo settings |
| ☒ | ☐ | For hierarchical and complex designs, identification of the appropriate level for tests and full reporting of outcomes |
| ☒ | ☐ | Estimates of effect sizes (e.g. Cohen's *d*, Pearson's *r*), indicating how they were calculated |

*Our web collection on statistics for biologists contains articles on many of the points above.*

## Software and code

Policy information about availability of computer code

| Data collection | For Western Blot imaging, we used the Chemidoc MP Imaging system (Bio-Rad).. For FACS, we used a Miltenyi MACSQuant Analyzer 10 Flow Cytometer with the 488 nm laser and a 692/75 nm band pass filter. Time lapse image acquisition was performed with an inverted microscope (Nikon) coupled to the Dragonfly spinning disk (Andor) using a 100X Plan Apo 1.45 NA oil immersion objective. |
|---|---|
| Data analysis | ChopChop v3, ImageJ v2.14, Labkit v0.4, TrackMate v7, FlowJo v10, FastQC v0.12.1, Cutadapt v0.6.10, HiC-Pro v3.1.0, Bowtie2 v2.2.6.2, Juicer 1.19.02, FIMO v5.3.0, pgltools v2.7.1, MACS v2.1, Chromagnon v0.94, LAMMPS Nov16, Python v3.8+. The code (v095) used to generate polymer simulations, process live-cell microscopy images, quantify the distance between TAD anchors, and process the Micro-Capture C data is available at: https://github.com/imodpasteur/Sabate_et_al_TAD_Anchors and on Zenodo: 10.5281/zenodo.16949930. |

For manuscripts utilizing custom algorithms or software that are central to the research but not yet described in published literature, software must be made available to editors and reviewers. We strongly encourage code deposition in a community repository (e.g. GitHub). See the Nature Portfolio guidelines for submitting code & software for further information.

## Data

Policy information about availability of data

All manuscripts must include a data availability statement. This statement should provide the following information, where applicable:

- Accession codes, unique identifiers, or web links for publicly available datasets
- A description of any restrictions on data availability
- For clinical datasets or third party data, please ensure that the statement adheres to our policy

Capture Micro-C data have been uploaded to the Gene Expression Omnibus (GEO) under accession GSE273257. This paper analyzed existing, publicly available Hi-C, ChIP-Seq and PRO-Seq data from GEO under accession GSE104334. Raw and quality-filtered distance time series are available on Zenodo94 at: 10.5281/zenodo.16949930.

## Research involving human participants, their data, or biological material

Policy information about studies with human participants or human data. See also policy information about sex, gender (identity/presentation), and sexual orientation and race, ethnicity and racism.

| | |
|---|---|
| Reporting on sex and gender | *Use the terms sex (biological attribute) and gender (shaped by social and cultural circumstances) carefully in order to avoid confusing both terms. Indicate if findings apply to only one sex or gender; describe whether sex and gender were considered in study design; whether sex and/or gender was determined based on self-reporting or assigned and methods used. Provide in the source data disaggregated sex and gender data, where this information has been collected, and if consent has been obtained for sharing of individual-level data; provide overall numbers in this Reporting Summary. Please state if this information has not been collected. Report sex- and gender-based analyses where performed, justify reasons for lack of sex- and gender-based analysis.* |
| Reporting on race, ethnicity, or other socially relevant groupings | *Please specify the socially constructed or socially relevant categorization variable(s) used in your manuscript and explain why they were used. Please note that such variables should not be used as proxies for other socially constructed/relevant variables (for example, race or ethnicity should not be used as a proxy for socioeconomic status). Provide clear definitions of the relevant terms used, how they were provided (by the participants/respondents, the researchers, or third parties), and the method(s) used to classify people into the different categories (e.g. self-report, census or administrative data, social media data, etc.) Please provide details about how you controlled for confounding variables in your analyses.* |
| Population characteristics | *Describe the covariate-relevant population characteristics of the human research participants (e.g. age, genotypic information, past and current diagnosis and treatment categories). If you filled out the behavioural & social sciences study design questions and have nothing to add here, write "See above."* |
| Recruitment | *Describe how participants were recruited. Outline any potential self-selection bias or other biases that may be present and how these are likely to impact results.* |
| Ethics oversight | *Identify the organization(s) that approved the study protocol.* |

Note that full information on the approval of the study protocol must also be provided in the manuscript.

# Field-specific reporting

Please select the one below that is the best fit for your research. If you are not sure, read the appropriate sections before making your selection.

☒ Life sciences          ☐ Behavioural & social sciences          ☐ Ecological, evolutionary & environmental sciences

For a reference copy of the document with all sections, see nature.com/documents/nr-reporting-summary-flat.pdf

# Life sciences study design

All studies must disclose on these points even when the disclosure is negative.

| | |
|---|---|
| Sample size | No statistical method was used to predetermine sample size. Live-cell imaging was performed in 2-6 biological replicates, resulting in 12,269-78,268 distances measured. Further statistics are summarized in Supplementary Table 1. Capture Micro-C was performed in two biological replicates, except for the Half TAD and the untagged cell lines where a single replicate was performed, following standard in the field. For FACS, 3 biological replicates were recorded with at least 12,000 cells within the final gate per condition. When bootstrap was used to evaluate the standard deviation of our estimates, we used 5,000-10,000 bootstrap samples. Number of replicates was chosen based on standards in the field. |
| Data exclusions | No data were excluded from the analysis, except during quality filtering of image time series, as detailed in the 'Quality filtering of trajectories' (see Supplementary Information). |
| Replication | For live-cell imaging experiments, we performed 2-6 biological replicates with 1-5 technical replicates each. Capture Micro-C was performed |

| Replication | in two biological replicates, except for the Half TAD and untagged cell lines where a single replicate was performed. For FACS, 3 biological replicates were recorded with at least 12,000 cells within the final gate per condition. Western blot was performed in 3-7 biological replicates. |
| Randomization | No randomization was performed as the study did not require sample allocation into different groups. Cells imaged in live-cell microscopy were chosen randomly. |
| Blinding | Blinding was not possible for data collection, as data acquisition required identification of the samples for further processing. Data analysis was not performed blind to the conditions of the experiments, except for Capture Micro-C analysis. |

# Reporting for specific materials, systems and methods

We require information from authors about some types of materials, experimental systems and methods used in many studies. Here, indicate whether each material, system or method listed is relevant to your study. If you are not sure if a list item applies to your research, read the appropriate section before selecting a response.

## Materials & experimental systems

| n/a | Involved in the study |
|---|---|
| ☐ | ☒ Antibodies |
| ☐ | ☒ Eukaryotic cell lines |
| ☒ | ☐ Palaeontology and archaeology |
| ☒ | ☐ Animals and other organisms |
| ☒ | ☐ Clinical data |
| ☒ | ☐ Dual use research of concern |
| ☒ | ☐ Plants |

## Methods

| n/a | Involved in the study |
|---|---|
| ☒ | ☐ ChIP-seq |
| ☐ | ☒ Flow cytometry |
| ☒ | ☐ MRI-based neuroimaging |

## Antibodies

| Antibodies used | Rabbit polyclonal anti-RAD21 Abcam Cat# ab154769, RRID:AB_2783833, Dilution: 1/1500<br>Mouse monoclonal anti-GAPDH Abcam Cat# ab8245, RRID:AB_2107448, Dilution: 1/50000, clone [6C5]<br>Goat anti-rabbit IR800 Advansta R-05060, Dilution: 1/10000<br>Goat anti-mouse IR800 Advansta R-05061, Dilution: 1/10000 |
| Validation | The anti-RAD21 antibody was validated by: (i) a shift in the protein size in cells where the endogenous size was modified with a mAID-SNAP tag, as compared to WT cells,<br>(ii) the absence of detected bands upon auxin-dependent degradation.<br>The GAPDH antibody was tested by the manufacturer (Abcam) on human samples for both Western blot and immunofluorescence. |

## Eukaryotic cell lines

Policy information about cell lines and Sex and Gender in Research

| Cell line source(s) | All cell lines described are derived from WT HCT116 cells from ATCC CCL-247. |
| Authentication | Cell lines have been recurrently used and have not been authenticated. Homozyguous insertion of repeat arrays showed two fluorescent spots as expected from this diploid cell line. |
| Mycoplasma contamination | Cells were tested monthly for the presence of Mycoplasma spp., Ureaplasma spp. and A. laidlawii by qPCR. |
| Commonly misidentified lines (See ICLAC register) | No commonly misidentified cell line was used. |

## Plants

| Seed stocks | Report on the source of all seed stocks or other plant material used. If applicable, state the seed stock centre and catalogue number. If plant specimens were collected from the field, describe the collection location, date and sampling procedures. |
| Novel plant genotypes | Describe the methods by which all novel plant genotypes were produced. This includes those generated by transgenic approaches, gene editing, chemical/radiation-based mutagenesis and hybridization. For transgenic lines, describe the transformation method, the number of independent lines analyzed and the generation upon which experiments were performed. For gene-edited lines, describe the editor used, the endogenous sequence targeted for editing, the targeting guide RNA sequence (if applicable) and how the editor was applied. |
| Authentication | Describe any authentication procedures for each seed stock used or novel genotype generated. Describe any experiments used to assess the effect of a mutation and, where applicable, how potential secondary effects (e.g. second site T-DNA insertions, mosiacism, off-target gene editing) were examined. |

# Flow Cytometry

## Plots

Confirm that:

☒ The axis labels state the marker and fluorochrome used (e.g. CD4-FITC).

☒ The axis scales are clearly visible. Include numbers along axes only for bottom left plot of group (a 'group' is an analysis of identical markers).

☒ All plots are contour plots with outliers or pseudocolor plots.

☒ A numerical value for number of cells or percentage (with statistics) is provided.

## Methodology

| | |
|---|---|
| Sample preparation | 300,000 cells were seeded and grown in 6-well plates and cells were collected 48 hours later. Cells were trypsinized, washed in PBS, and resuspended in 500 μL PBS. 5.5 mL of ice-cold 70% ethanol was added for fixation.<br>Cells were kept at 4°C in 70% ethanol for at least 12 hours until staining. For staining, cells were washed twice in PBS and incubated for 5 min at room temperature with 50 μL of a 50 μg/mL RNAse A solution (Promega A7973).<br>Finally, 400 μL of 50 μg/mL propidium iodide solution was added and cells were incubated for 15 min at room temperature before FACS sorting. |
| Instrument | We used a Miltenyi MACSQuant Analyzer 10 Flow Cytometer with the 488 nm laser and a 692/75 nm band pass filter. |
| Software | FlowJo v10 (BD Biosciences) |
| Cell population abundance | >12,000 cells in the final gate |
| Gating strategy | Forward / Side scatters were used to discard large cells with high granularity. Propidium iodide height vs propidium iodide area was used to discard doublets, cells in M phase and cells with a ploidy stricly higher than 2.<br>The histogram of propidium area was then fitted by an unconstrained Dean-Jett-Fox model using FlowJo to obtain the fraction of cells in G1. |

☒ Tick this box to confirm that a figure exemplifying the gating strategy is provided in the Supplementary Information.

