## [Peer Review File · Nature Genetics]

Uniform dynamics of cohesin-mediated loop extrusion

Corresponding Author: Dr Thomas Sabaté

A version of this paper was originally rejected for publication by Nature Genetics, however that decision was reconsidered after appeal by the authors.

Version 0:

Decision Letter:

11th Sep 2024

Dear Dr Sabaté,

Your Article entitled "Universal dynamics of cohesin-mediated loop extrusion" has now been seen by 2 referees, whose comments are attached. While they find your work of potential interest, they have raised serious concerns which in our view are sufficiently important that they preclude publication of the work in Nature Genetics, at least in its present form.

Reviewer #1 acknowledges that your study is cutting-edge and interesting; however, they question the validity of some of the assumptions your paper is based on.

Reviewer #2 is overall more positive, and, among other requests, they highlight the need to tone down the overstatement of your results' broad generalizability.

Should further experimental data allow you to fully address these criticisms we would be willing to consider an appeal of our decision (unless, of course, something similar has by then been accepted at Nature Genetics or appeared elsewhere). This includes submission or publication of a portion of this work someplace else.

The required new experiments and data include, but are not limited to those detailed here. We hope you understand that until we have read the revised manuscript in its entirety we cannot promise that it will be sent back for peer review.

If you are interested in attempting to revise this manuscript for submission to Nature Genetics in the future, please contact me to discuss a potential appeal. Otherwise, we hope that you find our referees' comments helpful when preparing your manuscript for resubmission elsewhere.

Thank you.

All the best,
Chiara

Chiara Anania, PhD
Associate Editor
Nature Genetics
<https://orcid.org/0000-0003-1549-4157>

Referee expertise:

Referee #1: imaging, 3D genome organization

Referee #2: chromatin biology

Referee #3:

Reviewers' Comments:

Reviewer #1:

Remarks to the Author:

Sabate et al. report on live-cell dynamics of TADs and use computer simulations to extract quantitative information on various aspects of TAD dynamics. They fluorescently tag three TADs anchor regions in HCT116 colon cancer cells and confirm by timelapse imaging earlier observations demonstrating that TADs anchor distances fluctuate and that TAD anchors are only paired transiently. They define three anchor pairing states (open, extruding, closed) and using this approach they then determine various properties of anchor sites, including the rate at which anchors are brought together and extrusion rate. A major conclusion is that most TADs are being extruded at any time. Computer simulations are then used to probe the effect of residence times, extrusion rates etc. on TADs and to compare simulations to experimental data. Using this approach, the in vivo extrusion speed is determined to be ~0.1kb/s.

These experiments are similar to two recent studies which analyzed the live-cell dynamics of TAD boundaries. Many of reported results are in line with the published studies, including the observation of generally dynamic TAD behavior and low frequency of anchor pairing. This study extends the earlier ones in that it analyzes three instead of only one TAD, does so in human non-stem cells, and uses more elaborate simulation approaches to extract additional quantitative information on cohesion loading frequency etc.

Although this is a technically cutting-edge approach and quantitative information about TAD dynamics are of interest, there is concern that the methods are not quite sensitive enough to draw some of the conclusions presented and the validity of the deduced numbers is difficult to determine. The limited resolution of imaging for detection of the very small distance fluctuations the TAD anchors undergo combined with the use of various necessary assumptions built into computational models reduce the precision – and confidence – in the reported quantitative measurements.

1) the analysis and interpretation are based on the central assumption stated on p. 8 that anchor pairs exist in three states (open, extruding, closed). It is not clear that this is a correct assumption for the analysis. Particularly, it is questionable that all anchor pairs which are not paired or are not fully open must be extruding. For example, after extrusion is complete and cohesin dissociates, anchors may separate as the domain relaxes into an open state. This non-extruding state would be mis-assigned in this analysis as loop extrusion. Similarly, as extrusion commences, anchors are separate but may be extruding (they would be mis-classified as open and not extruding). The authors also assume that a closed state can only be maintained by bound cohesin; I am not sure there is any good evidence to demonstrate that assumption. As is, it is not clear how accurately active loop extrusion can be ascertained from these proximity measurements, thus strongly confounding the interpretation of the data. Unless the authors can show that there is a strict relationship between distance and actual extrusion activity, the three-state model should probably not be used to extract information about extrusion dynamics. As a consequence, several key conclusions, including the most interesting finding that most TADs undergo constant extrusion, are not convincingly supported. The inaccuracy of the state assignments may also explain the relatively large ranges of values the authors obtain for various parameters (see below).

2) even accepting the three-state model, the interpretation of the data relies heavily on accurate calling of paired or unpaired anchors. It is not evident from the presented data how accurately these states can be called from the rather noisy data. The data shown in figure 1g suggests quite small distance differences between paired and unpaired anchors and the traces are quite noisy (as expected for these types of measurements). This concern is highlighted in Ext. Fig 8, where large spikes in distances in the adjacent control appear to be called proximal, whereas several "flat" regions in L1, L2 and L3 are called non-proximal. Similarly, the authors only find 44% of anchor pairs in proximity in their adjacent TAD control which consist of two labelled sites only 6kb apart. Why is this pairing rate not much higher? It is not clear how accurately these states can be called using the imaging methods used and how the calling of the states was validated other than by simulation.

3) the non-TAD control behaved in several respects similar to the TAD anchor sites. For example, there was no apparent difference in lifetime of TAD anchors vs the non-TAD anchor and the fraction of proximal anchors was only about 2-fold lower for non-TAD anchors compared to TAD anchors. Similarly, the fraction of proximal pairs was reduced by ~ 3-fold both for TADs and the non-TAD control, with no apparent preference for the TAD anchors. It was overall not clear to me whether the authors argue for distinct behavior of TADs vs non-TADs anchors or whether they interpret their findings as similar behavior of TADs and non-TADs. If the authors want to claim distinct behavior, please provide statistical analysis of these measurements comparing TAD vs non-TAD anchors. If the authors, on the other hand, want to argue that there is no difference in TAD vs non-TAD anchors, this conclusion needs to be strengthened. It is currently made based on analysis of a single non-TAD region, making it very tenuous. Additional non-TAD regions should be analyzed to put it on firmer grounds.

4) While the computer simulations are useful, the ranges of many of the estimates seem very large to the point of not being very meaningful; for example, the cohesin residence time is estimated to be between 3-33min and cohesin loading rates ranging from 0.2-2.8. These are order of magnitude estimates and as such not overly useful. If these ranges represent technical limitations of the approach, it lessens the impact of the methods used here; if they reflect biological variability, this should be shown more directly.

Additional points:

While figure 1c shows an effect of depletion of RAD21, it does not show as the authors claim on p. 7 that tagging does not

affect the TAD. For this they would need to compare the tagged line with the untagged line. I did not see this comparison in figure 1c or elsewhere. Please include.

The authors show that Auxin treatment led to a shortening of proximal state lifetimes by only ~1.3-fold (Fig. 2d), which is surprisingly low if the authors' assumption that proximal states requires cohesin binding as stated in their model description. They argue the small effect "might be a consequence of residual loop formation due to incomplete cohesin degradation (Extended Data Fig. 4e)". However, they state on p. 8, that degradation of RAD21 was >94% efficient. These two statements seem to contradict each other. If degradation was only partial, the interpretation of these data is in jeopardy, if the degradation was efficient, another explanation for the lack of effect on proximal states needs to be found.

Reviewer #2:

Remarks to the Author:

In this work, Sabate et al. describe the first CTCF-CTCF loop dynamics in living human cells, concluding that also in human cells, the CTCF-CTCF loops are rare and short-lived. The work is well done, and the conclusions are strong. Even if this is not the first study to quantify CTCF-CTCF loop dynamics, it is the first one to address this in human cells, whose genomes and cell cycles are significantly different from the mouse ones. Therefore, the fact that this process is conserved among species, makes this work valuable for a journal like Nature Genetics. To my opinion, the strongest point of this paper is that no matter the species, CTCF-CTCF loops are short-lived.

I believe that this paper is almost ready for publications after some polishing.

1) While it's true that in Gabriele and Mach they used mESCs and only one single locus they made important quantifications, which are a bit underdiscussed in the introduction. In the paragraph in line 78 the authors only to present the limitations of those two studies without stressing enough the relevance of the results. In my opinion, to be fair, this paragraph should discuss some of the conclusions of those two papers, highlighting that the frequency of looping depends on CTCF configuration, but the duration of the looped state is independent and short-lived in both studies. This could serve as a bridge also to justify the current work like "those two studies quantified that in mouse loop duration is short-lived, therefore we investigated if this holds true also in human cells, in multiple loci".

2) The authors used the HCT116 line, which is a near-diploid cancer line. It would be useful to know if the authors checked the mutational burden on RAD21, NIPBL, WAPL, and CTCF. Do any of these genes or components of cohesin carry mutations?

3) The conclusion at line 94 are very strong since they also did not test all loops but only three (plus a control region) out of thousands. This part could be modified such as "given the results of other studies in mouse and ours, we can suggest that extrusion dynamics is conserved..." Saying that is universal is a big claim and should be done with caution. We know that for example in *Drosophila* can be different. Moreover, the three loci are all in compartment A. Therefore, saying that these results are universal (as well as stated in the title) can be an overclaim and this should affirmation should be tuned down, both in the text and in the title.

4) Fig. 1C provides contact maps for the L1, L2, and T1 loci but do not deliver information regarding gene composition, which are only showed in the extended data. I believe that for a better understanding, the gene tracks, pro-seq, and CTCF peaks should be represented also in main as polished version and not as screenshot from presumably from HiGlass, which makes visualization difficult. Also it seems there is an artifact in the plotting of L1 map since the diagonal is not extending for the whole length of the box and it appears truncated with respect on the underlying tracks.

5) In L1 and L2, 3 hours of auxin treatment removes most of the interaction, but it's possible to notice residual contact despite the efficiency of RAD21 depletion. Can the authors speculate on the cause of that residual contact?

6) Fig 1g, for clarity I would rename the "- Auxin" to "no auxin" as the minus could be confused with a dash.

7) Panel 1C is mentioned in the text before 1B, would it be better to invert the label?

8) Line 410: I have a potentially naïve question. The authors state that in the auxin-treated cells, the estimate of the simulated cohesin density of 2-fold. A decrease of 2-fold, to me, is surprisingly mild. I am not sure what I would have expected, but only 2-fold leads me to think that maybe this approach is too conservative. Can the author explain this part better?

9) In line 463, the authors state that they performed the first measurement of cohesin-dependent extrusion speed in living cells. This is only partially true and holds only for humans. In Gabriele et al they quantified cohesin processivity (processivity = lifetime × extrusion speed). With the calculated cohesin density of processivity of 150 kb and the base lifetime of ~20 minutes, it's easy to calculate the extrusion speed (as also indicated in their supplementary materials). Remarkably, $150\text{kb}/1200\text{s} = 0.125\text{kb/s}$, which is the very same as Sabate et al. best estimate. Similarly, also in Mach et al the loop extrusion was quantified from simulated data, ranging between 0.1kb/s to 1kb/s. Therefore, the authors here should rephrase their claim of "first measurement". However, this is an important piece of information because this paper, taken together with Gabriele et., strongly agree on the estimated extrusion speed. Taken together, they independently suggested that loop extrusion speed in vivo is generally the same in mice and humans. Despite this, I still believe that the term "universal" does not hold true because suggest that the same mechanism is indeed "universal". This term can be used with the genetic code, with exceptions, but in this context "conserved" is much more appropriate than universal.

10) Again, the statement in line 523 is not true and holds only as the first estimate in human cells.

11) Similarly, in discussion at line 572, claiming that these results are universal should be supported by experiments and simulation also in compartment B, lamina-associated domains, etc. In this paper, for the first time, they studied three loci instead of single loci as in Gabriele and Mach. However, still, these are only three loci and might not represent universally cohesin behavior outside compartment A.

12) The statement starting from line 583 "Although differences in Hi-C patterns between A and B compartments have been shown to originate from differential CTCF binding and not from differences in cohesin dynamics, it still remains important to determine parameters of cohesin dynamics in repressive chromatin contexts." Agrees with my concern about the use of

“universal” since these experiments are only done in compartment A.

13) Extended data Fig.3 panel A represent the image analysis pipeline. Given the fact that they performed also tetraspeck beads correction as stated in the method, I would suggest that this panel would feature both beads+actin.

14) In case the authors performed 3D genomics before and after removing the Bsd and Neo resistances, it would be interesting and useful to make these data available to the scientific community to have a quantification of the impact of having highly transcribed genes at the border of the TAD. Especially in light of the retention of resistance genes observed in a small fraction of the T1 line, as they state in the methods and the fact that the contact maps of T1 after cohesin depletion seems not to be affected by this small contamination.

In Summary, I recommend this paper for publication after minor but important revisions (included other comments/questions above):

- removal of the word “universal,” both from title and text which is an overstatement, as also declared by the authors in line 584. At most, it could be in the discussion as a hypothesis, underscoring limitations.
- correct the possible artifact of L1 map
- double-check label progression
- correct of the statement “first paper to measure in living cells loop extrusion speed” and discussion of Mach and Gabriele papers regarding speed extrusion calculation in vivo. In which in the first they use a range, and in the latter they have exactly 0.125kb/s outside the regions where CTCF boosts cohesin residency.

Version 1:

Decision Letter:

IMPORTANT: Please note the reference number: NG-A66287R-Z Sabaté. This number must be quoted whenever you communicate with us regarding this paper.

25th Mar 2025

Dear Dr. Sabaté,

Thank you for asking us to reconsider our decision on your manuscript "Universal dynamics of cohesin-mediated loop extrusion". I have now discussed the points of your letter with my colleagues, and we think that you have some valid points. We therefore invite you to revise your manuscript along the lines that you propose.

When preparing a revision, please ensure that it fully complies with our editorial requirements for format and style; details can be found in the Guide to Authors on our website (<http://www.nature.com/ng/>).

Please be sure that your manuscript is accompanied by a separate letter detailing the changes you have made and your response to the points raised. At this stage we will need you to upload:

1) a copy of the manuscript in MS Word .docx format.

2) The Editorial Policy Checklist:

<https://www.nature.com/documents/nr-editorial-policy-checklist.pdf>

3) The Reporting Summary:

(Here you can read about the role of the Reporting Summary in reproducible science:

<https://www.nature.com/news/announcement-towards-greater-reproducibility-for-life-sciences-research-in-nature-1.22062>)

Please use the link below to be taken directly to the site and view and revise your manuscript:

Link Redacted

With kind wishes,
Chiara

Chiara Anania, PhD

Associate Editor

Nature Genetics

<https://orcid.org/0000-0003-1549-4157>

Version 2:

Decision Letter:

12th May 2025

Dear Dr Sabaté,

Your Article, "Uniform dynamics of cohesin-mediated loop extrusion" has now been seen by 3 referees. You will see from their comments below that while they find your work of interest, some important points are raised. We are interested in the possibility of publishing your study in Nature Genetics, but would like to consider your response to these concerns in the form of a revised manuscript before we make a final decision on publication. Please do not hesitate to get in touch if you would like to discuss these issues further.

We therefore invite you to revise your manuscript taking into account all reviewer and editor comments. Please highlight all changes in the manuscript text file. At this stage we will need you to upload a copy of the manuscript in MS Word .docx or similar editable format.

*2) If you have not done so already please begin to revise your manuscript so that it conforms to our Article format instructions, available

[here](http://www.nature.com/ng/authors/article_types/index.html).

*3) Include a revised version of any required Reporting Summary: <https://www.nature.com/documents/nr-reporting-summary.pdf>

Please be aware of our [guidelines](https://www.nature.com/nature-research/editorial-policies/image-integrity) on digital image standards.

EXTENDED DATA FIGURES

Link Redacted

We hope to receive your revised manuscript within four to eight weeks. If you cannot send it within this time, please let us know.

Nature Genetics is committed to improving transparency in authorship. As part of our efforts in this direction, we are now

requesting that all authors identified as 'corresponding author' on published papers create and link their Open Researcher and Contributor Identifier (ORCID) with their account on the Manuscript Tracking System (MTS), prior to acceptance. ORCID helps the scientific community achieve unambiguous attribution of all scholarly contributions. You can create and link your ORCID from the home page of the MTS by clicking on 'Modify my Springer Nature account'. For more information please visit www.springernature.com/orcid.

Sincerely,
Chiara

Chiara Anania, PhD
Associate Editor
Nature Genetics
<https://orcid.org/0000-0003-1549-4157>

Referee expertise:

Referee #1:

Referee #2:

Referee #3: 3D genome, imaging

Reviewers' Comments:

Reviewer #1 (Remarks to the Author):

Sabate et al. have submitted a revised manuscript and a 46 page rebuttal! Many of the new data is only provided to reviewers, which is a missed opportunity to provide clarifying information.

The authors make two major conclusions as per their abstract: First, "that TADs are dynamic structures whose anchors are brought in proximity about once per hour and for 6-19 min (~16% of the time)". This conclusion is confirmatory of earlier studies in the field. Second, "TADs are continuously subjected to extrusion by multiple cohesin complexes". This would be a novel conclusion, but I remain unconvinced by the strength of the data to support this claim.

I maintain my point that the authors have not demonstrated that they can determine whether a loop is extruding or not or that they can distinguish extruding from closed loops. At no point do the authors measure loop extrusion and as such they can not deduce information about when loop extrusion occurs. The estimates are either purely based on simulations or use RAD21 degradation data. The authors assume that loss of RAD21 only affects extruding loops, but this assumption is not validated. Cohesin does not only bind at TAD anchors but also affects chromatin organization via TAD-independent mechanisms. In the absence of clear orthogonal validation of the ability to distinguish extruding and non-extruding states, the main conclusion regarding pervasiveness of extrusion is not convincingly supported.

Another major weakness of the study which persists is that the authors do not take advantage of their simulations to make non-trivial prediction then test them experimentally rather than to rely on simulations to draw correlative conclusions.

Overall, while this is an impressive tour-de-force of loop simulations, I am not convinced that the conclusions and the numerical values generated are correct.

Reviewer #2 (Remarks to the Author):

I am impressed by the amount of new insights Sabaté and colleagues collected to address Reviewers' comment. The paper has greatly improved. I believe that this paper is cutting-edge, and with these revision it is now ready for publication since they addressed all points.

Reviewer #2 (Remarks on code availability):

A last advice is to upload the code on zenodo as well, together with the microscopy revised data. The code is sufficiently documented.

Reviewer #3 (Remarks to the Author):

I have read the revised manuscript by Sabaté et al. with great interest, and found that the comments raised by both Reviewers helped the Authors to greatly improve its quality. Indeed, the Authors did a fantastic job replying to their comments. I find the work solid, and I appreciate that the Authors have tuned down their overstatements, which were pointed out by the Reviewers.

I agree with the Reviewers that the work is an extension (and a confirmation) of the two previous works, this time done in a human cell line and on 3 different loci. I do think that the fact that the previous findings were confirmed using a different approach is of importance (especially given the relevance of the topic) and given that I find the quality of the data high, I am in favour of accepting the work in its current form. In my mind the novelty is that the various parameters linked to the loop dynamics seem to remain pretty stable when we start adding additional DNA loci to those measurements (albeit with relatively broad ranges). And even though it might not be a major leap forward, given how tricky these measurements are, obtaining those values in as robust manner as the Authors achieved in this work seems to me very valuable.

My only concern are those rather broad ranges, which the Reviewer #1 is also critical about. The Authors argue against it, but in reality this work does not bring us significantly closer to more precise estimates. Perhaps the Authors could still discuss this issue more in the Discussion section (using some of the reasoning from the rebuttal letter).

Version 3:

Decision Letter:

Our ref: NG-A66287R2

10th Jul 2025

Dear Dr. Sabaté,

Thank you for submitting your revised manuscript "Uniform dynamics of cohesin-mediated loop extrusion" (NG-A66287R2). We find that the paper has improved in revision, and therefore we'll be happy in principle to publish it in Nature Genetics, pending minor revisions to comply with our editorial and formatting guidelines.

Congratulations!

Sincerely,
Chiara

Chiara Anania, PhD
Associate Editor
Nature Genetics
<https://orcid.org/0000-0003-1549-4157>

We sincerely thank both reviewers for their thorough and high-quality evaluation of our manuscript. Their comments motivated us to perform extensive new experiments and analyses, which confirm and significantly strengthen our original conclusions. We now submit a revised manuscript incorporating these additions that we believe fully address the reviewers' concerns. Below, we summarize the key improvements:

- **High frequency live-cell imaging data:** To address concerns about the resolution of our methods and the confidence in our measurements, we acquired new live-cell imaging data at 3-fold higher temporal resolution. The analysis of this new dataset yielded highly similar results and confirmed our main findings, including the key finding that TADs are constantly undergoing extrusion. See R1.2.c.
- **Stringent filtering of the original live-cell imaging data:** To address concerns about the effect of measurement noise and limited spatial resolution, we analyzed a subset of our live-cell imaging data with significantly more stringent quality control filters (e.g. higher spatial resolution and less tracking errors). The analysis of this new dataset again confirmed and reinforced our original measurements and findings. See R1.2.b. Together with the high frequency imaging, these results demonstrate that our findings are robust to variations in both spatial and temporal resolution, and greatly strengthen our conclusions.
- **Validation of the three-state model:** We show that our three-state model accurately captures the different loop configurations and demonstrate that polymer relaxation occurs on timescales faster than our imaging frequency, thereby not affecting our conclusions. See R1.1.b and R1.1.c.
- **TADs are in constant extrusion:** We provide an additional and new model-free analysis (based on comparing experimental distance distributions with and without cohesin depletion) demonstrating that open states are rare or absent. This analysis, which does not require any assumptions about loop states, further strengthens our conclusion that TADs are constantly undergoing extrusion. See R1.1.f and Appendix A5.
- **Behavior of the No TAD control:** This control exhibits one fluorescent reporter located away from CTCF sites, whereas the second reporter is located at a CTCF border. We believe that 'No TAD' was actually a misnomer and therefore renamed it 'Half TAD'. Importantly, we demonstrate with polymer simulations that its observed behavior in living cells is in fact expected, confirming that the Half TAD serves as a valid control. See R1.3.
- **Capture Micro-C of untagged cells:** We performed new Capture Micro-C experiments comparing untagged and tagged cells, confirming that the insertion of repeat arrays does not affect chromatin looping. See R1.5.
- **Characterization of proximal and closed states:** We clarified the distinction between closed and proximal states and quantitatively characterized their relationship. See R1.1.a and Appendix A1.
- **Claim of universality:** We rephrased the manuscript to tone down the statement of 'universality' and replaced 'universal' by 'uniform'. See R2.3.

We explain these improvements and respond to each comment in detail below.

We believe that our manuscript has been considerably improved thanks to the reviewers' insightful suggestions and hope that you will find it suitable for publication in *Nature Genetics*.

Best regards,
The authors

Reviewer #1:

Remarks to the Author:

Sabate et al. report on live-cell dynamics of TADs and use computer simulations to extract quantitative information on various aspects of TAD dynamics. They fluorescently tag three TADs anchor regions in HCT116 colon cancer cells and confirm by timelapse imaging earlier observations demonstrating that TADs anchor distances fluctuate and that TAD anchors are only paired transiently. They define three anchor pairing states (open, extruding, closed) and using this approach they then determine various properties of anchor sites, including the rate at which anchors are brought together and extrusion rate. A major conclusion is that most TADs are being extruded at any time. Computer simulations are then used to probe the effect of residence times, extrusion rates etc. on TADs and to compare simulations to experimental data. Using this approach, the in vivo extrusion speed is determined to be ~ 0.1 kb/s.

These experiments are similar to two recent studies which analyzed the live-cell dynamics of TAD boundaries. Many of reported results are in line with the published studies, including the observation of generally dynamic TAD behavior and low frequency of anchor pairing. This study extends the earlier ones in that it analyzes three instead of only one TAD, does so in human non-stem cells, and uses more elaborate simulation approaches to extract additional quantitative information on cohesion loading frequency etc.

Although this is a technically cutting-edge approach and quantitative information about TAD dynamics are of interest, there is concern that the methods are not quite sensitive enough to draw some of the conclusions presented and the validity of the deduced numbers is difficult to determine. The limited resolution of imaging for detection of the very small distance fluctuations the TAD anchors undergo combined with the use of various necessary assumptions built into computational models reduce the precision – and confidence – in the reported quantitative measurements.

R1.0: We thank the reviewer for the very thoughtful evaluation of our work. In the following, we address each specific comment in detail. We apologize for the length of some of our responses, but we believe it is essential to fully address all points in detail.

In summary, our responses will:

- Clarify that most of our findings do not rely on the assumption of 3 loop states questioned by the reviewer (see R1.1.a).

- Show that the model of 3 loop states accurately describes loop configurations, and that our conclusion that TADs are constantly undergoing extrusion is a robust finding of our study, independent of the 3-state model (see R1.1.b.c.f).
- Show that distance fluctuations observed in our imaging data are dominated by biophysical motion of the chromatin polymer rather than measurement noise, indicating that our imaging approach has sufficient sensitivity and resolution to support our conclusions (see R1.2.a).
- Show that our conclusions are unchanged when using experimental data with reduced levels of measurement noise (see R1.2.b).
- Provide new live-cell imaging data at a higher sampling frequency, demonstrating that our conclusions are also robust to temporal resolution (see R1.2.c).

Please note that to facilitate the reading of the rebuttal, we occasionally refer to the Appendix at the end of the document where more technical arguments are detailed. Overall, we believe that these new experimental data and analyses address the reviewer's concerns and provide strong additional support to our main conclusions.

1) the analysis and interpretation are based on the central assumption stated on p. 8 that anchor pairs exist in three states (open, extruding, closed). It is not clear that this is a correct assumption for the analysis. Particularly, it is questionable that all anchor pairs which are not paired or are not fully open must be extruding. For example, after extrusion is complete and cohesin dissociates, anchors may separate as the domain relaxes into an open state. This non-extruding state would be mis-assigned in this analysis as loop extrusion. Similarly, as extrusion commences, anchors are separate but may be extruding (they would be mis-classified as open and not extruding). The authors also assume that a closed state can only be maintained by bound cohesin; I am not sure there is any good evidence to demonstrate that assumption. As is, it is not clear how accurately active loop extrusion can be ascertained from these proximity measurements, thus strongly confounding the interpretation of the data. Unless the authors can show that there is a strict relationship between distance and actual extrusion activity, the three-state model should probably not be used to extract information about extrusion dynamics. As a consequence, several key conclusions, including the most interesting finding that most TADs undergo constant extrusion, are not convincingly supported. The inaccuracy of the state assignments may also explain the relatively large ranges of values the authors obtain for various parameters (see below).

R1.1: In this comment the reviewer expresses several concerns:

- a) The concern that our main findings are based on the assumption of 3 loop states (open, extruding, closed)
- b) The concern that polymer relaxation following cohesin dissociation is not taken into account in the 3-state model, potentially leading to errors in the estimation of extruding states
- c) The concern that some early extruding states may be misclassified as open states
- d) The concern that we assumed closed states to be necessarily maintained by bound cohesin
- e) The need to show a strict relationship between distance and extrusion activity
- f) The concern that our finding that TADs undergo constant extrusion is insufficiently supported

g) The concern that the large range of values estimated for some parameters may result from incorrect state assignments

We address these concerns one by one below.

R1.1.a: The concern that our main findings are based on the assumption of 3 loop states (open, closed, extruding)

We would like to clarify that our main conclusions are in fact *not* based on an assumption of three states. We believe that the reviewer's concern is due to a potentially misleading description of our analyses in the submitted manuscript, for which we apologize. Although we indeed considered (as schematized in Fig. 1b) that a pair of anchors is theoretically either in a closed, extruding or open state at any time, we did *not* assign each time point of our distance time series to one of these three states in subsequent analyses. Instead, most of our analyses distinguish solely between 'proximal' and 'non-proximal' states, which are segmented in distance time series. (Note that proximal states are related to, but not identical to closed states, as detailed in Appendix A1, now included as the new Extended Data Fig. 10a-f). Thus, our estimates of proximal state fractions, frequencies and lifetimes (Fig. 2b-d), our estimates of closing rates (Fig. 3), and cohesin densities, residence times and extrusion speed by comparing simulations to experiments (Fig. 5) are all independent of the 3-state assumption. The only analysis relying on this assumption is the estimation of loop state fractions in Fig. 2e and Extended Data Fig. 5, where we fit an analytical mixture model of distances with 3 component distributions (proximal, extruding and open), instead of segmenting distance time series into proximal states. Below, we address in detail concerns about the 3-state analysis and its conclusion that TADs are continuously extruded (R1.1.b, R1.1.c and R1.1.f). Importantly, we included a novel, model-free estimation of the maximum fraction of open states (Appendix A5), which confirms that open states are quasi absent, without modeling assumptions.

We apologize for the lack of clarity about the 3-state model in the original text and have now clarified the difference between our analyses in the Results sections 'TADs are dynamic structures' and 'TADs are constantly extruded'.

R1.1.b: The concern that polymer relaxation following cohesin dissociation is not taken into account in the 3-state model, potentially leading to errors in the estimation of extruding states

In principle, we agree with the reviewer that relaxation of the polymer after cohesin dissociation could potentially affect the estimations of extruding state fractions. However, polymer simulations indicate that this is unlikely to affect our conclusions for two reasons: (i) relaxation following a closed state is rare (it occurs after only 8% of closed states) and (ii) relaxation is very fast (~1 min, equaling 2 imaging frames), hence barely captured at our imaging frequency. These findings are detailed in Appendix A2, now included as the new Extended Data Fig. 5c-e.

R1.1.c: The concern that some early extruding states may be misclassified as open states

First, we recall that, in untreated cells, our estimates of open state fractions are 0% for both L1 and L2 and only 11% for T1 (Fig. 2e). Thus, a potential overestimation of open states is not relevant for L1 and L2 and would represent, at worst, 11% for the T1 TAD.

Second, we would like to reiterate that our 3-state model does not assign individual timepoints to loop states. Instead, we fit an analytical model to entire distributions of anchor-anchor coordinate differences to determine the statistical proportions of the three states in the overall distribution

(see Methods section ‘Estimation of loop state fractions’). Consequently, there is no ‘classification’ of single timepoints into one of the three loop states (proximal, extruding or open). Third, using polymer simulations we show in Appendix A3 (now included as the new Extended Data Fig. 5f) that open states are accurately estimated by the analytical model for the large majority of parameter combinations, and especially those consistent with experiments.

R1.1.d: The concern that we assumed closed states to be necessarily maintained by bound cohesin

Cohesin depletion studies, including our own (Fig. 1c), have demonstrated that the cohesin complex is the major factor responsible for closed states¹⁻⁷. However, closed states are distinct from proximal states (Appendix A1), and while we acknowledge that other mechanisms can potentially contribute to proximal states, cohesin-mediated loop extrusion remains the main contributor (since cohesin removal largely decreases proximal state fraction and frequency by ~70%; Fig. 2b,c). Considering that assuming closed states to be necessarily maintained by cohesin is not essential, and does not impact our downstream analyses, we rephrased this part and removed the reference to cohesin.

R1.1.e: The need to show a strict relationship between distance and extrusion activity

Confidently assigning individual timepoints to a loop state (e.g. open, extruding or closed) based only on a single anchor-anchor distance is not possible, because distance distributions overlap very significantly between loop states, as shown by experiments with or without auxin (Fig. 1g and Extended Data Fig. 3g) and simulations (Fig. 4e). However, the complete distance distribution or the temporal evolution of distances provide the necessary information to distinguish between loop states. For example, while distances in the open state can occasionally be smaller than in the extruding state, such short distances are not sustained over time in the open state (Figs. 1f and 4e). Extrusion activity is not strictly related to static single distances but is instead captured in the temporal variations of distances, underscoring the importance of dynamic data to quantify loop extrusion dynamics. In Appendix A4, we recall multiple independent lines of evidence showing that the loop extrusion process affects the overall distribution and temporal dynamics of 3D distances.

R1.1.f: The concern that our finding that TADs undergo constant extrusion is insufficiently supported

Our finding that TADs are almost never in the open state and constantly undergo extrusion is indeed a main finding of our paper. We summarize below the 4 distinct analyses supporting this conclusion, highlighting that 2 depend on the 3-state model, but 2 others do not.

(i) Analyses based on the 3-state model:

- a. As indicated above, we have validated the 3-state model with both experimental and simulated data (R1.1.c and Extended Data Fig. 5b), and demonstrated that polymer relaxation does not affect our results (R1.1.b). Moreover, we identify proximal states as proxies for closed states, but proximal states are generally longer than closed states (Appendix A1). This leads to an overestimation of proximal states at the expense of extruding states, which are thus underestimated (Appendix A3). Consequently, the case that TADs are constantly extruded can only be stronger.

- b. We show that live-cell imaging datasets with lower measurement noise and acquired at higher frequency also support this conclusion (see R1.2.b,c below).
- (ii) Analyses independent of the 3-state model:
- a. Polymer simulations show that TADs are found nearly 100% of the time in the extruding state (Fig. 7a). Importantly, this conclusion is independent of the 3-state model, since the estimation of simulated parameters (Fig. 5) does not rely on this model.
 - b. Finally, we have obtained a new estimate for the maximal fraction of open state, that only relies on experimentally measured distance distributions, and does not rely on any model assumptions. By fitting the distance distribution measured in auxin-treated cells beneath the distance distribution of untreated cells, we show that the latter can accommodate at most 18%, 4%, and 5% of open states for L1, L2, and T1, respectively (Appendix A5). This directly shows that open states are rare or nearly absent in untreated cells, without requiring any further assumptions.

Thus, our finding of constant extrusion is supported by four analyses, involving experimental and simulated data, two of which do not rely on the 3-state assumption and one is entirely model-free. This conclusion is thus a robust result of our study.

R1.1.g: The concern that the large range of values estimated for some parameters may result from incorrect state assignments

The large range of estimated parameter values mainly concerns cohesin residence time (Fig. 5). We again would like to clarify that the 3-state model was not used to estimate these parameters. Instead, we relied only on the distinction between proximal and non-proximal states obtained by segmenting time series, together with model-free metrics such as distance distributions, two-point MSD and $P(s)$. We show below that inaccurate state assignment is not the origin of the large ranges of values estimated (see R1.4).

2) even accepting the three-state model, the interpretation of the data relies heavily on accurate calling of paired or unpaired anchors. It is not evident from the presented data how accurately these states can be called from the rather noisy data. The data shown in figure 1g suggests quite small distance differences between paired and unpaired anchors and the traces are quite noisy (as expected for these types of measurements). This concern is highlighted in Ext. Fig 8, where large spikes in distances in the adjacent control appear to be called proximal, whereas several “flat” regions in L1, L2 and L3 are called non-proximal. Similarly, the authors only find 44% of anchor pairs in proximity in their adjacent TAD control which consist of two labelled sites only 6kb apart. Why is this pairing rate not much higher? It is not clear how accurately these states can be called using the imaging methods used and how the calling of the states was validated other than by simulation.

R1.2: We understand that the reviewer is primarily concerned here that our calling of proximal states may be inaccurate because of noise in our imaging measurements, and that this may lead to inaccurate quantitative estimates and interpretations (including the fraction of proximal states, their frequency and lifetime).

We agree that distance time series are subject to measurement noise, such as random localization errors (as we had mentioned in the original article), and recognize that it is important

to evaluate its impact on our quantitative estimates. To address this concern directly, we provide new experimental data and new analyses (detailed below):

- a) We quantitatively compare the distance fluctuations emerging from measurement noise to the distance fluctuations due to biophysical motion of the chromatin fiber (see R1.2.a)
- b) We reanalyze a subset of the data with lower levels of measurement noise, and compare the results to those obtained with the original dataset (see R1.2.b)
- c) We performed new live-cell imaging experiments sampled at higher frequency, and compare their results to those of the original dataset (see R1.2.c)

We reasoned that if our analyses were significantly affected by measurement noise or limited by spatio-temporal resolution, the new datasets in (b) and (c) would yield different conclusions as compared to the original dataset. However, as we show in R1.2.b,c, the results from these new analyses are highly consistent with our initial findings, demonstrating that the precision of our distance measurements and the temporal resolution of our time series are sufficient to accurately quantify TAD dynamics.

We also address below the other concerns of the reviewer:

- d) Some large 'spikes' in distances in the Adjacent locus are occasionally assigned to proximal states whereas some "flat" distance regions in TADs are assigned to non-proximal states (see R1.2.d)
- e) The Adjacent locus with two labels separated by 6 kb has only 44% of proximal states (see R1.2.e)
- f) Our calling of proximal states was validated only by simulations (see R1.2.f)

R1.2.a: Distance fluctuations arise from chromatin motion rather than measurement noise

We stress that apparently random fluctuations in distance time series can originate either from measurement noise or from the actual stochastic motion of the chromatin polymer. A major challenge of our study (and prior related studies) is to disentangle the biophysical motion of the chromatin fiber (with or without extrusion) from the fluctuations due to measurement noise^{8,9}. Several sources of measurement noise affect our 3D distance measurements: (i) random localization errors of fluorescent spots, (ii) photobleaching, which increases localization errors over time, (iii) the genomic separation between fluorescent reporters and CTCF anchors. In our study, we estimated the localization precision of each fluorescent spot at each time point (see 'Refining localizations and measuring localization precision' section in Methods). This allowed us to synthetically incorporate localization errors consistent with experiments (and increasing with time due to photobleaching) into simulated time series (see 'Comparison of simulations to experiments' section in Methods), thereby mimicking experimental noise. Furthermore, these polymer simulations also took into account the genomic separation between TAD anchors and the reporters.

Comparing simulations with or without measurement noise to the experimental data allows us to estimate the relative contribution of noise to biological fluctuations of distances. Visual inspection of time series reveals that fluctuations in distances are present in both experiments and simulations (Fig. F1a). Specifically, fast fluctuations (~1 min), as well as sudden peaks in distances are observed even in simulations without measurement noise (Fig. F1a). Furthermore, simulated time series with or without measurement noise exhibit only minor differences with experimental ones, even at the end of time series where photobleaching is maximal (Fig. F1a). These two observations provide initial evidence that the fluctuations observed in our experiments are primarily due to chromatin motion rather than measurement noise.

To quantitatively assess the contribution of measurement noise to the observed distance fluctuations, we computed the Root Mean Squared Error (RMSE) between raw distances and rolling-averaged time series, where fast fluctuations are smoothed out (Fig. F1b). Comparing the

RMSE of experimental time series to simulated ones without measurement noise revealed that 74-91% of fluctuations could be explained by biophysical motion of the chromatin fiber alone, while measurement noise accounted for only 9-14% of these fluctuations (Fig. F1c). The remaining 0-17% may stem from other types of errors specific to experiments, not accounted for in our simulations, such as tracking errors.

In summary, the distance fluctuations observed in both simulated and experimental time series largely arise from the biophysical motion of the chromatin fiber rather than from measurement noise. This indicates that our imaging approach has sufficient resolution to describe the biological dynamics of TADs and that measurement noise does not obscure our data.

We added this new analysis as the new Extended Data Fig. 6a-c and in the new Results section ‘Quantifications of TAD dynamics are robust to spatial and temporal resolutions’.

Fig. F1: Fluctuations in distance time series primarily arise from biophysical chromatin motion rather than from measurement noise.

a, Left: Simulated time series with (purple) or without (green) random localization errors, photobleaching and reporter-anchor separation. Right: Experimental distance time series. **b**, Raw (colored) and rolling averaged (black) distance time series from experiments (red) or simulations with (purple) or without (green) measurement noise. The Root Mean Squared Error (RMSE) is computed between the raw time series and a rolling average with a time window of 5 min. **c**, RMSE computed between raw and rolling averaged time series for experimental data (red), polymer simulations without (green) and with (purple) measurement noise. The percentage of the experimental RMSE is indicated in the green and purple bars. For experimental and simulated time series, we used the L2 TAD as an example in **a, b**. Polymer simulations assumed a cohesin density of 12 Mb^{-1} , a residence time of 22 min and a motor speed of 0.25 kb/s.

R1.2.b: The spatial resolution of live-cell imaging is sufficient to accurately quantify TAD dynamics

We reasoned that if our quantifications of TAD dynamics were strongly affected by measurement errors, then our estimates would change substantially when analyzing data with less measurement noise and fewer errors.

To this end, we applied more stringent quality filters to raw localizations (Fig. F2a,b), which eliminated preferentially timepoints towards the end of time series, where photobleaching effects are more pronounced and fluorescent spot localization is more error-prone (Fig. F2d,e). While our original quality filter removed 8-26% of raw timepoints, this more stringent filtering removed 22-70% of them, *i.e.* ~3 times more (Fig. F2a,b). We then analyzed this data subset using the same methods as in the original dataset. We show that the estimated fractions, frequencies, and lifetimes of proximal states are essentially the same between the original and the stringent datasets (Fig. F2a,b). Across all cell lines, treatments and for proximal state fraction, frequency and lifetime, the estimates remained within 0.7- to 1.2-fold of the initial estimates (Fig. F2a,b). Importantly, these new estimates remained within the ranges reported in the article, obtained with the original dataset (Extended Data Fig. 4f-h), further reinforcing our findings. Similarly, the absolute difference in the proportions of loop states did not differ by more than 8% from the initial estimate, while closing rates changed by only 0.9-1.4-fold in the stringent dataset (Fig. F2a,b). Thus, our conclusions that TADs are dynamic structures and that anchor-anchor contacts are transient are robust to measurement noise. This also adds support to our first conclusion above (R1.2.a) that measurement noise does not dominate the observed distance fluctuations.

We added these new analyses to a new Results section entitled ‘Quantifications of TAD dynamics are robust to spatial and temporal resolutions’ and as the new Extended Data Fig. 6d,e,g,h.

R1.2.c: The temporal resolution of live-cell imaging is sufficient to accurately quantify TAD dynamics

Our quantifications of TAD dynamics may be limited by the temporal sampling at which we acquire microscopy images. To address this, we acquired a new and large experimental imaging dataset at a frequency of 1 image per 9 s, more than three times higher than the original frequency of 1 image per 30 s (Fig. F2c,f). We acquired images with and without auxin treatment for the L1 and Half TAD (formerly named ‘No TAD’), and also for the untreated Adjacent locus, resulting in five novel experimental datasets (summarized in the updated Supplementary Table 1).

Proximal state fraction, frequency and lifetime estimated from this high frequency dataset closely matched those from the original dataset, regardless of the temporal threshold used for proximal state segmentation (Fig. F2a,c). For instance, we found proximal state fractions of 14% vs 16% for L2 and 9% vs 8% for the Half TAD in the high frequency and original datasets, respectively (Fig. 2a,c). This was also true for the fraction of loop states (Fig. F2a,c), thus further supporting our conclusion that TADs are constantly undergoing extrusion. As a result, these new live-cell microscopy time lapses demonstrate that our conclusions are also robust to temporal resolution.

We now included these new data and analyses in the new Extended Data Fig. 6f,i and in the new Results section ‘Quantifications of TAD dynamics are robust to spatial and temporal resolutions’.

In conclusion, our quantifications of TAD dynamics, along with our findings that proximal states are transient and that TADs are most often in an extruding state are firmly supported by multiple analyses. These results remain robust to both measurement noise and temporal resolution. Additionally, we demonstrated that the observed fluctuations in distances are primarily driven by the biophysical motion of the chromatin fiber, rather than by measurement noise.

Fig. F2: Quantifications of TAD dynamics are robust to measurement noise and imaging frequency. **a-c**, Comparison of results obtained with the original dataset (used throughout the article, **a**), the stringent dataset obtained after stringent quality filtering (**b**), and a dataset acquired at higher temporal frequency (one image per 9 s instead of 30 s, **c**). Blue boxes show criteria used for quality filtering of raw time series. The min-max percentage of raw timepoints eliminated across all cell lines and treatments is indicated. We could not estimate the closing rate in the high frequency dataset because of an insufficient number of identified proximal states. Note that the estimated parameters are very similar across the three datasets. **d**, Histogram of 3D anchor-anchor distance from the Adjacent locus in the raw, original and stringent datasets. **e**, Distance time series from the Adjacent locus in the raw, original and stringent datasets. **f**, Distance time series acquired at an imaging frequency of 1 image per 9 s (red), and interpolated and undersampled at the original frequency of 1 image per 30 s (black). Dark red and black bars indicate segmented proximal states using the high frequency or undersampled time series, respectively.

R1.2.d: The concern that some large “spikes” in distances in the Adjacent locus are occasionally assigned to proximal states whereas some “flat” distance regions in TADs are assigned to non-proximal states

Please note that the former Extended Data Fig. 8 has been relabeled as Extended Data Fig. 4a. Our segmentation method assumes that a proximal state should exhibit distances below a spatial threshold for a duration at least equal to the temporal threshold. The spatial threshold is defined as the distance below which we can expect to find 95% of closed states, taking into account localization errors and other sources of measurement noise (see Methods section ‘Proximal state segmentation’). Therefore, larger localization errors raise the spatial threshold used to segment proximal states. Because the Adjacent locus exhibits the highest localization errors compared to other cell lines (Supplementary Tables 1 and 2), it is expected that proximal states can be associated with larger distances in this cell line.

The temporal threshold aims to prevent calling proximal states when anchors temporarily get closer solely because of random motion of the chromatin fiber⁸, as small distances can occur regardless of the state of the loop (R1.1.e). In practice, we apply a rolling average to the initial binary classifications obtained after thresholding (see Methods section ‘Proximal state segmentation’) in order to prevent fragmentation of proximal states due to occasional large distances that may result from tracking errors. This procedure can result in the merging of two adjacent proximal states separated by a short ‘spike’ in distance, which explains the occasional segmentation of distance spikes as proximal states.

Conversely, distances occasionally exceeding the spatial threshold (even just slightly) in ‘flat’ regions prevent the segmentation of proximal states. Note that L1, L2 and T1 are associated with lower spatial thresholds because of their better localization precision (Supplementary Tables 1 and 2). Smaller distances are thus required to call proximal states, as compared to the Adjacent locus.

These effects are an expected outcome of our segmentation approach, which takes into account experimental noise to ensure consistent segmentation of proximal states across cell lines with different levels of measurement noise.

R1.2.e: The concern that the Adjacent locus with two labels separated by 6 kb has only 44% of proximal states

We acknowledge that a positive control for closed states should ideally yield a higher percentage of proximal states. However, designing a suitable positive control is challenging, since no experimental method currently exists for trapping loops in their closed state in living cells. We designed the Adjacent locus only as a qualitative proxy for closed states (as mentioned in the Results section ‘Visualizing the dynamics of TAD anchors’), not as a perfect positive control.

Taking into account the 6 kb of DNA between the two reporters, the semi-flexible polymer behavior, localization errors and photobleaching, we expect median distances of 120 nm for the Adjacent locus, smaller than the 178 nm observed experimentally (Fig. F3a). A possible explanation for this discrepancy may be the stiffening of the chromatin polymer due to the high density of protein bound to chromatin at the repeat array location, potentially hindering nucleosome deposition¹⁰. To test this hypothesis, we varied the chromatin persistence length (a measure of chromatin stiffness) assumed in our theoretical polymer model of distances in the closed state (see Method section ‘Proximal state segmentation’). We found that increasing the persistence length from 40 to 100 nm could explain the observed experimental distance distribution of the Adjacent locus (Fig. F3b).

Although the Adjacent locus does not constitute a perfect positive control of closed states, it still serves as a valid qualitative control, since it exhibits the highest fraction, frequency and lifetime of proximal states compared to all other loci (Fig. 2b-d).

Notably, our observations align with those of Mach *et al*¹¹ and Gabriele *et al*¹², who also reported distances higher than expected for control loci consisting of perfectly colocalizing spots and spots separated by 10 kb, respectively.

We modified the manuscript and now mention: “Distances measured in the Adjacent locus were the smallest (Fig. 1g), but slightly larger than expected, possibly due to increased chromatin stiffness at repeat arrays (Extended Data Fig. 4b,c), and as previously observed for adjacent pairs in locus tracking studies^{26,27}” and included Fig. F3a,b as the new Extended Data Fig. 4b,c.

Fig. F3: The Adjacent control exhibits larger than expected reporter-reporter distances, possibly due to increased chromatin stiffness.

a, Distribution of distances of the Adjacent locus from experiments (grey) and expected from polymer theory (black) or polymer simulations (green). **b**, Experimental (grey) and theoretical (colored) distance distributions in the Adjacent locus for persistence lengths L_p ranging from 20 to 200 nm, in accordance with prior determinations of 16-134 nm^{13–17}. Rigidity (persistence length) increases from blue to yellow.

R1.2.f. The concern that our calling of proximal states was validated only by simulations

In response to this comment, we would like to emphasize that we included a total of six biological controls to validate the calling of proximal states, namely:

- (i) The Adjacent locus that we discussed above (R1.2.e), which exhibits the largest fraction, frequency and lifetime of proximal states (Fig. 2b-d)
- (ii) The ‘Half TAD’ region (formerly called ‘No TAD’), which exhibits the smallest fraction and frequency of proximal states compared to TAD regions (Fig. 2b,c, see R1.3)
- (iii) Cohesin depletion for all four cell lines L1, L2, T1 and Half TAD, which eliminated the majority of proximal states (~70% decrease in the fraction and frequency of proximal states, as compared to untreated cells; Fig. 2b,c).

In addition to these experimental controls, and because it is currently not experimentally feasible to obtain a pure population of anchors trapped together in a closed state, we have indeed characterized proximal states on polymer simulations (Appendix A1).

Finally, we would like to stress that we varied the temporal threshold used for calling proximal states and reported the resulting ranges of proximal state fractions, frequencies and lifetimes (Extended Data Fig. 4f-h), rather than relying on a single value, thereby minimizing the dependence of our findings on the proximal segmentation approach. Importantly, the full range of quantitative estimates is consistent with the main conclusion that proximal states are transient.

In summary, we have shown that our quantifications of TAD dynamics, and our conclusions that proximal states are transient and that TADs undergo constant extrusion, are robust to both spatial and temporal resolution, to variations in segmentation thresholds, and are validated by both polymer simulations and multiple experimental controls.

3) the non-TAD control behaved in several respects similar to the TAD anchor sites. For example, there was no apparent difference in lifetime of TAD anchors vs the non-TAD anchor and the fraction of proximal anchors was only about 2-fold lower for non-TAD anchors compared to TAD anchors. Similarly, the fraction of proximal pairs was reduced by ~ 3-fold both for TADs and the non-TAD control, with no apparent preference for the TAD anchors. It was overall not clear to me whether the authors argue for distinct behavior of TADs vs non-TADs anchors or whether they interpret their findings as similar behavior of TADs and non-TADs. If the authors want to claim distinct behavior, please provide statistical analysis of these measurements comparing TAD vs non-TAD anchors. If the authors, on the other hand, want to argue that there is no difference in TAD vs non-TAD anchors, this conclusion needs to be strengthened. It is currently made based on analysis of a single non-TAD region, making it very tenuous. Additional non-TAD regions should be analyzed to put it on firmer grounds.

R1.3: The reviewer expresses concern that the control cell line we had called “No TAD” does not behave as expected but behaves in several ways similarly to TADs.

We appreciate the reviewer’s comments, but we believe that they could result from a poor terminology on our part. Indeed, this cell line consists of a 5’ fluorescent reporter far from a CTCF site, whereas the 3’ reporter labels a CTCF site defining the 3’ border of the L1 TAD (Extended Data Fig. 1a). Because this cell line still involves a fluorescently labeled TAD anchor, “No TAD” was arguably a misnomer and we now refer to this locus as “Half TAD” instead (here and in the manuscript). In the following, we explain why the behavior of the Half TAD control is, in fact, consistent with expectations.

The Half TAD reporters are not expected to form a closed state since only one CTCF site, rather than two, is labelled. However, during extrusion, cohesin complexes can stop at the 3’ CTCF anchor while they continue to extrude DNA on the other side, thereby temporarily bringing the 5’ reporter in close vicinity to the 3’ reporter (Fig. F4a). This allows the formation of proximal states, which are thus expected to occur in the Half TAD cell line, albeit at a lower rate than in TAD regions. Please refer to Appendix A1 for the distinction between closed and proximal states.

In agreement with this expectation, we found that both the proximal state fraction and frequency in the Half TAD were significantly reduced, halving the levels observed in untreated cells from TADs L1, L2 and T1 (Fig. F4b,c). This difference is predicted to be even slightly lower by polymer simulations, where we found a 1.0-1.4-fold difference in proximal fraction and frequency between TADs (L1, L2, T1) and the Half TAD (Fig. F4d,e). For proximal state lifetimes, experiments indeed revealed similar values between TADs and non-TADs, with only 1.1-1.3-fold difference (Fig. F4f). However, this seemingly counterintuitive result was also predicted by polymer simulations, which estimated lifetimes of 4.9 min in the Half TAD and 4.8-5.2 min in TADs (1.0-1.1-fold difference; Fig. F4f), closely matching experimental observations.

Furthermore, decreasing measurement noise or increasing the imaging frequency yielded very similar fractions, frequencies, and lifetimes of proximal states, further supporting our estimates on this cell line (Fig. F2a-c). Thus, the behavior of the Half TAD locus is in fact consistent with expectations and is also predicted by polymer simulations, confirming that this locus serves as a valid control.

Please note that apart from the Half TAD, we performed five additional controls, as summarized above (R1.2.f). In light of this comprehensive set of controls for the three TADs that we study here, we do not find it necessary to analyze additional non-TAD regions.

Fig. F4: Quantifications of proximal states in the Half TAD region are as expected.

a, Schematized explanation of how proximal states are expected in the Half TAD cell line. The distance observed at each timepoint is schematized below, together with the potential segmented proximal state (purple). **b,c**, Same as Fig. 2b,c but with an additional statistical analysis comparing Half TAD to the three other TADs. ***: P-value < 0.001 from a post hoc Dunn's test following a Kruskal-Wallis test and using Bonferroni correction for multiple test comparison. N=10,000 bootstrap samples. **d-f**, Proximal state fraction (**d**), frequency (**e**) and lifetime (**f**) computed on polymer simulations (left) or experimental data (right). A cohesin density of 12 Mb^{-1} , a residence time of 22 min and a motor speed of 0.25 kb/s were assumed in simulations. N=450 simulated time series were used for estimations, matching experimental sampling size (Supplementary Table 1).

4) While the computer simulations are useful, the ranges of many of the estimates seem very large to the point of not being very meaningful; for example, the cohesin residence time is estimated to be between 3-33min and cohesin loading rates ranging from 0.2-2.8. These are order of magnitude estimates and as such not overly useful. If these ranges represent technical limitations of the approach, it lessens the impact of the methods used here; if they reflect biological variability, this should be shown more directly.

R1.4: The reviewer raised concerns about the wide range of cohesin parameter values inferred from comparing simulations to experiments and wonders whether these ranges reflect technical limitations or biological variability.

First, we agree that it is important to clarify if the parameter ranges reflect technical limitations or biological variability. To address this, we tested whether our parameter estimation method could retrieve the correct parameters from simulations with perfectly known cohesin dynamics parameters. We compared these simulations to those obtained with varying parameter combinations, as we did for experimental data. These simulations included measurement noise consistent with experiments but lacked biological variability, allowing us to describe uncertainties due solely to methodological limitations. Critically, our method successfully recovered the correct cohesin density, for both high and low cohesin density (12 and 2 Mb⁻¹), and for cohesin residence times above 5 min (Fig. F5). Given that previous experimental assessments of cohesin residence times ranged from 5 to 24 min^{2,18-21}, our method allows accurate estimation of cohesin density at biologically relevant scales. However, we could not precisely determine the residence time from these simulated data, as we obtained a broad range of values (Fig. F5). Nevertheless, this range was still reliable since it included the ground truth value. Therefore, the broad range of residence times estimated from experiments does not necessarily reflect biological variability, but rather a limitation of the method itself.

This analysis was included as the new Extended Data Fig. 9b and we added the following statement in the Results section 'TAD dynamics is consistent with uniform cohesin dynamics and is predicted by CTCF binding': "*We could successfully identify the correct cohesin density at both high and low densities (Extended Data Fig. 9b), for residence times longer than 5 min, which corresponds to the range of experimental estimates of 5-24 min^{35,56,58,60,61}. Although a broad range of cohesin residence times were consistent with the simulations, this range correctly encompassed the ground truth value (Extended Data Fig. 9b). This indicated that our approach could accurately estimate cohesin density and provide a broad but reliable range for residence times.*"

Fig. F5: The parameter estimation method provides an accurate value of cohesin density and a reliable range of residence times.

Estimation of cohesin dynamics parameters on experimental data (left, same as Fig. 5a) and simulated data of known cohesin density (center: 12 Mb⁻¹; right: 2 Mb⁻¹) and residence time (22 min in both cases). Solid and dashed grey lines indicate the 10% and 25% best parameter sets, respectively. Red rectangles highlight the ground truth parameters, also indicated above each map. Simulations assumed a motor speed of 1 kb/s. The L2 TAD was used as an example.

Second, we would like to point out that we have revised our method to estimate closing rates (see section ‘Additional changes’ at the end of this rebuttal). This new analysis resulted in an upper bound on the estimated motor speed of 0.5 kb/s. For consistency with this upper bound, we now use a motor speed of 0.25 kb/s (instead of the 1 kb/s previously used) for our estimates of cohesin density, residence time and loading rate. This led us to estimate a cohesin density of 12-18 Mb⁻¹, a cohesin residence time of 15-25 min (Fig. 7a), and a loading rate of 0.5-1.2 (min x Mb⁻¹).

Cohesin density: The range of cohesin density we estimate (12-18 Mb⁻¹) is in fact rather narrow (max/min = 1.5) and thus provides a solid foundation for understanding TAD dynamics.

Cohesin residence time: we acknowledge that our previous estimates of residence time (and hence loading rate, see below) spanned approximately one order of magnitude. However, we now provide a narrower estimate of 15-25 min (max/min = 1.7).

Loading rate: We would like to clarify that while we estimated cohesin density, residence time, and motor speed by comparing simulations to experiments, the loading rate was simply computed as the ratio of cohesin density to residence time (Extended Data Fig. 8l and Methods section ‘Comparison of simulations to experiments’). Hence the range obtained for the cohesin loading rate (0.5-1.2 (Mb x min)⁻¹) is a direct consequence of the range of residence times. We would like to stress that, to our knowledge, we provide the first estimation of cohesin loading rate in human cells, and that this estimated range is 40% narrower than the range of 0.06-1.2 (min x Mb)⁻¹ reported in mESCs by Mach *et al*¹¹.

Third, we illustrate below that our estimated parameters are biologically meaningful by using them to predict the kinetics of cohesin reloading after mitotic exit. Mitotic chromosomes are depleted from cohesin, and cohesin density gradually increases after mitotic exit, until it reaches an equilibrium where cohesin unloading is balanced by cohesin loading²². Our estimated cohesin

density (12-18 Mb⁻¹) and residence time (15-25 min) allows us to predict that this equilibrium is reached after 69-103 min (Fig. F6 and Appendix A6). This estimate is consistent with ChIP-Seq and Hi-C analyses, showing that cohesin levels recover 60-120 min following mitotic exit or after RAD21 depletion in unsynchronized cells^{1,23}. Thus, our estimations of cohesin dynamics enable quantitative predictions consistent with previous experimental observations of cohesin reloading following mitotic exit.

In summary, we demonstrated that our approach provides accurate and narrow estimates of cohesin density and that the larger range of estimated cohesin residence times reflects a methodological limitation rather than biological variability. Nevertheless, we showed that our estimates allow meaningful and testable biological predictions, as illustrated by predicting cohesin reloading after mitotic exit, that are fully consistent with experimental observations.

Fig. F6: Estimated cohesin dynamics parameters predict kinetics of cohesin reloading after mitotic exit.

Solid curves show the predicted levels of chromatin-bound cohesin relative to the equilibrium values for two combinations of parameters. Purple: lowest cohesin density and shortest residence time consistent with experiments. Orange: highest cohesin density and longest residence time consistent with experiments. Dotted lines indicate the time after which 95% of equilibrium cohesin is bound to chromatin. See Appendix A6 for details on the computation of these curves.

Additional points:

5) While figure 1c shows an effect of depletion of RAD21, it does not show as the authors claim on p. 7 that tagging does not affect the TAD. For this they would need to compare the tagged line with the untagged line. I did not see this comparison in figure 1c or elsewhere. Please include.

R1.5: We agree with the reviewer and we apologize for this claim, which was initially made because the 'corner peak' could still be observed in the contact maps of tagged cell lines. To address this, we performed new Capture Micro-C experiments on the untagged parental cell line and found no apparent difference with the tagged cell lines (Fig. F7).

We note that this is consistent with previous reports from Gabriele *et al*¹² and Mach *et al*¹¹ who similarly observed no structural impact of repeat array insertion.

We included this figure as Extended Data Fig. 1b,c.

Figure F7: Capture Micro-C maps of untagged and tagged cell lines.

a. Capture Micro-C maps of tagged (top) and untagged (bottom) cell lines. To facilitate visual comparison, tagged maps (top) were randomly subsampled to match the total number of counts in the untagged map (bottom). **b.** Normalized differential maps of contact counts. Bins within 20 kb around the diagonal were excluded from the maps. Maps are shown at 5 kb resolution in **a,b**.

6) The authors show that Auxin treatment led to a shortening of proximal state lifetimes by only ~ 1.3 -fold (Fig. 2d), which is surprisingly low if the authors' assumption that proximal states requires cohesin binding as stated in their model description. They argue the small effect "might be a consequence of residual loop formation due to incomplete cohesin degradation (Extended Data Fig. 4e)". However, they state on p. 8, that degradation of RAD21 was $>94\%$ efficient. These two statements seems to contradict each other. If degradation was only partial, the interpretation of these data is in jeopardy, if the degradation was efficient, another explanation for the lack of effect on proximal states needs to be found.

R1.6: We acknowledge that the 1.3-fold reduction in proximal state lifetime may seem unexpected. However, we argue below that this is an expected, if somewhat counterintuitive result.

First, we would like to clarify that our model previously stated that closed states require cohesin binding, but not proximal states (we acknowledged above that closed states are primarily cohesin-dependent, although not necessarily exclusively, see R1.1.d). Proximal states generally include closed states but also encompass other conformations (see Appendix A1 for a detailed characterization of proximal and closed states).

The effect of cohesin levels on proximal state lifetimes is not straightforward. To investigate it, we computed proximal state lifetimes in time series from polymer simulations (Fig. F8a). In particular, we compared a cohesin density of 12 Mb^{-1} to a cohesin density of 1 Mb^{-1} , a 92% reduction in cohesin levels that closely mimics the 94% depletion observed experimentally (Extended Data Fig. 3d,e, previously Extended Data Fig. 4d,e mentioned by the reviewer). Strikingly, we observed only a 1.1-1.3-fold reduction in proximal state lifetime upon reducing cohesin levels (Fig. F8a,c) across all targeted loci. Thus, similar proximal state lifetimes can be maintained despite a substantial depletion of cohesin levels. These observations from simulations are very similar to the 1.2-1.4-fold reduction observed experimentally (Fig. 2d and F8b).

This mild dependence of proximal state lifetimes on cohesin levels may be counterintuitive but it can be explained by the difference in residence times between CTCF and cohesin: cohesin remains bound to chromatin longer than CTCF does (22 min vs 2.5 min^{18,24}, respectively), implying that proximal state lifetimes are primarily dictated by CTCF binding. Any residual cohesin reaching TAD anchors in auxin-treated cells results in proximal state lifetimes similar to those in untreated cells. In agreement with this explanation, simulations show that significant increase in proximal state lifetimes occurs only at high cohesin densities (>20 Mb⁻¹), where multiple cohesin molecules can sequentially reach CTCF anchors and collectively extend proximal states (Fig. F8d). Finally, we obtained very similar results for proximal state lifetimes with and without auxin from our stringent and high frequency live-cell imaging datasets (Fig. F2a-c), further supporting these estimates.

In summary, proximal states become rarer upon cohesin degradation (proximal state frequency is reduced by ~65% upon auxin treatment, Fig. 2c) but proximal lifetimes only mildly depend on cohesin levels and the modest reduction in lifetimes observed after auxin treatment is expected and predicted by polymer simulations. These findings underscore that proximal state lifetimes are governed primarily by CTCF dynamics rather than cohesin abundance.

In the revised article, after reporting the results on proximal lifetimes for auxin-treated cells, we now mention that “*While counterintuitive, this result is in fact in good agreement with predictions from polymer simulations (Extended Data Figure 4e)*”, and we include simulation predictions of proximal lifetimes (Fig. F8a,b) as the new Extended Data Figure 4e.

Fig. F8: Expected effect of cohesin depletion on proximal state lifetimes.

a,b, Proximal state lifetimes computed on polymer simulations (**a**) or experimental data (**b**). A cohesin density of 12 Mb⁻¹ was assumed in simulations of untreated cells, and of 1 Mb⁻¹ for auxin-treated cells, matching the 94% RAD21 depletion observed in experiments (Extended Data Fig. 3e). **c**, Example fits of proximal state lifetime distributions in simulations of the L1 TAD for a low (1 Mb⁻¹, left) and high (12 Mb⁻¹, right) cohesin density and a fixed cohesin residence time of 22 min. In **a,c**, a cohesin residence time of 22 min and a motor speed of 0.25 kb/s were assumed, and N=450 simulated time series were used for estimation of lifetimes, matching experimental sampling size (Supplementary Table 1). **d**, Proximal state lifetime in polymer simulations depending on cohesin density and residence time, for a motor speed of 0.25 kb/s. Black squares correspond to non-assessed combinations of parameters or conditions where no proximal state was observed.

Reviewer #2:
Remarks to the Author:

In this work, Sabate et al. describe the first CTCF-CTCF loop dynamics in living human cells, concluding that also in human cells, the CTCF-CTCF loops are rare and short-lived. The work is well done, and the conclusions are strong. Even if this is not the first study to quantify CTCF-CTCF loop dynamics, it is the first one to address this in human cells, whose genomes and cell cycles are significantly different from the mouse ones. Therefore, the fact that this process is conserved among species, makes this work valuable for a journal like Nature Genetics. To my opinion, the strongest point of this paper is that no matter the species, CTCF-CTCF loops are short-lived.

I believe that this paper is almost ready for publications after some polishing.

R2.0: We thank the reviewer for these positive comments.

1) While it's true that in Gabriele and Mach they used mESCs and only one single locus they made important quantifications, which are a bit underdiscussed in the introduction. In the paragraph in line 78 the authors only to present the limitations of those two studies without stressing enough the relevance of the results. In my opinion, to be fair, this paragraph should discuss some of the conclusions of those two papers, highlighting that the frequency of looping depends on CTCF configuration, but the duration of the looped state is independent and short-lived in both studies. This could serve as a bridge also to justify the current work like "those two studies quantified that in mouse loop duration is short-lived, therefore we investigated if this holds true also in human cells, in multiple loci".

R2.1: We agree that we should have given more information on the Gabriele *et al* and Mach *et al* articles in the Introduction, even though we have discussed their results in the Discussion section, alongside our findings.

We have now revised the introduction and added the following sentence to present the results of these two papers and stress their significance:

"Two recent studies in living mouse embryonic stem cells (mESCs) provided crucial insights by directly revealing that TADs are dynamic structures, whose anchors contact only rarely and briefly, proving the transient nature of cohesin-dependent chromatin interactions^{11,12}."

We also already emphasized in the Discussion that our conclusions are consistent with results obtained in mice:

"We thereby extend results initially obtained on mESCs^{26,27} to human cells, which have a ~10-fold longer G1 phase."

2) The authors used the HCT116 line, which is a near-diploid cancer line. It would be useful to know if the authors checked the mutational burden on RAD21, NIPBL, WAPL, and CTCF. Do any of these genes or components of cohesin carry mutations?

R2.2: We thank the reviewer for raising this point. Based on Cellosaurus, none of these proteins carry mutations in HCT116 cells²⁵. Additionally, STAG1 and STAG2, two mutually exclusive subunits of the cohesin ring, are also present in their WT form in HCT116 cells²⁵.

We now mention this in the Methods section 'Cell culture':

“None of the proteins directly involved in cohesin-dependent loop extrusion (RAD21, NIPBL, WAPL, CTCF, STAG1, STAG2) carry mutations in HCT116 cells.”

3) The conclusion at line 94 are very strong since they also did not test all loops but only three (plus a control region) out of thousands. This part could be modified such as “given the results of other studies in mouse and ours, we can suggest that extrusion dynamics is conserved...” Saying that is universal is a big claim and should be done with caution. We know that for example in *Drosophila* can be different. Moreover, the three loci are all in compartment A. Therefore, saying that these results are universal (as well as stated in the title) can be an overclaim and this should affirmation should be tuned down, both in the text and in the title.

R2.3: We agree that our use of the adjective ‘universal’ may have been misleading and apologize for this poor terminology. We used the term ‘universal’ not to imply constancy of cohesin dynamics across species but to imply constancy across the genome, as indicated by the consistency of these parameters estimated independently for three genomic regions (Fig. 5c). However, we agree that ‘universal’ may still be an overstatement in this sense, as we only analyzed three loci in the same compartment out of thousands of loops and TADs in the genome^{26,27}. Nevertheless, we consider the fact that these three loci are consistent with the same cohesin dynamics parameters as an important result that does not prove, but does at least suggest uniformity of these parameters across the genome.

Throughout the manuscript and in the title, we have replaced the term ‘universal’ by ‘uniform’ and rephrased our conclusions of uniform cohesin dynamics. We now indicate in the Discussion: *“Thus, our study suggests uniform dynamics of cohesin across the genome, rather than local tuning (Fig. 7b)”*.

We also now highlight as a limitation in the Discussion that our conclusions are based on only three TADs in the same compartment. Specifically:

“First, our conclusions are based on only three genomic regions, and the chosen regions are particularly strong TADs and loops compared to genomic averages (Extended Data Fig. 1d,e).”, followed by: *“Second, the chosen domains were all located in the A compartment”*.

Concerning conservation among species: again, we had no intention of claiming conservation of cohesin dynamics across species when using the word ‘universal’, and fully agree with the reviewer concerning the variability of loop extrusion models across species. Nonetheless, we find it noteworthy that our estimated parameters for cohesin dynamics agree with those reported in mESCs by Gabriele *et al.* and Mach *et al.*, as mentioned by the reviewer in his first comment.

We therefore now mention in the Discussion:

“Moreover, the above mentioned consistency of our estimates with previous studies in mESCs raises the possibility that cohesin dynamics is conserved among mammalian species^{26,27”}

4) Fig. 1C provides contact maps for the L1, L2, and T1 loci but do not deliver information regarding gene composition, which are only showed in the extended data. I believe that for a better understanding, the gene tracks, pro-seq, and CTCF peaks should be represented also in main as polished version and not as screenshot from presumably from HiGlass, which makes visualization difficult. Also it seems there is an artifact in the plotting of L1 map since the diagonal is not extending for the whole length of the box and it appears truncated with respect on the underlying tracks.

R2.4: We agree that including additional genomic information in the main Fig. 1c would be informative. However, due to space constraints, adding 1D tracks below each Micro-C map would result in tracks that are too small to be readable (see Fig. F9c as an example). This is the reason why we included them in Extended Data. Fig. 1 instead.

The maps in Fig. 1c are not screenshots from HiGlass but were generated using *cooltools*²⁸, while the maps and 1D genomic tracks in Extended Data Fig. 1 were created with *CoolBox*²⁹.

Regarding the L1 map, the blank line is not an artifact but corresponds to a region where probes for the Capture part of the Capture Micro-C protocol were not designed. This results in an absence of signal at this genomic location. This blank line is nevertheless displayed to show all Capture Micro-C maps in Fig. 1c with the same total genomic size, enabling better comparisons of genomic sizes between the different TADs.

We mention this explanation in the revised legend of Fig. 1c:

“The white space in the L1 map corresponds to a region that was not covered by capture probes.”

Fig. F9: Example Fig. 1 if 1D genomic information was included below each Capture Micro-C map. Labels and 1D genomic tracks are not readable in panel c.

5) In L1 and L2, 3 hours of auxin treatment removes most of the interaction, but it's possible to notice residual contact despite the efficiency of RAD21 depletion. Can the authors speculate on the cause of that residual contact?

R2.5: We agree that residual contact peaks can be observed at the CTCF anchors in L1 and L2 regions despite auxin treatment. We provide two possible explanations for this residual contact. First, in WT conditions, it has been estimated that ~40-60% of the total cohesin pool is bound to chromatin^{18,19}. Therefore, even with a depletion of 94% of the total cohesin pool (Extended Data Fig. 3e), 10-15% of the initial cohesin levels may still be chromatin-bound, assuming that all remaining cohesin is chromatin-bound. This residual cohesin could maintain a small amount of contact between anchors and hence explain the residual peaks despite efficient cohesin depletion.

Second, as proximal states are reduced by ~72% upon auxin depletion (Fig. 2b), but not completely abolished, it is also possible that additional, cohesin-independent processes contribute to the residual contacts (see R2.8).

6) Fig 1g, for clarity I would rename the "- Auxin" to "no auxin" as the minus could be confused with a dash.

R2.6: We agree with the reviewer that the label 'No auxin' reduces ambiguity as compared to '- Auxin'.

We have accordingly updated the labels in all main and supplementary figures.

7) Panel 1C is mentioned in the text before 1B, would it be better to invert the label?

R2.7: Fig. 1b is referenced alongside Fig. 1a, at the end of the first paragraph of the Results section 'Visualizing the dynamics of TAD anchors', prior to the first mention of Fig. 1c:

"[...] respectively bound by TetR-splitGFPx16-NLS and CymR-NLS-2xHalo (imaged with the bright and photostable dye JFX646⁴¹; Fig. 1a,b)".

Therefore, we believe the current order is consistent with the text.

8) Line 410: I have a potentially naïve question. The authors state that in the auxin-treated cells, the estimate of the simulated cohesin density of 2-fold. A decrease of 2-fold, to me, is surprisingly mild. I am not sure what I would have expected, but only 2-fold leads me to think that maybe this approach is too conservative. Can the author explain this part better?

R2.8: We agree with the reviewer that this result is not expected and apologize for not having discussed it sufficiently before.

To clarify what should be expected from a strong depletion of cohesin, we used polymer simulations with high (12 Mb^{-1}) and low (2 Mb^{-1}) densities of cohesin, and asked if our method could retrieve the correct density (F5b,c in R1.4). As detailed in R1.4, our method can accurately retrieve both low and high cohesin densities. This suggests that the 2-fold reduction in our estimates of cohesin density upon auxin treatment does not arise from a methodological limitation. Importantly, using untreated cells, the six metrics derived from live-cell imaging and Micro-C converged to a similar cohesin density and the same range of residence times (Fig. F10a), with the minor exception of proximal lifetimes, which however do not vary strongly in the considered parameter range compared to other metrics. However, using auxin-treated cells, the metrics diverged and did not agree with each other: for instance, distance distributions pointed to a cohesin density below 1 Mb^{-1} (a >90% reduction from the untreated condition), as expected, but

proximal fraction or frequency pointed to a density of 6 Mb^{-1} (a 50% reduction only; Fig. F10b). This indicates that our approach cannot reliably estimate cohesin dynamics parameters in auxin-treated cells. Thus, our initial statement of “a 2-fold reduction in cohesin density after auxin depletion” was inaccurate. A more accurate conclusion is that our method fails to provide a consensus estimate, thereby preventing us from quantifying cohesin density in auxin-treated cells.

Together with the validation of our method on simulated data, this observation suggests that an additional biological process, not included in our simulations, could influence chromatin dynamics in a visible way in absence of cohesin. We cannot identify this process, but can suggest potential candidates such as A/B compartmentalization^{4,30,31} (a process orthogonal to loop extrusion that strengthens upon cohesin depletion^{4,31}) or the presence of other extruding complexes distinct from cohesin (e.g. SMC5/6, which also binds to chromatin in G1³²). Nevertheless, our experimental data indicate that cohesin-dependent loop extrusion is the dominant force shaping chromatin dynamics at TAD anchors under normal conditions. Indeed, in untreated cells, the large majority of proximal states (~72%) are dependent on cohesin (Fig. 2b), and all metrics consistently converge to similar parameters.

We added this explanation in the corresponding section ‘TAD dynamics is consistent with uniform cohesin dynamics and is predicted by CTCF binding’ and included Fig. F10b as the new Extended Data Fig. 9c.

Fig. F10: Estimation of cohesin dynamics parameters in untreated and auxin-treated cells.
a,b Metrics measuring the deviation of simulations from experimental data using untreated (**a**) or auxin-treated (**b**) cells. The sum of the 6 quantities yields the overall deviation of simulations from experiments. Solid and dashed grey lines indicate the 10% and 25% best parameter sets, respectively. In untreated cells, the six metrics agree with similar cohesin densities and residence times, whereas they disagree with each other in auxin-treated cells. We assumed a motor speed of 1 kb/s and the L2 TAD was used as an example.

9) In line 463, the authors state that they performed the first measurement of cohesin-dependent extrusion speed in living cells. This is only partially true and holds only for humans. In Gabriele et al they quantified cohesin processivity (processivity = lifetime × extrusion speed). With the calculated cohesin density of processivity of 150 kb and the base lifetime of ~20 minutes, it's easy to calculate the extrusion speed (as also indicated in their supplementary materials). Remarkably, $150\text{kb}/1200\text{s} = 0.125\text{kb/s}$, which is the very same as Sabate et al. best estimate. Similarly, also in Mach et al the loop extrusion was quantified from simulated data, ranging between 0.1kb/s to 1kb/s. Therefore, the authors here should rephrase their claim of “first measurement”. However, this is an important piece of information because this paper, taken together with Gabriele et., strongly agree on the estimated extrusion speed. Taken together, they independently suggested that loop extrusion speed in vivo is generally the same in mice and humans. Despite this, I still believe that the term “universal” does not hold true because suggest that the same mechanism is indeed “universal”. This term can be used with the genetic code, with exceptions, but in this context “conserved” is much more appropriate than universal.

R2.9: We agree with the reviewer's comment.

First, we do not wish to insist on having provided the first measurement of cohesin speed in living cells (even though we consider our method somewhat more direct) and we therefore removed this statement.

Second, we have revised our method used to compute closing rates (see ‘Additional changes’ section below). Based on this new analysis, we no longer provide a single estimated motor speed of ~0.1 kb/s but instead determine an upper bound of 0.5 kb/s. This upper bound remains consistent with our initial prediction of ~0.1 kb/s and with the estimate of 0.125 kb/s from Gabriele *et al* and we acknowledge that the agreement between these independent estimates is worth mentioning.

To address the reviewer's comment and reflect this new analysis, we have modified the Discussion to:

- (i) clarify that we now provide an upper bound for extrusion speed in live *human* cells
- (ii) mention the agreement between our findings and those of Gabriele *et al*
- (iii) mention the putative conservation of cohesin dynamics among mammals (see R2.3)

As for the term ‘universal’, please also see R2.3 above.

10) Again, the statement in line 523 is not true and holds only as the first estimate in human cells.

R2.10: We addressed this comment in R2.9 above.

11) Similarly, in discussion at line 572, claiming that these results are universal should be supported by experiments and simulation also in compartment B, lamina-associated domains, etc. In this paper, for the first time, they studied three loci instead of single loci as in Gabriele and Mach. However, still, these are only three loci and might not represent universally cohesin behavior outside compartment A.

R2.11: We agree with the reviewer and have revised our terminology and phrasing, as detailed in R2.3 above. Concerning this specific sentence in the Discussion, we rephrased it as “*Thus, our study suggests uniform dynamics of cohesin across the genome, rather than local tuning*”, which we believe is sufficiently cautious. Additionally, we also now explicitly acknowledge in the

Discussion that our study's conclusions are based on three regions located in the A compartment only (see R2.3).

12) The statement starting from line 583 "Although differences in Hi-C patterns between A and B compartments have been shown to originate from differential CTCF binding and not from differences in cohesin dynamics, it still remains important to determine parameters of cohesin dynamics in repressive chromatin contexts." Agrees with my concern about the use of "universal" since these experiments are only done in compartment A.

R2.12: We agree with the reviewer's concern. As previously stated (see R2.3), we have revised the manuscript to tone down this statement and no longer use the word 'universal'.

13) Extended data Fig.3 panel A represent the image analysis pipeline. Given the fact that they performed also tetraspeck beads correction as stated in the method, I would suggest that this panel would feature both beads+actin.

R2.13: We apologize for the lack of clarity here. All live-cell imaging time lapses were actually corrected for chromatic aberration using dual-color actin images from living cells, and not using fluorescent beads (Extended Data Fig. 2a,b, previously Extended Data Fig. 3a,b). We only used fluorescent beads to estimate the performance of the chromatic aberration correction determined from the dual-color actin images. We already included an uncorrected image of a fluorescent bead in Extended Data Fig. 2b and an image of actin in Extended Data Fig. 2a.

We have modified the Methods section 'Correcting for chromatin aberrations' to clarify this point.

14) In case the authors performed 3D genomics before and after removing the Bsd and Neo resistances, it would be interesting and useful to make these data available to the scientific community to have a quantification of the impact of having highly transcribed genes at the border of the TAD. Especially in light of the retention of resistance genes observed in a small fraction of the T1 line, as they state in the methods and the fact that the contact maps of T1 after cohesin depletion seems not to be affected by this small contamination.

R2.14: Unfortunately, we did not perform Capture Micro-C before the removal of blasticidin and neomycin resistance genes. However, in Mach *et al*¹¹, the authors conducted this experiment (in mESCs) and found no difference in the Capture-C maps with or without the strong PGK promoter (Extended Data Fig. 6f,g of Mach *et al*¹¹). Furthermore, they observed no difference in cohesin-dependent chromatin motion with or without active transcription (Extended Data Fig. 7c of Mach *et al*¹¹). These findings suggest that the retention of antibiotic resistance genes in a small fraction of the T1 cell line is not expected to significantly affect the cohesin-dependent dynamics of the TAD.

In Summary, I recommend this paper for publication after minor but important revisions (included other comments/questions above):

- removal of the word "universal," both from title and text which is an overstatement, as also declared by the authors in line 584. At most, it could be in the discussion as a hypothesis, underscoring limitations.
- correct the possible artifact of L1 map
- double-check label progression
- correct of the statement "first paper to measure in living cells loop extrusion speed" and discussion of Mach and Gabriele papers regarding speed extrusion calculation in vivo. In which

in the first they use a range, and in the latter they have exactly 0.125kb/s outside the regions where CTCF boosts cohesin residency.

R2.15: We thank the reviewer for this appreciation and hope that we have satisfactorily addressed the specific requests above.

Additional changes not requested by the reviewers:

Beside the above-mentioned modifications, we introduced additional changes to the manuscript:

Additional change #1:

We have expanded the set of parameter combinations explored in our simulations with a motor speed of 0.25 kb/s (Extended Data Fig. 9d) and now include these in Fig. 5a,c.

Additional change #2:

We improved our method to fit closing rates because the original approach produced an incorrect trend with simulated data, where increasing cohesin motor speed led to a reduced closing rate. With our new fitting method (detailed in Appendix A7), the measured closing rates consistently increase with cohesin motor speed (Fig. F11). Importantly this new fitting approach did not affect the estimations of closing rates, since we now obtain the range 0.07-0.16 kb/s (Fig. 3e), which remains very similar to our previous estimates of 0.06-0.19 kb/s.

Moreover, our new method allows to determine the maximum motor speed up to which our imaging frequency allows to detect the linear decrease in distances expected from extrusion. Our analysis indicates that this maximum speed is 0.5 kb/s (see new Fig. 5e and new Extended Data Fig. 9e). For speeds of 0.5 kb/s or higher, the linear decrease cannot be detected at our imaging frequency (Extended Data Fig. 9f,g). Since we actually detect this decrease in our experimental data, we are able to place an upper bound on the cohesin motor speed of 0.5 kb/s (consistent with our previous estimated speed of ~0.1 kb/s). However, closing rates are influenced by both motor speed, which we vary, and residence time, which we fixed (motor speed \approx processivity / residence time). Since fitting closing rates requires a large amount of time series and fully taking into account the possible range of residence times would require unfeasibly extensive exploration of the parameter space, we could not narrow down further our estimate of cohesin motor speed.

Fig. F11: Comparing closing rates measured in experiments and simulations.

Closing rates determined from experimental (red) or simulated (shades of purple) distance time series. Simulations assumed a residence time of 22 min and a cohesin density of 12 Mb^{-1} , and explored cohesin motor speeds of 0.125, 0.2, 0.25, 0.5 or 1 kb/s. Boxplot lines show median, lower and upper quartiles, while whiskers extend from the 10 to 90 percentiles. From $N=5,000$ bootstrap samples, the number of bootstraps exhibiting at least 15 time series and consistent with linearly decreasing distances is indicated. Closing rates are not measured for T1 and Half TAD at 0.125 kb/s because $N=0$. ***: $P\text{-value} < 0.001$ from a two-sided Mann-Whitney U test, adjusted for multiple testing by Bonferroni correction. This figure was removed from the manuscript because it only considers a unique cohesin residence time, and varying this parameter would be necessary for precisely estimating the motor speed.

Additional change #3:

Based on our new simulations and analyses of closing rates discussed above, we have updated the parameters used to compute TAD dynamics in the final new section 'Implications for TAD dynamics', from a cohesin density of 8 Mb^{-1} , a residence time of 18 min and a motor speed of 1 kb/s, to a cohesin density of 12 Mb^{-1} , a residence time of 22 min and a motor speed of 0.25 kb/s. This new parameter combination better represents all genomic regions (Fig. 5c) while accounting for the upper bound of 0.5 kb/s on motor speed, which is consistent with a speed of 0.25 kb/s, unlike the speed of 1 kb/s previously used. These parameters are now used in the new Figure 6, the new Extended Data Figs. 9h-j and 10, and the updated Supplementary Video 2.

Additional change #4:

We reorganized Extended Data Figures to comply with the limit of 10 supplementary figures, which involved merging figures and eliminating the original Extended Data Fig. 2.

Appendix

A1. Proximal states generally encompass closed states but are not limited to them

To help clarify the link between closed and proximal states, we provide a detailed comparison of the two states using polymer simulations. We recall that closed states are strictly defined as direct interactions between the two TAD anchors, excluding cases where multiple loops connect the anchors (Fig. F12a). In comparison, proximal states are defined as time intervals longer than the temporal threshold, during which all distances fall below the spatial threshold.

We show that closed state intervals generally overlap (*i.e.* intersect) with proximal state intervals (78-100% of closed state intervals intersect a proximal one; Fig. F12a), although proximal intervals are not limited to closed state intervals (94-100% of proximal state intervals do not overlap a closed state interval; Fig. F12b). Nevertheless, these proximal intervals distinct from closed states occur toward the end of the extrusion phase, when the unextruded genomic distance between anchors is already small (Fig. F12c,d). Proximal state intervals overlapping closed states generally include the closed state interval and extend before and after closed state starting and ending timepoints, respectively (Fig. F12c), leading to proximal states lasting longer than closed states (Fig. F12e). The cases where closed state intervals are not detected as proximal (*i.e.* no overlap between proximal and closed intervals) generally correspond to closed state intervals shorter than the temporal threshold (3 min) used for segmentation (Fig. F12f).

We included this analysis as the new Extended Data Fig. 10a-f and updated the section 'Characterization of proximal and closed states' in the Methods.

Fig. F12: Detailed comparison of proximal and closed states.

a,b, Percentages of closed state intervals overlapping proximal state intervals (**a**) and percentages of proximal state intervals overlapping closed state intervals (**b**). ‘Multiple intervals’ designates cases where more than one interval overlapped. **c**, Reproduction of Fig. 6c. Anchor-anchor distance time series (red) from polymer simulations. The shortest 1D path (unextruded genomic length) connecting TAD anchors is indicated in grey. Green and purple rectangles at the bottom of time series indicate closed states and segmented proximal states, respectively. Note that proximal states (purple bars) encompass closed states (green bars) or correspond to the end of extrusion with cohesin complexes close to TAD anchors (low unextruded length, in grey) if they do not overlap with a closed state. **d**, Mean shortest 1D path (unextruded length) aligned on t_{start} of closed (orange) or segmented proximal (purple) states. **e**, Duration of all closed states compared to proximal states (top) and duration of all proximal states compared to closed states (bottom). If multiple intervals overlapped with a single one, the duration of the intervals was summed. The black area shows the kernel density estimation of the duration distribution. **f**, Histogram of closed state duration for closed states overlapped (orange) or not overlapped (grey) by at least one proximal state. Polymer simulations assumed a cohesin density of 12 Mb^{-1} , a residence time of 22 min and a motor speed of 0.25 kb/s .

A2. Polymer relaxation is rare and too fast to be captured at our imaging frequency

To examine the kinetics of relaxation, we used polymer simulations at high temporal resolution (1 simulated time point per 3 s). Simulated distance time series were aligned to the end of closed states, differentiating between events where cohesin dissociates or resumes extrusion. In the absence of measurement noise, relaxation following cohesin dissociation was completed in ~1 minute (Fig. F13a). This indicates that while polymer relaxation indeed occurs, it is a rapid process, barely captured by our live-cell imaging frequency (2 frames at the experimental imaging frequency of 1 image per 30 s), even in the absence of noise. In the presence of measurement noise, and after aligning time series to proximal instead of closed states, where cohesin may or may not be present at anchors, an indiscernible flat profile of distances with no detectable relaxation was observed (Fig. F13b, see Appendix A1 for detailed characterization of closed and proximal states). The same flat profile was observed when using experimental instead of simulated time series (Fig. F13c).

Additionally, cohesin dissociation following closed states is rare (~8% of events) compared to extrusion resumption (~92%, Fig. F13a), owing to cohesin's longer residence time relative to CTCF. Thus, both the rapid nature of polymer relaxation and the rarity of cohesin dissociation make relaxation events unlikely to affect TAD dynamics in our imaging approach.

This analysis is now included as the new Extended Data Fig. 5c-e and discussed in the Methods section 'Estimation of loop state fractions':

"However, we found that polymer relaxation is rare (it occurred for 8% of closed states, which are only observed 0-1% of the time) and too transient (~1 min, equaling 2 imaging frames) as compared to our imaging frequency to affect our conclusions (Extended Data Fig. 5c-e)".

Fig. F13: Polymer relaxation following cohesin dissociation is rare and too rapid to be captured. **a,b**, Simulated time series sampled at high frequency (1 snapshot / 3 s) were aligned to the end of closed (**a**) or proximal (**b**) states for events where cohesin dissociates from the polymer (purple) or resumes extrusion (green). Time series for all events, regardless of the cohesin fate, are shown in orange (overlapping with green). The proportion of closed state events resulting in cohesin dissociation or extrusion resumption is indicated. Time series without (**a**) or with measurement noise (**b**) were used. Simulations assumed a cohesin density of 12 Mb^{-1} , a residence time of 22 min and a motor speed of 1 kb/s. Dotted lines mark the first three timepoints corresponding to the experimental live-cell imaging frequency (1 image per 30 s). **c**, Same as **b**, but using experimental time series. In **b,c**, the small peak at $t = 0.5 \text{ min}$ is a result of our threshold-based segmentation of proximal states, where the distances at the first and second timepoints are required to be below and above the spatial threshold, respectively.

A3. Validating the 3-state model on simulated data

To quantitatively evaluate potential errors using the analytical 3-state model of coordinate differences, we used polymer simulations where the ground truth is perfectly known. We found that the open state was accurately estimated, with an absolute error of 0% under most conditions (Fig. F14), except at very low cohesin densities ($\leq 2 \text{ Mb}^{-1}$, Fig. F14) that are not consistent with experimental data (Fig. 5c). Additionally, we observed that proximal states were overestimated at the expense of extruding states for some parameter combinations. Hence, extruding states can be underestimated in our 3-state model, thereby reinforcing our conclusion that TADs are constantly extruded.

We added this analysis as the new Extended Data Fig. 5f.

Fig. F14: Quantification of errors in the estimation of the three loop states.

Absolute error in the estimated fractions of proximal, extruding and open states from polymer simulations, depending on cohesin residence time and density, assuming a motor speed of 1 kb/s. For each parameter pair, $N=100$ bootstrap samples were considered, where each bootstrap included the same number of time series than in the experimental live-cell imaging data. Black squares correspond to unexplored pairs of parameters or conditions with less than 50 distances to define the proximal state distribution. Note that open state fractions are accurately estimated except for very low cohesin densities ($<2 \text{ Mb}^{-1}$), which are inconsistent with experiments, and where maximal absolute errors only reach $\sim 20\%$.

A4. Relationship between distances and the loop extrusion process

While there is no direct relationship between single anchor-anchor distances and loop states, we recall here multiple independent lines of evidence showing that loop extrusion affects the distribution and temporal dynamics of anchor distances.

- Removal of cohesin, the protein driving loop extrusion, results in significantly higher overall distances (Fig. F15a).
- The alignment of time series on the proximal state starting timepoints shows a progressive decrease of distances prior to reaching the proximal state, as expected from the progressive enlargement of the loop during the loop extrusion process (Fig. F15b).
- Using the 3-state analytical model, we inferred that auxin-treated cells (lacking extrusion) display 90% of open states, as expected from a near absence of extrusion (Fig. F15c). In contrast, untreated cells exhibit lower levels of open states and higher proportions of extruding and proximal states (Fig. F15c).
- Increasing cohesin levels in polymer simulations leads to smaller mean anchor-anchor distances (Fig. F15d and Extended Data Fig. 8c).
- Cohesin depletion increases chromatin motion in experiments (Fig. F15e).

These results collectively demonstrate a clear relationship between loop extrusion and anchor-anchor distance distributions and dynamics.

Fig. F15: Evidence for the relationship between loop extrusion and anchor-anchor distances.

a, CDFs of anchor-anchor distances as in Fig. 1g. Note that distances increase in auxin-treated cells (no RAD21). **b**, Time series of squared anchor-anchor distances aligned to the start of proximal states, as in Fig. 3d. A linear decrease expected from loop extrusion is observed in untreated cells (red) and is absent in randomly shuffled time series (dark yellow). **c**, Estimated fractions of open, proximal and extruding states, as in Extended Data Fig. 5b, but without the proximal states. Auxin treatment leads to high fractions of open states. **d**, Mean 3D distance depending on cohesin level as predicted by polymer simulations. A cohesin density of 12 Mb^{-1} , a residence time of 22 min and a motor speed of 0.25 kb/s were assumed. **e**, Experimental 2-point Mean Squared Displacements (MSD) for untreated (shades of red) and auxin-treated (shades of blue) cell lines, and the Adjacent locus (grey), as in Extended Data Fig. 3h. Cohesin constrains chromatin dynamics, as observed by the higher 2-point MSD values upon auxin treatment.

A5. Model- and assumption-free estimation of open state fraction

As additional evidence for our claim that TADs constantly undergo extrusion, we provide below a new analysis showing that open states are rare or nearly absent. Unlike the 3-state model, this analysis does not use any model but relies only on the experimentally measured distance distributions in presence or absence of auxin.

Specifically, we considered the distance distribution measured in auxin-treated cells -where cohesin is depleted and the anchors are expected to be mostly in the open state- (Fig. F16, dark blue curves) and asked what proportion of distances measured in the untreated cells (Fig. F16, red curves) could possibly arise from this first (dark blue) distance distribution. To do this, we weighted the “+ Auxin” distribution by a multiplicative coefficient varying between 0 and 100% and determined for which values of this coefficient the weighted distribution fits under the “untreated” distribution (more exactly, if 95% of the distance bins have lower values in the weighted vs the untreated histograms). The untreated distribution is a sum of two distributions: the distribution of distances in open states and the distribution in non-open states. Therefore, the weighted distribution must be bound from above by the untreated distribution. This places a strong upper limit to the multiplicative coefficient, and hence on the fraction of open states consistent with distances in the auxin-treated cells. In practice, we determined maximal open state fractions of 18%, 4% and 5% for L1, L2 and T1, respectively (Fig. F16). Thus, without any model or assumption on loop states, we demonstrate that open states are rare or nearly absent.

Fig. F16: Model-free analysis of distance distribution shows that open states are rare

Anchor-anchor distance distributions from untreated (red), auxin-treated (dark blue) cells, and the distribution of auxin-treated cells, weighted by a multiplicative factor (<1) such that 95% of distance bins fit below the red histogram (light blue). This multiplicative factor provides an upper bound on the fraction of distances in the untreated cells corresponding to open states (“Max open”).

A6. Computing the time for cohesin to reestablish on chromatin following mitotic exit

To illustrate the biological significance of our estimated range of cohesin dynamics parameters, we computed the expected density of chromatin-bound cohesin ρ (in Mb⁻¹) as function of time following mitotic exit, and compared with experimental observations. From the estimated cohesin residence time τ (in s) and the cohesin loading rate λ (in (Mb x s)⁻¹), we can compute the evolution equation of the chromatin-bound cohesin density as:

$$\frac{d\rho}{dt} = \lambda - \frac{\rho}{\tau}$$

and the cohesin density as function of time, as displayed in Fig. F6:

$$\rho(t) = \lambda\tau(1 - e^{-t/\tau})$$

A7. Improved closing rate estimation

We refined the method used to fit closing rates and explain the improvements below (Fig. F17). The original approach produced an incorrect trend where increasing cohesin motor speed led to a reduced closing rate. This was due to the fact that we fitted the parameter R_{int}^2 . As shown in Fig. F17b below (green lines), this approach led to an almost flat model of distances at high motor speeds, where the highest slope is expected. Because a higher motor speed implies an increased processivity (for a constant cohesin residence time), the plateau of averaged distances ($R_{plateau}^2$) is lower (Fig. F17b), which resulted in a lower closing rate.

To address this issue, we now fix R_{int}^2 to the squared distance at t_{start} , thereby fitting the model up to the proximal state (Fig. F17). The new method provides the expected increase of closing rates with assumed cohesin motor speed, as shown in Fig. F11.

Fig F17: Improved method for closing rate fitting

a, Schematic summarizing the parameters of the closing rate fitting method, as in Fig. 3c. **b**, Mean squared anchor-anchor distances (shades of purple), aligned on t_{start} and averaged, from polymer simulations of increasing motor speeds, as in Fig. 5e. Green dashed lines show the result of the previous fitting approach (where R_{int}^2 is freely adjusted). Pink dashed lines show the results of our improved approach (where R_{int}^2 is fixed to the measured squared distance at $t = t_{start} = 0$). For simulated motor speeds exceeding 0.5 kb/s, our imaging frequency cannot capture the decrease of squared distances.

References:

1. Rao, S. S. P. *et al.* Cohesin Loss Eliminates All Loop Domains. *Cell* **171**, 305-320.e24 (2017).
2. Wutz, G. *et al.* Topologically associating domains and chromatin loops depend on cohesin and are regulated by CTCF, WAPL, and PDS5 proteins. *EMBO J.* **36**, 3573–3599 (2017).
3. Bintu, B. *et al.* Super-resolution chromatin tracing reveals domains and cooperative interactions in single cells. *Science* **362**, eaau1783 (2018).
4. Schwarzer, W. *et al.* Two independent modes of chromatin organization revealed by cohesin removal. *Nature* **551**, 51–56 (2017).
5. Haarhuis, J. H. I. *et al.* The Cohesin Release Factor WAPL Restricts Chromatin Loop Extension. *Cell* **169**, 693-707.e14 (2017).
6. Li, Y. *et al.* The structural basis for cohesin–CTCF-anchored loops. *Nature* **578**, 472–476 (2020).
7. Davidson, I. F. *et al.* CTCF is a DNA-tension-dependent barrier to cohesin-mediated loop extrusion. *Nature* **616**, 822–827 (2023).
8. Sabaté, T., Lelandais, B., Bertrand, E. & Zimmer, C. Polymer simulations guide the detection and quantification of chromatin loop extrusion by imaging. *Nucleic Acids Res.* **51**, 2614–2632 (2023).
9. Brandão, H. B., Gabriele, M. & Hansen, A. S. Tracking and interpreting long-range chromatin interactions with super-resolution live-cell imaging. *Curr. Opin. Cell Biol.* **70**, 18–26 (2021).

10. Carlier, F. *et al.* Remodeling of perturbed chromatin can initiate de novo transcriptional and post-transcriptional silencing. *Proc. Natl. Acad. Sci.* **121**, e2402944121 (2024).
11. Mach, P. *et al.* Cohesin and CTCF control the dynamics of chromosome folding. *Nat. Genet.* **54**, 1907–1918 (2022).
12. Gabriele, M. *et al.* Dynamics of CTCF- and cohesin-mediated chromatin looping revealed by live-cell imaging. *Science* **376**, 496–501 (2022).
13. Arbona, J.-M., Herbert, S., Fabre, E. & Zimmer, C. Inferring the physical properties of yeast chromatin through Bayesian analysis of whole nucleus simulations. *Genome Biol.* **18**, 81 (2017).
14. Dekker, J., Rippe, K., Dekker, M. & Kleckner, N. Capturing Chromosome Conformation. *Science* **295**, 1306–1311 (2002).
15. Dekker, J. Mapping in Vivo Chromatin Interactions in Yeast Suggests an Extended Chromatin Fiber with Regional Variation in Compaction *. *J. Biol. Chem.* **283**, 34532–34540 (2008).
16. Ringrose, L., Chabanis, S., Angrand, P., Woodroffe, C. & Stewart, A. F. Quantitative comparison of DNA looping in vitro and in vivo: chromatin increases effective DNA flexibility at short distances. *EMBO J.* **18**, 6630–6641 (1999).
17. Lesage, A., Dahirel, V., Victor, J.-M. & Barbi, M. Polymer coil–globule phase transition is a universal folding principle of *Drosophila* epigenetic domains. *Epigenetics Chromatin* **12**, 28 (2019).

18. Hansen, A. S., Pustova, I., Cattoglio, C., Tjian, R. & Darzacq, X. CTCF and cohesin regulate chromatin loop stability with distinct dynamics. *eLife* **6**, e25776 (2017).
19. Holzmann, J. *et al.* Absolute quantification of cohesin, CTCF and their regulators in human cells. *eLife* **8**, e46269 (2019).
20. Gerlich, D., Koch, B., Dupeux, F., Peters, J.-M. & Ellenberg, J. Live-Cell Imaging Reveals a Stable Cohesin-Chromatin Interaction after but Not before DNA Replication. *Curr. Biol.* **16**, 1571–1578 (2006).
21. Kueng, S. *et al.* Wapl Controls the Dynamic Association of Cohesin with Chromatin. *Cell* **127**, 955–967 (2006).
22. Morales, C. & Losada, A. Establishing and dissolving cohesion during the vertebrate cell cycle. *Curr. Opin. Cell Biol.* **52**, 51–57 (2018).
23. Zhang, H. *et al.* Chromatin structure dynamics during the mitosis-to-G1 phase transition. *Nature* **576**, 158–162 (2019).
24. Brunner, A. *et al.* Quantitative imaging of loop extruders rebuilding interphase genome architecture after mitosis. *J. Cell Biol.* **224**, e202405169 (2025).
25. Bairoch, A. The Cellosaurus, a Cell-Line Knowledge Resource. *J. Biomol. Tech. JBT* **29**, 25 (2018).
26. Krietenstein, N. *et al.* Ultrastructural Details of Mammalian Chromosome Architecture. *Mol. Cell* **78**, 554-565.e7 (2020).
27. Hsieh, T.-H. S. *et al.* Resolving the 3D Landscape of Transcription-Linked Mammalian Chromatin Folding. *Mol. Cell* **78**, 539-553.e8 (2020).

28. Open2C *et al.* Cooltools: Enabling high-resolution Hi-C analysis in Python. *PLOS Comput. Biol.* **20**, e1012067 (2024).
29. Xu, W. *et al.* CoolBox: a flexible toolkit for visual analysis of genomics data. *BMC Bioinformatics* **22**, 489 (2021).
30. Mirny, L. A., Imakaev, M. & Abdennur, N. Two major mechanisms of chromosome organization. *Curr. Opin. Cell Biol.* **58**, 142–152 (2019).
31. Nuebler, J., Fudenberg, G., Imakaev, M., Abdennur, N. & Mirny, L. A. Chromatin organization by an interplay of loop extrusion and compartmental segregation. *Proc. Natl. Acad. Sci. U. S. A.* **115**, E6697–E6706 (2018).
32. Venegas, A. B., Natsume, T., Kanemaki, M. & Hickson, I. D. Inducible Degradation of the Human SMC5/6 Complex Reveals an Essential Role Only during Interphase. *Cell Rep.* **31**, 107533 (2020).

Referee expertise:

Referee #1:

Referee #2:

Referee #3: 3D genome, imaging

We thank the three reviewers for their assessment of our revised manuscript. We respond to the remaining comments in detail below.

Reviewers' Comments:

Reviewer #1 (Remarks to the Author):

Sabate et al. have submitted a revised manuscript and a 46 page rebuttal! Many of the new data is only provided to reviewers, which is a missed opportunity to provide clarifying information.

The authors make two major conclusions as per their abstract: First, “that TADs are dynamic structures whose anchors are brought in proximity about once per hour and for 6-19 min (~16% of the time)”. This conclusion is confirmatory of earlier studies in the field. Second, “TADs are continuously subjected to extrusion by multiple cohesin complexes”. This would be a novel conclusion, but I remain unconvinced by the strength of the data to support this claim.

I maintain my point that the authors have not demonstrated that they can determine whether a loop is extruding or not or that they can distinguish extruding from closed loops. At no point do the authors measure loop extrusion and as such they can not deduce information about when loop extrusion occurs. The estimates are either purely based on simulations or use RAD21 degradation data. The authors assume that loss of RAD21 only affects extruding loops, but this assumption is not validated. Cohesin does not only bind at TAD anchors but also affects chromatin organization via TAD-independent mechanisms. In the absence of clear orthogonal validation of the ability to distinguish extruding and non-extruding states, the main conclusion regarding pervasiveness of extrusion is not convincingly supported.

Another major weakness of the study which persists is that the authors do not take advantage of their simulations to make non-trivial prediction then test them experimentally rather than to rely on simulations to draw correlative conclusions.

Overall, while this is an impressive tour-de-force of loop simulations, I am not convinced that the conclusions and the numerical values generated are correct.

We understand that the reviewer has concerns regarding:

1. The amount of data in our previous rebuttal that was not included in the revised manuscript
2. The strength of the evidence for our claim that TADs are under constant extrusion
3. The lack of experimental tests for simulation predictions

We reply to these three concerns (and to two additional related comments) below:

1. The amount of data in our previous rebuttal that was not included in the revised manuscript

We agree that important data should be presented in the paper and not only the rebuttal, but kindly point out that we had already included a majority of the new information in the revised manuscript. Of the 15 figures with new information provided in the rebuttal, 9 were included in full in the manuscript and 2 in parts.

In response to the reviewer's comment, we have now also included two additional figures from the rebuttal:

- *Fig. F16, which supports the quasi-absence of open states in TADs using a model-free analysis, is now included as the new Extended Data Fig. 5c*
- *Fig. F6, which shows our predictions of cohesin re-establishment after mitosis, is now included as the new Extended Data Fig. 9k.*

We now mention these results in the following sentences:

- Results section (page 10): *"This quasi-absence of open states in TADs was confirmed by a model-free analysis of the upper bound of open state fractions, yielding maximal fractions of 18%, 4% and 5% for L1, L2 and T1, respectively (Extended Data Fig. 5c)."*
- Discussion (page 22): *"We note that these kinetic parameters are quantitatively consistent with experimental observations of cohesin binding kinetics following mitotic exit^{1,2} (Extended Data Fig. 9k)."*

The two remaining figures that were not included addressed technical details of our methodology for closing rate measurement, which we do not deem essential to our manuscript.

We thus consider that we have included the vast majority of the new data and all essential information in the revised manuscript.

2. The strength of the evidence for our claim that TADs are under constant extrusion

More specifically, the reviewer comments that:

- A. Our estimates are *"either purely based on simulations or use RAD21 degradation data"* and lack *"clear orthogonal validation of the ability to distinguish extruding and non-extruding states"*
- B. *"Cohesin does not only bind at TAD anchors but also affects chromatin organization via TAD-independent mechanisms"*
- C. *"the main conclusion regarding pervasiveness of extrusion is not convincingly supported"*

Point A: We do agree with the reviewer that our data stop short of providing entirely orthogonal demonstrations of loop extrusion and acknowledge that our quantitative estimates are based either on RAD21 depletion experiments or on comparing experimental data to simulations. Specifically, our conclusions rest on the assumptions that: (a) RAD21 extrudes loops in living cells and (b) RAD21 degradation primarily affects loop extrusion.

We note that assumption (a) is solidly supported by a wealth of evidence from multiple technical approaches, including Hi-C, ChIP-seq, biochemical assays, multiplexed DNA FISH, polymer simulations, cryo-electron microscopy and *in vitro* imaging of DNA with purified extruding complexes^{1,3-14}. Together, they provide compelling evidence for the loop extrusion mechanism.

Point B: Regarding assumption (b), we are not aware of cohesin effects independent of loop extrusion that might affect TAD anchor dynamics, but we acknowledge that we cannot entirely rule out unknown effects of RAD21 depletion. However, we made multiple efforts to reduce potential confounding factors, by (i) restricting our analyses to cells in G1 to avoid cell-cycle effects, (ii) selecting chromatin regions with low levels (or absence) of gene expression and without enhancer or promoters at TAD boundaries in order to avoid effects due to transcription, (iii) limiting our data acquisition to 2-4 hours following RAD21 degradation in order to minimize potential long-term effects of cohesin depletion, and (iv) we selected TADs located entirely within a single A compartment and distant from two neighboring B compartments to minimize potential confounding effects of compartmentalization. Therefore, we consider that our analyses rest on a solid basis.

Point C: Concerning our finding that TADs are under constant extrusion, we would like to reiterate that this is now supported by four different datasets analyzed by three completely independent methods, including one that is entirely model-free, as detailed in our previous response R1.1.f. Briefly, these analyses are:

- a. The 3-state model, which was validated on polymer simulations and applied to live-cell imaging datasets (at two distinct imaging frequencies and with two distinct levels of measurement noise) -see Extended Data Fig. 6d-f.
- b. The model-free estimation of the maximal fraction of open states (Extended Data Fig. 5c)
- c. The estimation obtained by comparisons with polymer simulations (Fig. 7a).

As stated in our previous rebuttal, this finding of pervasive loop extrusion -under the assumption that RAD21 extrudes loops in living cells- is in fact one of the most robust findings of our study.

In order to better highlight the basis of our conclusions, we now explicitly acknowledge in the Discussion that our interpretation is contingent on the above-mentioned assumptions:

- Discussion section (page 23): *“Third, our results are contingent on the assumption that loop extrusion occurs in vivo and that after two hours of depletion, RAD21 degradation affects anchor dynamics via its loop extrusion activity, as supported by multiple lines of evidence^{6,7,9-14,26,27”}*

3. The lack of experimental tests for some simulation predictions

We appreciate the reviewer’s suggestion that simulations should drive new experimental hypotheses. Indeed, our parameter estimates now enable the formulation of quantitative and experimentally testable predictions. As an example, we now include in the manuscript the prediction of cohesin recovery kinetics after mitosis, which is in excellent agreement with previous experimental measurements^{1,2}, as the new Extended Data Fig. 9k (see point 1 above). Other interesting avenues include: (i) scaling our inferences to a genome-wide prediction of TAD dynamics based on Hi-C, DNA sequence and ChIP-Seq data of CTCF, calibrated with our live-cell imaging data, (ii) assessing how the frequent but transient contacts created by loop extrusion affect transcription (e.g. through enhancer-promoter interactions, nucleation of ‘transcriptional

hubs' or changes in transcription factor diffusion), or (iii) using simulations to evaluate whether the uniform cohesin dynamics, as inferred under steady-state conditions, can also explain perturbed states such as during DNA repair. While these are exciting questions, addressing them would represent entirely new projects by themselves and is thus far beyond the scope of this study. Nevertheless, by quantitatively characterizing TAD dynamics, our work now opens the door to such follow-up studies.

Additional points:

Concerning the comment that "*Cohesin does not only bind at TAD anchors*", we would like to clarify that our analyses do not assume that cohesin binds only at TAD boundaries. Instead, our simulations assume that cohesin binds randomly on the entire genome and extrudes DNA irrespective of its location on chromatin.

Concerning the comment that "*The authors have not demonstrated that they can determine whether a loop is extruding or not or that they can distinguish extruding from closed loops*", we would like to recall that we do not claim to identify closed loops, but instead identify proximal states as proxies of closed states, as previously detailed in the rebuttal (see R1.2.f and Extended Data Fig. 10). Additionally, we recall that we quantitatively assessed our ability to distinguish open, extruding and proximal states using polymer simulations (see R1.1.f in the previous rebuttal and Extended Data Fig. 5g), assuming the two above-mentioned hypotheses (see point 2).

In summary, we added critical data to the manuscript, provided converging evidence from multiple approaches in support of constant loop extrusion, and further clarified the assumptions underlying our findings. We believe that this addresses the reviewer's concerns as much as realistically possible within the scope of the present study.

Reviewer #2 (Remarks to the Author):

I am impressed by the amount of new insights Sabaté and colleagues collected to address Reviewers' comment. The paper has greatly improved. I believe that this paper is cutting-edge, and with these revision it is now ready for publication since they addressed all points.

We thank the reviewer for these positive comments.

Reviewer #2 (Remarks on code availability):

A last advice is to upload the code on zenodo as well, together with the microscopy revised data. The code is sufficiently documented.

We agree with the reviewer and we have now uploaded the code, together with the revised microscopy data, in a new version of the Zenodo repository with the associated DOI: 10.5281/zenodo.15723942.

Reviewer #3 (Remarks to the Author):

I have read the revised manuscript by Sabaté et al. with great interest, and found that the comments raised by both Reviewers helped the Authors to greatly improve its quality. Indeed, the Authors did a fantastic job replying to their comments. I find the work solid, and I appreciate that the Authors have tuned down their overstatements, which were pointed out by the Reviewers.

I agree with the Reviewers that the work is an extension (and a confirmation) of the two previous works, this time done in a human cell line and on 3 different loci. I do think that the fact that the previous findings were confirmed using a different approach is of importance (especially given the relevance of the topic) and given that I find the quality of the data high, I am in favour of accepting the work in its current form. In my mind the novelty is that the various parameters linked to the loop dynamics seem to remain pretty stable when we start adding additional DNA loci to those measurements (albeit with relatively broad ranges). And even though it might not be a major leap forward, given how tricky these measurements are, obtaining those values in as robust manner as the Authors achieved in this work seems to me very valuable.

My only concern are those rather broad ranges, which the Reviewer #1 is also critical about. The Authors argue against it, but in reality this work does not bring us significantly closer to more precise estimates. Perhaps the Authors could still discuss this issue more in the Discussion section (using some of the reasoning from the rebuttal letter).

We thank the reviewer for these encouraging comments and his/her interest in our findings. We appreciate in particular that the reviewer considers the relative stability of the parameters across multiple loci as the main novelty.

We would like to address the reviewer's remaining concern regarding the ranges of the estimated parameters. We provide estimates for three independent parameters of loop dynamics: cohesin density, residence time and extrusion speed.

Concerning our estimates for cohesin density and the residence time, we would like to stress that they range from 12 to 18 Mb⁻¹ and from 15 to 25 minutes, respectively, as we pointed out in the previous rebuttal. The corresponding ratios between the maximum and minimum are 1.5 and 1.7, respectively. We consider these estimates rather narrow. From these two parameter estimates, we inferred the cohesin loading rate ranging from 0.5 to 1.2 (Mb x min)⁻¹, which is larger but still 40% narrower than a previous estimate by Mach *et al.*¹¹.

We also note that these estimates allow us to make non-trivial predictions about cohesin re-establishment after mitotic exit, in excellent agreement with experimental data^{1,2}, as shown in our new Extended Data Fig. 9k. We have added the following sentence in the Discussion section, to highlight the implications of these ranges:

- Discussion (page 23): “We note that these kinetic parameters are quantitatively consistent with experimental observations of cohesin binding kinetics following mitotic exit^{1,2} (Extended Data Fig. 9k).”

Concerning the cohesin motor speed *in vivo*, however, we admittedly provided only an upper bound of 0.5 kb/s, leaving a rather large range. Although speeds of ~0.2 kb/s provided the best agreement with experimental data (Fig. 5d,e) at a fixed cohesin residence time of 22 min and density of 12 Mb⁻¹, narrowing down the range of motor speeds further would require a systematic exploration of the whole 3D parameter space in our polymer simulations (*i.e.* varying cohesin density, residence time and motor speeds and testing all combinations independently). Unfortunately, this is unfeasible with the computational resources that we have access to. Please note that the computation time on our Institute's HPC cluster for the simulations used in our article was already about 5 months, with around 400 CPUs operating at full capacity (without parallelization, the simulations would thus have taken 166 years). Given the computational constraints of our analyses and rather than reporting a potentially misleading point estimate, we therefore chose to report a conservative upper bound to the motor speed.

We acknowledge this limitation in the Discussion in the following sentence:

- Discussion (p. 23): *“Fourth, our analyses only provide an upper bound to the motor speed, in contrast to the ranges estimated for cohesin density and residence time. A more precise estimate would require a systematic and computationally much more intensive exploration of the parameter space in polymer simulations.”*

References:

1. Rao, S. S. P. *et al.* Cohesin Loss Eliminates All Loop Domains. *Cell* **171**, 305-320.e24 (2017).
2. Zhang, H. *et al.* Chromatin structure dynamics during the mitosis-to-G1 phase transition. *Nature* **576**, 158–162 (2019).
3. Bintu, B. *et al.* Super-resolution chromatin tracing reveals domains and cooperative interactions in single cells. *Science* **362**, eaau1783 (2018).
4. Fudenberg, G. *et al.* Formation of Chromosomal Domains by Loop Extrusion. *Cell Rep.* **15**, 2038–2049 (2016).
5. Davidson, I. F. *et al.* DNA loop extrusion by human cohesin. *Science* **366**, 1338–1345 (2019).
6. Kim, Y., Shi, Z., Zhang, H., Finkelstein, I. J. & Yu, H. Human cohesin compacts DNA by loop extrusion. *Science* **366**, 1345–1349 (2020).
7. Nuebler, J., Fudenberg, G., Imakaev, M., Abdennur, N. & Mirny, L. A. Chromatin organization by an interplay of loop extrusion and compartmental segregation. *Proc. Natl. Acad. Sci. U. S. A.* **115**, E6697–E6706 (2018).
8. Sanborn, A. L. *et al.* Chromatin extrusion explains key features of loop and domain formation in wild-type and engineered genomes. *Proc. Natl. Acad. Sci. U. S. A.* **112**, E6456–6465 (2015).
9. Davidson, I. F. *et al.* CTCF is a DNA-tension-dependent barrier to cohesin-mediated loop extrusion. *Nature* **616**, 822–827 (2023).
10. Gabriele, M. *et al.* Dynamics of CTCF- and cohesin-mediated chromatin looping revealed by live-cell imaging. *Science* **376**, 496–501 (2022).
11. Mach, P. *et al.* Cohesin and CTCF control the dynamics of chromosome folding. *Nat. Genet.* **54**, 1907–1918 (2022).
12. Shaltiel, I. A. *et al.* A hold-and-feed mechanism drives directional DNA loop extrusion by condensin. *Science* **376**, 1087–1094 (2022).

13. Nomidis, S. K., Carlon, E., Gruber, S. & Marko, J. F. DNA tension-modulated translocation and loop extrusion by SMC complexes revealed by molecular dynamics simulations. *Nucleic Acids Res.* **50**, 4974–4987 (2022).
14. Srinivasan, M. *et al.* The Cohesin Ring Uses Its Hinge to Organize DNA Using Non-topological as well as Topological Mechanisms. *Cell* **173**, 1508-1519.e18 (2018).